# Predicting Ground State Properties: Constant Sample Complexity and Deep Learning Algorithms

**Marc Wanner**
Computer Science and Engineering
Chalmers University of Technology
and University of Gothenburg
wanner@chalmers.se

**Laura Lewis**
Applied Mathematics and Theoretical Physics
University of Cambridge
Cambridge, United Kingdom
llewis@alumni.caltech.edu

**Chiranjib Bhattacharyya**
Computer Science and Automation
Indian Institute of Science
Bangalore, India
chiru@iisc.ac.in

**Devdatt Dubhashi**
Computer Science and Engineering
Chalmers University of Technology
and University of Gothenburg
dubhashi@chalmers.se

**Alexandru Gheorghiu**
Computer Science and Engineering
Chalmers University of Technology
and University of Gothenburg
aleghe@chalmers.se

## Abstract

A fundamental problem in quantum many-body physics is that of finding ground states of local Hamiltonians. A number of recent works gave *provably efficient* machine learning (ML) algorithms for learning ground states. Specifically, Huang et al. in [1], introduced an approach for learning properties of the ground state of an $n$-qubit gapped local Hamiltonian $H$ from only $n^{\mathcal{O}(1)}$ data points sampled from Hamiltonians in the same phase of matter. This was subsequently improved by Lewis et al. in [2], to $\mathcal{O}(\log n)$ samples when the geometry of the $n$-qubit system is known. In this work, we introduce two approaches that achieve a *constant* sample complexity, independent of system size $n$, for learning ground state properties. Our first algorithm consists of a simple modification of the ML model used by Lewis et al. and applies to a property of interest known in advance. Our second algorithm, which applies even if a description of the property is not known, is a deep neural network model. While empirical results showing the performance of neural networks have been demonstrated, to our knowledge, this is the first rigorous sample complexity bound on a neural network model for predicting ground state properties. We also perform numerical experiments on systems of up to 45 qubits that confirm the improved scaling of our approach compared to [1, 2].

## 1 Introduction

One of the most important problems in quantum many-body physics is that of finding ground states of quantum systems. This is due to the fact that the ground state describes the behavior of electronic systems (e.g., metals, magnets, etc.) at room temperature well. Thus, understanding

38th Conference on Neural Information Processing Systems (NeurIPS 2024).

the ground state can provide insights into, for example, chemical properties of molecules, leading to many potential applications in chemistry and materials science. However, despite extensive research [3, 4, 5, 6, 7, 8, 9, 10, 11, 12, 13, 14], an efficient classical algorithm solving this problem in full generality remains out of reach. On the other hand, researchers have successfully leveraged classical *machine learning* (ML) techniques to solve (albeit largely heuristically) the ground state problem and other related quantum many-body problems [15, 16, 17, 18, 19, 20, 21, 22, 23, 24, 25, 26, 27, 28, 29, 30, 31, 32, 33, 34, 35, 36, 37, 38, 39, 40, 41, 42, 43, 44, 45, 46]. Rather than solving these problems directly from first principles, ML algorithms are given some training data collected from physical experiments and are asked to generalize it to new inputs. Intuitively, this additional data can make the problem easier and thus may open the door to obtaining provably efficient classical ML algorithms for finding ground states. This data-driven approach is in some sense necessary, since finding the ground state from the Hamiltonian alone is known to be QMA-hard in general [47], and thus out of reach for both efficient classical and quantum algorithms.

In a recent work [1], Huang et al. proposed the first *provably efficient* ML algorithm for predicting ground state properties of gapped geometrically local Hamiltonians. In particular, the algorithm in [1] uses an amount of training data (or *sample complexity*) that scales as $\mathcal{O}(n^{1/\epsilon})$, where $n$ is the system size and $\epsilon$ is the prediction error of the ML algorithm. Recently, [2] improved this guarantee, achieving $\mathcal{O}(\log(n)2^{\mathrm{polylog}(1/\epsilon)})$, an *exponential improvement* with respect to the system size $n$. The same sample complexity was obtained by [48] for the task of learning thermal state properties with exponential decay of correlations. Moreover, [49] extended this to Lindbladian phases of matter [50] with local rapid mixing, including both ground states of gapped Hamiltonians and thermal states. The work of [51] obtains a similar guarantee assuming the continuity of quantum states in the parameter range of interest but focusing on the scaling with respect to $1/\epsilon$ rather than system size.

These previous works drastically improve the sample complexity of the original Huang et al. result [1], but none prove sample complexity *lower bounds* for their respective tasks, leaving open the possibility of further reducing the sample complexity. In addition, [2, 48, 49] all use fairly simple learning models, i.e., regularized linear regression or taking empirical averages of classical shadows [52], respectively. With the emergence of neural networks as a popular model in practical ML, one may wonder if these more powerful ML tools may be useful to predict ground state properties as well. In fact, recent works [37, 36] empirically demonstrate a favorable sample complexity using neural-network-based ML algorithms. However, there are currently no rigorous theoretical guarantees regarding the amount of training data needed to achieve a desired prediction error. These remarks lead us to the following two central questions of this work.

**Question 1.** *Can classical ML algorithms predict ground state properties with even less than* $\mathcal{O}(\log(n)2^{\mathrm{polylog}(1/\epsilon)})$ *data?*

This is especially relevant for systems approaching the thermodynamic limit, where the system size can be arbitrarily large. Needing fewer samples also means less work for experimentally preparing ground states of the system. The second question, stated as an open question in [1, 2] is:

**Question 2.** *Can we obtain rigorous sample complexity guarantees for neural-network-based ML algorithms for predicting ground state properties?*

**Our results**

We give positive answers to both questions. We consider the same assumptions as [2] with minimal additional ones that we mention here. First, we show that a simple modification to the approach in [2] allows us to achieve a sample complexity that is *independent of the system size*. This does, however require knowledge of the property we wish to predict in advance, whereas this is not a requirement in [2]. We view this as a reasonable assumption, since in practice we can imagine preparing ground states of some system in order to measure a specific property of interest. We show the following theorem, stated informally here. The formal statement, including all the assumptions required for proving the result, can be found in Appendix B.

**Theorem 1** (Informal)**.** *Let $H(x)$ be an $n$-qubit gapped, geometrically local Hamiltonian with ground state $\rho(x)$. Given an observable $O$, with a known decomposition as a sum of local Pauli operators and given training data $\{(x_\ell, y_\ell)\}_{\ell=1}^N$ sampled from an arbitrary distribution, with $y_\ell \approx \mathrm{tr}(O\rho(x_\ell))$,*

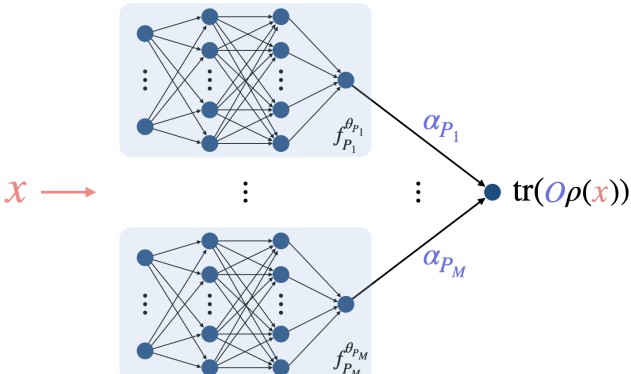

Figure 1: **A deep network model for predicting ground state properties.** Given a vector $x \in [-1, 1]^m$ that parameterizes a quantum many-body Hamiltonian $H(x)$, the algorithm uses geometric structure to create "local" neural network models $f_{P_i}^{\theta_{P_i}}$. The ML algorithm then combines the outputs of these local models to predict a property $\text{tr}(O\rho(x))$, where $\rho(x)$ is the ground state of $H(x)$. Here, we decompose $O = \sum_{i=1}^{M} \alpha_{P_i} P_i$ for Pauli operators $P_i$, where the final layer takes a linear combination of the outputs of the local models weighted by some trainable parameters $w_{P_i}$ that intuitively should approximate the Pauli coefficients $\alpha_{P_i}$.

*there is an ML algorithm for predicting ground state properties* $\text{tr}(O\rho(x))$ *to within precision* $\epsilon > 0$ *using* $N = \mathcal{O}\left(2^{\text{polylog}(1/\epsilon)}\right)$ *training samples.*

Note that the number of samples $N$ depends only on the desired prediction error $\epsilon$ and is independent of the system size. In particular, this means that for a fixed prediction error our algorithm requires only a *constant* amount of training data. Moreover, the computational complexity of our algorithm improves upon [2], having $\mathcal{O}(n)$ runtime, compared to the previous $\mathcal{O}(n \log n)$. While removing the $\log n$ factor may seem like a small improvement, in practice this can make a significant difference. For instance, for a system of $n \sim 1000$ qubits, removing the $\log n$ factor would result in a ten-fold reduction in training data and time.

Much like in [2], this result also extends to learning classical representations of $\rho(x)$. In other words, if the algorithm is instead given *classical shadows* [52] of the ground state as training data, it can then predict a classical representation of $\rho(x)$ for new parameters $x$. This can mitigate the requirement that the observable is known in Theorem 1, as predicting properties from a classical representation clearly requires knowledge of the observable.

Our second result shows the same sample complexity guarantee for a *neural network* ML algorithm (Figure 1) [53], in which one does not need to know the observable being measured in advance. An additional constraint that we require in this case is that the training data is not sampled according to an arbitrary distribution, but a distribution satisfying some technical assumptions. We note that these assumptions are satisfied for common distributions such as uniform and Gaussian.

**Theorem 2** (Informal). *Let $H(x)$ be an $n$-qubit gapped, geometrically local Hamiltonian with ground state $\rho(x)$. For any observable $O$, expressible as a sum of local Pauli operators and given training data $\{(x_\ell, y_\ell)\}_{\ell=1}^{N}$, sampled from a distribution satisfying certain assumptions with $y_\ell \approx \text{tr}(O\rho(x_\ell))$, there is a neural network ML algorithm for predicting ground state properties $\text{tr}(O\rho(x))$, for uniform $x$, to within precision $\epsilon > 0$ using $N = \mathcal{O}\left(2^{\text{polylog}(1/\epsilon)}\right)$ training samples under mild assumptions on training.*

We prove this result by making use of the Koksma-Hlawka inequality from quasi-Monte Carlo theory [54, 55, 56, 57, 58] and combining it with the spectral flow formalism [59, 60, 61].

Similar to Theorem 1, we can also extend this result to learning classical representations of $\rho(x)$ when given classical shadow training data. The formal statement and its proof can be found in Appendix C. We also remark that, much like the setting in [51], a more favorable scaling with respect to $\epsilon$ can be achieved if the number of parameters that the Hamiltonian depends on is constant. In particular, it was shown in [51] that if the number of parameters is constant, the sample complexity scales

as $N = \mathsf{poly}(1/\epsilon, \log(n))$. For our results, this similarly yields $N = \mathsf{poly}(1/\epsilon)$, preserving the independence on the system size while also achieving a polynomial scaling in $1/\epsilon$.

Furthermore, we perform numerical experiments on system sizes of up to 45 qubits, which support our theoretical findings, and show that, in practice, our deep learning algorithm outperforms previous methods [2]. We describe them in detail in Appendix D, and they are illustrated in Figure 2.

## 2 Preliminaries

### 2.1 Problem statement

First, we formally describe the problem setting, which is the same as [2]. We consider a family of $n$-qubit Hamiltonians $H(x)$ smoothly parameterized by an $m$-dimensional vector $x \in [-1, 1]^m$. We assume that these Hamiltonians are gapped for all choices of parameters $x \in [-1, 1]^m$ and geometrically local such that they can be written as a sum of local terms

$$H(x) = \sum_{j=1}^{L} h_j(\vec{x}_j), \tag{2.1}$$

where the parameter vector $x$ is a concatenation of the constant-dimensional vectors $\vec{x}_1, \ldots, \vec{x}_L$. Each of these constant-dimensional vectors $\vec{x}_j$ parameterizes the local interaction term $h_j(\vec{x}_j)$. Crucially, we assume that each local term $h_j$ only depends on a constant number of parameters rather than the entire parameter vector $x$. We also assume that the geometry of the $n$-qubit system is known.

Throughout this work, we use $\rho(x)$ to denote the ground state of the Hamiltonian $H(x)$ and $O$ to denote an observable that can be written as a sum of geometrically local observables with bounded spectral norm $\|O\|_\infty \leq 1$. Here, the ground states $\rho(x)$ form a gapped quantum phase of matter. Given samples of quantum states drawn from this phase, we wish to predict expectation values of observables $O$ with respect to other states in the same phase. In other words, we are given training data $\{(x_\ell, y_\ell)\}_{\ell=1}^{N}$, where $y_\ell \approx \mathrm{tr}(O\rho(x_\ell))$ approximates the ground state property for a parameter choice $x_\ell \in [-1, 1]^m$ sampled from some distribution $\mathcal{D}$ over the parameter space. We aim to learn a function $h^*(x)$ that approximates the ground state property $\mathrm{tr}(O\rho(x))$ for some unseen parameter $x$ while minimizing the amount of training data, or sample complexity, $N$. How well we learn the ground state property is quantified by the average prediction error

$$\mathop{\mathbb{E}}_{x \sim \mathcal{D}} |h^*(x) - \mathrm{tr}(O\rho(x))|^2 \leq \epsilon. \tag{2.2}$$

We describe the precise conditions under which Theorems 1 and 2 hold in the following sections.

### 2.2 Review of previous algorithm

In this section, we review the previous algorithm from [2], as our proofs rely on similar ideas. For full details, we refer the reader to [2] and our more detailed presentation in Appendix A.1. The ML algorithm proposed in [2] requires some geometric definitions. Fix a geometrically local Pauli observable $P \in \{I, X, Y, Z\}^{\otimes n}$.

Let $\delta_1, B > 0$ be efficiently-computable hyperparameters that we define in Appendix A.1. Define the set $I_P$ of coordinates $c$ such that $x_c$ parameterizes some local term $h_{j(c)}$ that is close to the Pauli $P$. Here, the distance between two observables $d_{\mathrm{obs}}$ is defined as the minimum distance between the qubits that the observables act on, where the distance between qubits is given by the geometry of the system, which we assume to be known. Formally, we define this set of local coordinates as

$$I_P \triangleq \{c \in \{1, \ldots, m\} : d_{\mathrm{obs}}(h_{j(c)}, P) \leq \delta_1\}, \tag{2.3}$$

where $h_{j(c)}$ is the local term in the Hamiltonian $H(x)$ whose parameters $\vec{x}_{j(c)}$ include the variable $x_c$. The intuition behind this set of coordinates is that it indexes the parameters $x_c$ that influence the ground state property $\mathrm{tr}(P\rho(x))$ corresponding to the Pauli $P$. The algorithm consists of two steps. First, it maps the parameter space $[-1, 1]^m$ to a high dimensional space via a nonlinear feature map $\phi$. Second, it runs $\ell_1$-regularized linear regression (LASSO) [62, 63, 64] over the feature space.

This first step encodes the geometry of the problem. The feature map intuitively projects a given parameter vector $x$ onto the local parameter space $\{x_c : c \in I_P\}$. We define this precisely in

Appendix A.1. Following the feature mapping, the ML algorithm uses LASSO [62, 63, 64] to learn functions of the form $\{h(x) = \mathbf{w} \cdot \phi(x) : \|\mathbf{w}\|_1 \leq B\}$ for a chosen hyperparameter $B > 0$. We denote the learned function by $h^*(x) = \mathbf{w}^* \cdot \phi(x)$. For our purposes, we set $B = 2^{\mathcal{O}(\text{polylog}(1/\epsilon_1))}$. This algorithm obtains the following rigorous guarantee.

**Theorem 3** (Theorem 1 in [2]). *Given $n, \delta > 0$, $\frac{1}{e} > \epsilon > 0$ and a training data set $\{(x_\ell, y_\ell)\}_{\ell=1}^N$ of size*

$$N = \log(n/\delta)2^{\text{polylog}(1/\epsilon)}, \tag{2.4}$$

*where $x_\ell$ is sampled from an unknown distribution $\mathcal{D}$ and $|y_\ell - \text{tr}(O\rho(x_\ell))| \leq \epsilon$. With a proper choice of the efficiently computable hyperparameters $\delta_1, \delta_2$, and $B$, the learned function $h^*(x) = \mathbf{w}^* \cdot \phi(x)$ satisfies*

$$\mathbb{E}_{x \sim \mathcal{D}} |h^*(x) - \text{tr}(O\rho(x))|^2 \leq \epsilon \tag{2.5}$$

*with probability at least $1 - \delta$. The training and prediction time of the classical ML model are bounded by $\mathcal{O}(nN) = n\log(n/\delta)2^{\text{polylog}(1/\epsilon)}$.*

A crucial step in the proof is that ground state properties can indeed be approximated by linear functions over the feature space. Along the way, [2] proves that the ground state property can be approximated by a linear combination of "local functions," which are local in that they only depend on parameters with coordinates in the set $I_P$. We relegate further details to Appendix A.1 and [2].

## 3 Main results

In this section, we discuss our rigorous guarantees for predicting ground state properties with constant sample complexity and with neural-network-based ML algorithms.

### 3.1 Constant sample complexity

In this section, we show that a simple modification of the algorithm from [2] can achieve a sample complexity that is independent of the system size $n$, under the additional assumption that the observable $O$ is known. In practice, a scientist often has a specific ground state property in mind that they wish to study, so we view this as a natural assumption. Moreover, this is still an interesting learning problem, as when obtaining the training data via quantum experiments, preparing the ground state $\rho(x)$ in the laboratory for a new choice of parameters $x$ may be difficult experimentally. This in turn means that accurately predicting some property $\text{tr}(O\rho(x))$ for a new choice of $x$ may be challenging, even if the property of interest, $O$, is known. ML algorithms can allow us to circumvent this issue and generalize from the results of few training data points without needing to prepare the ground state directly.

Let $O = \sum_{P \in \{I,X,Y,Z\}^{\otimes n}} \alpha_P P$ be an observable that can be written as a sum of geometrically local observables. Because $O$ is assumed to be known, we can find this decomposition of $O$ in terms of the Pauli observables $P$. The overall structure of the algorithm remains the same: perform a nonlinear feature mapping followed by linear regression. However, there are two key differences from the previous algorithm [2]. First, we change the feature mapping of [2] to incorporate the Pauli coefficients $\alpha_P$. We define this new feature mapping $\tilde{\phi}$ in Appendix B. The second difference from [2] is that we use ridge regression [65, 66] instead of LASSO [62, 63, 66]. Recall that LASSO learns hypothesis functions of the form $\{h(x) = \mathbf{w} \cdot \tilde{\phi}(x) : \|\mathbf{w}\|_1 \leq B\}$ for some hyperparameter $B > 0$. In contrast, ridge regression replaces the $\ell_1$-norm constraint $\|\mathbf{w}\|_1 \leq B$ with an $\ell_2$-norm constraint: $\|\mathbf{w}\|_2 \leq \Lambda$, for some hyperparameter $\Lambda > 0$. Namely, for a chosen efficiently-computable hyperparameter $\Lambda > 0$, ridge regression finds a vector $\mathbf{w}^*$ that minimizes the training error subject to the constraint that $\|\mathbf{w}\|_2 \leq \Lambda$, i.e.,

$$\min_{\substack{\mathbf{w} \in \mathbb{R}^{m_\phi} \\ \|\mathbf{w}\|_2 \leq \Lambda}} \frac{1}{N} \sum_{\ell=1}^N |\mathbf{w} \cdot \tilde{\phi}(x_\ell) - y_\ell|^2. \tag{3.1}$$

Standard results in machine learning theory give sample complexity upper bounds for ridge regression in terms of $\Lambda$ and the $\ell_2$-norm of the feature vector $\tilde{\phi}(x)$ [65, 66]. The key idea is that with our new feature mapping, we can still approximate the ground state property by a linear function over

the feature space, as in [2], to obtain a low training error. Meanwhile, by incorporating the Pauli coefficients $\alpha_P$ into the feature map, we can bound the $\ell_2$-norm of $\tilde{\phi}(x)$ by a quantity independent of system size, leveraging bounds on the $\ell_1$-norm of the Pauli coefficients [2, 67]. We note that naively applying ridge regression with the feature map from [2] does not achieve the same guarantees and in fact gives worse scaling than [2]. Similarly, we can also choose a suitable $\Lambda > 0$ independent of system size. Thus, we obtain the following guarantee.

**Theorem 4** (Constant sample complexity). *Given* $n, \delta > 0$, $1/e > \epsilon > 0$ *and a training data set* $\{(x_\ell, y_\ell)\}_{\ell=1}^N$ *of size*

$$N = \log(1/\delta)2^{\mathrm{polylog}(1/\epsilon)}, \tag{3.2}$$

*where* $x_\ell$ *is sampled from an unknown distribution* $\mathcal{D}$ *and* $|y_\ell - \mathrm{tr}(O\rho(x_\ell))| \leq \epsilon$. *With a proper choice of the efficiently computable hyperparameters* $\delta_1, \delta_2, \Lambda$, *the learned function* $h^*(x)$ *satisfies*

$$\mathop{\mathbb{E}}_{x \sim \mathcal{D}} |h^*(x) - \mathrm{tr}(O\rho(x))|^2 \leq \epsilon \tag{3.3}$$

*with probability least* $1 - \delta$. *The training and prediction time of the classical ML model are bounded by* $\mathcal{O}(n)\mathrm{polylog}(1/\delta)2^{\mathrm{polylog}(1/\epsilon)}$.

We compare this result to Theorem 3. For a constant prediction error $\epsilon = \mathcal{O}(1)$, our proposed algorithm achieves a constant sample complexity $N = \mathcal{O}(1)$, compared to the logarithmic sample complexity $N = \mathcal{O}(\log n)$ of [2]. Moreover, we also improve the computational complexity, achieving a linear-in-$n$ runtime, compared to the previous $\mathcal{O}(n \log n)$. The scaling with respect to the prediction error $\epsilon$ is the same as the previous algorithm [2]. This means that regardless of how large our quantum system is, we need the same amount of samples to predict ground state properties well. This is especially important for settings in which obtaining training data for large systems is difficult.

Thus far, we have only considered the setting in which we learn a specific ground state property $\mathrm{tr}(O\rho(x))$ for a fixed observable $O$. Because our training data is given in the form $\{(x_\ell, y_\ell)\}_{\ell=1}^N$, where $y_\ell$ approximates $\mathrm{tr}(O\rho(x))$ for this fixed observable $O$, if we want to predict a new property for the same ground state $\rho(x)$, we would need to generate new training data. Thus, it may be more useful to learn a ground state representation, from which we could predict $\mathrm{tr}(O\rho(x))$ for many different choices of observables $O$ without requiring new training data. In this case, suppose we are instead given training data $\{x_\ell, \sigma_T(\rho(x_\ell))\}_{\ell=1}^N$, where $\sigma_T(\rho(x_\ell))$ is a classical shadow representation [52, 68, 69, 70, 71] of the ground state $\rho(x_\ell)$. An immediate corollary of Theorem 4 is that we can predict ground state representations with the same sample complexity. This follows from the same proof as Corollary 5 in [2].

**Corollary 1** (Learning representations of ground states). *Let* $n, \delta > 0$, $1/e > \epsilon > 0$ *and* $\delta > 0$. *Given training data* $\{(x_\ell, \sigma_T(\rho(x_\ell))\}_{\ell=1}^N$ *of size*

$$N = \log(1/\delta)2^{\mathcal{O}(\mathrm{polylog}(1/\epsilon))}, \tag{3.4}$$

*where* $x_\ell$ *is sampled from* $\mathcal{D}$ *and* $\sigma_T(\rho(x_\ell)$ *is the classical shadow representation of the ground state* $\rho(x_\ell)$ *using* $T$ *randomized Pauli measurements. For* $T = \tilde{\mathcal{O}}(\log(n/\delta)/\epsilon^2)$, *with probability at least* $1 - \delta$, *the ML algorithm will produce a ground state representation* $\hat{\rho}_{N,T}(x)$ *that achieves*

$$\mathop{\mathbb{E}}_{x \sim \mathcal{D}} |\mathrm{tr}(O\hat{\rho}_{N,T}(x)) - \mathrm{tr}(O\rho(x))|^2 \leq \epsilon \tag{3.5}$$

*for any observable with* $\|O\|_\infty \leq 1$ *that can be written as a sum of geometrically local observables.*

### 3.2 Rigorous guarantees for neural networks

In this section, we prove the existence of a deep neural network model that can predict ground state properties using a constant number of training samples. In particular, we prove that after training on a constant number of samples from a distribution $\mathcal{D}$ on $[-1, 1]^m$ satisfying certain technical assumptions, our model can achieve a low prediction error under mild assumptions on training. In this case, for predicting properties $\mathrm{tr}(O\rho(x))$, the observable $O$ need not be known in advance. However, we need to assume that all mixed first order derivatives of the Hamiltonian $\|\partial^m H(x)/\partial x_1 \cdots \partial x_m\|_\infty \leq 1$ exist and are bounded. This is not much stronger than [2], which assumes that directional derivatives $\partial h_j/\partial \hat{u}$ are bounded by one for any direction $\hat{u}$. Moreover, we

also need the training data to be sampled from a distribution $\mathcal{D}$ with probability density function $g$ satisfying the following assumptions: $g$ has full support and is continuously differentiable on $[-1, 1]^m$. Also, $g$ is of the form $g(x) = \prod_{j=1}^{L} g_j(\vec{x}_j)$. This resembles our assumption on the form of the Hamiltonian $H(x)$. Furthermore, the average prediction error is measured with respect to the same distribution $\mathcal{D}$. We note that these assumptions are satisfied for common distributions such as uniform and Gaussian.

As in the previous algorithm [2], we leverage the geometry of the $n$-qubit system to approximate the ground state properties by a linear combination of smooth local functions, which only depend on parameters with coordinates in the local coordinate set $I_P$ defined in Equation (2.3). Crucially, the size $\tilde{m} \triangleq |I_P|$ of the domains of these local functions is independent of the system size.

Instead of using a feature map and linear regression to learn the ground state properties, we utilize a deep neural network model defined as follows. Inspired by the local approximation of ground state properties, we define "local models" $f_P^{\theta_P} : [-1, 1]^{\tilde{m}} \to \mathbb{R}$, which are neural networks consisting of three layers of affine transformations and applications of a nonlinear activation function. In particular, $f_P^{\theta_P}$ has two hidden layers with the affine transformations given by the trainable weights and biases denoted by $\theta_P$. We take hyperbolic tangent, $\tanh$, as the activation function. These local models are then combined into a model $f^{\Theta, w} : [-1, 1]^m \to \mathbb{R}$ given by

$$f^{\Theta, w}(x) = \sum_{P \in S^{(\text{geo})}} w_P f_P^{\theta_P}(x), \tag{3.6}$$

where $w_P \in \mathbb{R}$ are the weights in the last layer and $\Theta = \{\theta_P\}_{P \in S^{(\text{geo})}}$. This model is schematically illustrated in Figure 1. We refer to Definition 6 in Appendix C for a full description of the model.

Consider training data $\{(x_\ell, y_\ell)\}_{\ell=1}^{N}$, where $x_\ell$ are sampled according to a distribution $\mathcal{D}$ satisfying the assumptions described above and $|y_\ell - \text{tr}(O\rho(x_\ell))| \leq \epsilon$. The ML algorithm first initializes the weights via standard deep learning initialization procedures, e.g., Xavier initialization [72]. Then, the algorithm performs quasi-Monte Carlo training given the training data, e.g., Adam [73], to find weights $\Theta^*, w^*$ which minimize the training objective function

$$\frac{1}{N} \sum_{\ell=1}^{N} |f^{\Theta, w}(x_\ell) - y_\ell|^2 + \lambda \|w\|_1, \tag{3.7}$$

where $\lambda$ is some regularization parameter that may depend on $\epsilon$. For this algorithm, we prove the following theorem bounding the average prediction error of our deep neural network model.

**Theorem 5** (Neural network sample complexity guarantee). *Let $1/e > \epsilon > 0$. Let $\mathcal{D}$ be a distribution with probability density function $g$ satisfying the properties stated above. Let $f^{\Theta^*, w^*} : [-1, 1]^m \to \mathbb{R}$ be a neural network model trained on data $\{(x_\ell, y_\ell)\}_{\ell=1}^{N}$ of size*

$$N = \mathcal{O}\left(\log(1/\delta) 2^{\text{polylog}(1/\epsilon)}\right), \tag{3.8}$$

*where the $x_\ell$'s are sampled from $\mathcal{D}$ and $|y_\ell - \text{tr}(O\rho(x_\ell))| \leq \epsilon$. Suppose that $f^{\Theta^*, w^*}$ achieves a value no larger than $\mathcal{O}(\epsilon)$ on the training objective (Equation (3.7)) with $\lambda(\epsilon) = \mathcal{O}(\epsilon)$. Additionally, suppose that all parameters $\Theta_i^*$ of $f^{\Theta^*, w^*}$ satisfy $|\Theta_i^*| \leq W_{\max}$, for some $W_{max} > 0$ that is independent of $n$. Then*

$$\mathbb{E}_{x \sim \mathcal{D}} |f^{\Theta^*, w^*}(x) - \text{tr}(O\rho(x))|^2 \leq \epsilon. \tag{3.9}$$

Similar to Theorem 4, for a constant prediction error $\epsilon = \mathcal{O}(1)$, the deep neural network algorithm achieves constant sample complexity $N = \mathcal{O}(1)$. In contrast to Theorem 4, we do not require knowledge about the observable, $O$. This is a direct consequence of the regularity of $w_P$, which is achieved when the training objective is small. Theorem 6 guarantees, that a model with such regularity can yield a small prediction error.

There are, however, some caveats compared to the previous result. First, the training data is restricted to being sampled from a distribution satisfying our technical assumptions stated previously, in contrast to Theorem 4 which holds for data sampled from any arbitrary unknown distribution. Second, in regards to the model, the weights must be bounded by a constant $W_{\max}$. Finally, we cannot guarantee

*a priori* that the network will indeed achieve a low training error. This is due to the fact that our training objective is non-convex and thus, globally optimal weights cannot be found efficiently in general [74]. Even so, we are still able to prove the existence of suitable weights such that the resulting network approximates $\text{tr}(O\rho(x))$ for any $x \in [-1,1]^m$ (see Theorem 6 in the next section).

However, we view the assumptions made in Theorem 5 as being mild in practice. Small training objectives are commonly achieved in deep learning so we expect our training algorithm to produce a model which fulfills the assumptions of Theorem 5 after $\mathcal{O}(1)$ training steps and $\mathcal{O}(n)$ runtime. Moreover, it is known that gradient descent provably converges to the global optimum for *overparametrized* deep neural networks, while the weights remain small, when properly initialized [75]. We verify that these conditions are satisfied in practice through our numerical experiments in Figure 2 and **??** . To our knowledge, Theorem 5 is the first rigorous sample complexity bound on a neural network model for predicting ground state properties.

We also note that if the training data is instead sampled according to a *low-discrepancy sequence* (LDS) [55, 56, 76, 77, 78, 57, 79, 58, 80, 81], we can obtain better guarantees, but these improvements are hidden in the polylogarithmic factors in the exponential. We discuss learning given data from a LDS in Appendix C. Intuitively, a LDS is a collection of points in the parameter space that covers the space such that there are no large gaps, or discrepancies.

Similar to Corollary 1, if we are instead given training data $\{x_\ell, \sigma_T(\rho(x_\ell))\}_{\ell=1}^N$, where $\sigma_T(\rho(x_\ell))$ is a classical shadow representation [52, 68, 69, 70, 71] of the ground state $\rho(x_\ell)$, then we obtain the following immediate corollary of Theorem 5.

**Corollary 2** (Learning representations of ground states with neural networks). *Let $n, \delta > 0$, $1/e > \epsilon > 0$ and $\delta > 0$. Given training data $\{(x_\ell, \sigma_T(\rho(x_\ell))\}_{\ell=1}^N$ of size*

$$N = \mathcal{O}\left(\log(1/\delta)2^{\text{polylog}(1/\epsilon)}\right), \tag{3.10}$$

*where $x_\ell$ is sampled from a distribution $\mathcal{D}$ satisfying the same assumptions as Theorem 5 and $\sigma_T(\rho(x_\ell))$ is the classical shadow representation of the ground state $\rho(x_\ell)$ using $T$ randomized Pauli measurements. For $T = \tilde{\mathcal{O}}(\log(n/\delta)/\epsilon^2)$, with probability at least $1 - \delta$, the ML algorithm will produce a ground state representation $\hat{\rho}_{N,T}(x)$ that achieves*

$$\mathop{\mathbb{E}}_{x \sim \mathcal{D}} |\text{tr}(O\hat{\rho}_{N,T}(x)) - \text{tr}(O\rho(x))|^2 \leq \epsilon \tag{3.11}$$

*for any observable with $\|O\|_\infty \leq 1$ that can be written as a sum of geometrically local observables.*

### 3.2.1 Proof ideas for neural network guarantee

To prove Theorem 5, we first show that our neural network model $f^{\Theta,w}$ can approximate the ground state properties well. In particular, we show that there exist weights $\Theta', w'$ such that $f^{\Theta',w'}$ approximates the ground state properties and thus achieves small value for the training objective (Equation (3.7)). Then, we bound the prediction error using tools from deep learning and quasi-Monte Carlo theory [54, 55, 56, 57, 58]. We ensure the existence of $f^{\Theta',w'}$ in the following theorem.

**Theorem 6.** *For any $1/e > \epsilon > 0$ and width $W$, there exist weights $\Theta', w'$ such that the neural network model $f^{\Theta',w'}$ satisfies*

$$|f^{\Theta',w'}(x) - \text{tr}(O\rho(x))|^2 \leq \epsilon, \quad \forall x \in [-1,1]^m. \tag{3.12}$$

*Moreover, each parameter $\Theta_i$ of the network has a magnitude of $|\Theta_i| = 2^{\mathcal{O}(\text{polylog}(1/\epsilon))}$.*

This implies that for a suitable choice of regularization parameter $\lambda = \mathcal{O}(\epsilon)$, the training objective from Equation (3.7) is also small. We prove this statement by combining results in deep learning regarding tanh neural networks approximating functions [53] with the geometric locality of the system and smoothness of the ground state properties. We note that the weights $\Theta', w'$ in Theorem 6 are not necessarily the weights $\Theta^*, w^*$ found via the neural network training procedure in Theorem 5. Because the training objective is non-convex, we cannot guarantee convergence to these weights $\Theta', w'$. However, assuming that $f^{\Theta^*,w^*}$ does indeed achieve a low training error (which is often satisfied in practice), we are able to rigorously guarantee that the model will generalize well and achieve a low prediction error in Theorem 5.

Notice that the guarantee of Theorem 6 holds for all $x$ and, in particular, does not require our assumptions on the distribution $\mathcal{D}$. The assumption that the network is trained on such data only becomes relevant when bounding the prediction error. While not explicitly stated here, we also note that Theorem 6 gives a bound on the number of trainable parameters $|\Theta_i|$ that has a similar dependence on $\epsilon$ as the model in [2]. Furthermore, the parameters are independent of system size, $n$. Additional smoothness assumptions on the Hamiltonian $H(x)$ can yield mild improvements on the dependence in terms of $\epsilon$, as briefly discussed in Appendix C.1. Moreover, because of this bound on $|\Theta_i|$, applying an additional penalty on the $\ell_2$-norm of the weights $\Theta$ can help ensure that the weights remain small. In practice, this is usually satisfied during training when the weights are initialized properly and the inputs are regularized, e.g. [53]. Thus, the condition that $|\Theta_i^*| \le W_{\max}$ is often satisfied in practice and is not considered a strong assumption in deep learning.

To prove the prediction error bound in Theorem 5 assuming that a low training error is achieved, we combine techniques from quasi-Monte Carlo theory applied to deep learning [54] (see Appendix A.2 for a review) along with our knowledge of the geometry of the $n$-qubit system. In contrast to [54], we need to characterize the dimension of the input domain in our approach. The reason for doing this is that the approximation error depends on the size $\tilde{m} = |I_P|$ (Equation (2.3)) of our local models $f_P^{\theta_P}$.

The central result we use here is the Koksma-Hlawka inequality [56] (see Theorem 10 in Appendix A.2) from quasi-Monte Carlo theory. This produces a bound on the prediction error in terms of the star-discrepancy (see Definition 2 in Appendix A.2) and the Hardy-Krause variation. The star-discrepancy can be controlled by known bounds on the star-discrepancy of random points [82]. We bound the Hardy-Krause variation by explicitly computing the mixed derivatives of the local models $f_P^{\theta_P}$ and the ground state properties $\mathrm{tr}(O\rho(x))$. In particular, we bound the latter using tools from the spectral flow formalism [59, 60, 61], and this is where the assumption that the mixed first order derivatives of the Hamiltonian are bounded is needed. Putting these steps together, we arrive at the rigorous guarantee in Theorem 5.

## 4 Numerical experiments

We conduct numerical experiments to observe the performance of our model in practice. The results demonstrate that our assumptions in Theorem 5 are often satisfied in practice and that our deep learning algorithm outperforms the previous best-known method [2]. Moreover, we generate and utilize significantly more training data than in prior works [1, 2]. The code can be found at `https://github.com/marcwannerchalmers/learning_ground_states.git`.

We consider the classical neural network model discussed in the previous section and defined formally in Definition 6. For each of the local models $f_P^{\theta_P}$, we use fully connected deep neural networks with five hidden layers of width 200. We train the model with the AdamW optimization algorithm [83]. We measure the training error and prediction error via the root-mean-square error (RMSE). The model is discussed further in Appendix D.

As in [2], we consider the two-dimensional antiferromagnetic random Heisenberg model on between 20 to 45 qubits and predict two-body correlation functions. The corresponding Hamiltonian is

$$H = \sum_{\langle ij \rangle} J_{ij}(X_iX_j + Y_iY_j + Z_iZ_j), \tag{4.1}$$

where $\langle ij \rangle$ denotes all pairs of neighboring sites on the lattice. The coupling terms $J_{ij}$ correspond to the parameters $x$ of the Hamiltonian and are sampled uniformly from $[0, 2]$.

We generate training data similarly to [1, 2], using the density-matrix renormalization group (DMRG) [9] based on matrix-product-states (MPS) [84]. To assess the performance of our model, we consider both uniformly randomly distributed $J_{ij}$ and coupling parameters, which are distributed as a Sobol sequence. It is easy to see that the distributions and $H$ satisfy the requirements of Theorem 5.

In Figure 2 (Left), we see that our deep learning algorithm consistently outperforms the previous best-known ML algorithm from [2], achieving approximately half the prediction error on the same training data. The prediction error also exhibits a constant scaling with respect to system size, agreeing with our rigorous guarantee in Theorem 5. Another noteworthy observation is that the ML algorithm's performance on LDS is nearly equivalent to its performance on uniformly random points. We discuss a potential reason for this in Appendix D.

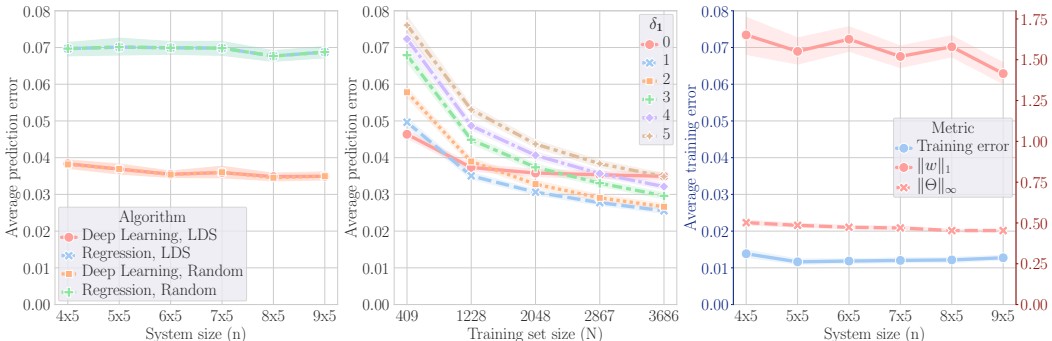

Figure 2: **Numerical experiments. (Left)** Comparison with previous methods. Each point indicates the prediction error (RMSE) of our deep learning model or the regression model of [2], fixing the training set size $N = 3686$ and the size of the local neighborbood $\delta_1 = 0$ (Equation (2.3)). We train both algorithms on either LDS or uniformly random points. **(Center)** Scaling with training size. Each point indicates the prediction error of our deep learning model given LDS training data for various $\delta_1$ and training data sizes. **(Right)** Neural network weights and training error. Blue points correspond to the training error of the neural network model. Red points correspond to the $\ell_1$ norm of parameters in the last layer or the largest absolute value of the parameters of the neural network, fixing $N = 3686$ and $\delta_1 = 1$. This shows that the assumptions in Theorem 5 are achieved in practice. The shaded areas denote the 1-sigma error bars across the assessed ground state properties.

Figure 2 (Center) illustrates the prediction error scaling with respect to the training set size for various choices of $\delta_1$ (size of the local neighborhood from Equation (2.3)). For $\delta_1 = 0$, the error arising from approximating the ground state property via local functions dominates. For $\delta_1 > 0$, we observe a smaller local approximation error and thus achieve a smaller prediction error for sufficiently large training sets. This is consistent with our theoretical results.

Finally, Figure 2 (Right) illustrates that our assumptions in Theorem 5 are mild in practice. Namely, the blue points show that a small training error can be achieved. The red points also demonstrate that the $\ell_1$-norm of the parameters in the last layer and the largest absolute value of the parmeters in the trained neural network remain small. In particular, in Figure 2, the weights exhibit a scaling *independent* of system size $n$. Hence, we find that the assumptions needed to guarantee the prediction error bound in Theorem 5, namely that the training objective is small and the weights of the neural network are small and independent of system size, are fulfilled in our numerical experiments. We provide further details of the numerical experiments in Appendix D.

## 5   Discussion

We have shown that we can construct ML models for predicting ground state properties that require only a constant number of training samples, for a fixed prediction error. Specifically, we showed that a simple modification to the linear regression model in [2] only requires $2^{\mathsf{polylog}(\epsilon^{-1})}$ samples in order to achieve a prediction error of $\epsilon$, provided that we know a decomposition of the observable of interest in terms of Pauli operators. We then showed that a neural network model which is trained on $2^{\mathsf{polylog}(\epsilon^{-1})}$ training samples and which achieves $\mathcal{O}(\epsilon)$ training error on these samples will also have a prediction error of at most $\epsilon$. In this case, knowledge of the observable $O$ is no longer required.

Our work leaves open several avenues for future exploration. First, it would be desirable to understand the conditions under which we can prove convergence for the training error. For instance, could the model be changed so as to use a convex objective, thereby avoiding the issues associated with finding a global optimum in a non-convex landscape? Following [49, 50], we would also like to know whether the results obtained for neural networks can be extended to thermal states or Lindbladian phases of matter. Finally, for both results it would be desirable to improve the scaling with respect to the error $\epsilon$. Currently, the models have quasipolynomial scaling in $1/\epsilon$ and the only case in which we know how to achieve $\mathsf{poly}(1/\epsilon)$ scaling is when the number of parameters, $m$, is constant (as in [51]).

## Acknowledgements

MW and DD are supported by SSF (Swedish Foundation for Strategic Research), grant number FUS21-0063. LL is supported by a Marshall Scholarship. CB is partially supported by a grant from DIA-COE. AG is supported by the Knut and Alice Wallenberg Foundation through the Wallenberg Centre for Quantum Technology (WACQT). This work was done in part while a subset of the authors were visiting the Simons Institute for the Theory of Computing.

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

# Appendices

## Contents

These appendices provide the technical details of the ideas discussed in the main text. In Appendix A, we review several important concepts for our proofs such as the algorithm and rigorous guarantee from [2] in Appendix A.1 and background on classical deep learning techniques in Appendix A.2. In Appendix B, we build on [2] to obtain a sample complexity upper bound for predicting ground state properties independent of system size. In Appendix C, we prove our guarantee for predicting ground state properties using neural networks.

## A Preliminaries

### A.1 Review of previous algorithm and proof

In this section, we review the previous algorithm from [2] along with intermediate results we use throughout our proofs. For full details, we refer the reader to [2]. Throughout this section, let $1/e > \epsilon_1, \epsilon_2, \epsilon_3 > 0$. One can think of $\epsilon_1$ as the approximation error caused by the hypothesis of our ML algorithm not exactly capturing the ground state property; $\epsilon_2$ represents the noise in the training data; $\epsilon_3$ corresponds to the generalization error.

Recall that we consider a family of $n$-qubit Hamiltonians $H(x)$ smoothly parameterized by an $m$-dimensional vector $x \in [-1, 1]^m$ that satisfies the following assumptions, which we restate from [2].

(a) *Physical system:* We consider $n$ finite-dimensional quantum systems that are arranged at locations, or sites, in a $d$-dimensional space, e.g., a spin chain ($d = 1$), a square lattice ($d = 2$), or a cubic lattice ($d = 3$). Unless specified otherwise, our big-$\mathcal{O}, \Omega, \Theta$ notation is with respect to the thermodynamic limit $n \to \infty$.

(b) *Hamiltonian:* $H(x)$ decomposes into a sum of geometrically local terms $H(x) = \sum_{j=1}^{L} h_j(\vec{x}_j)$, each of which only acts on an $\mathcal{O}(1)$ number of sites in a ball of $\mathcal{O}(1)$ radius. Here, $\vec{x}_j \in \mathbb{R}^q, q = \mathcal{O}(1)$ and $x$ is the concatenation of $L$ vectors $\vec{x}_1, \dots, \vec{x}_L$ with dimension $m = Lq = \mathcal{O}(n)$. Individual terms $h_j(\vec{x}_j)$ obey $\|h_j(\vec{x}_j)\|_\infty \leq 1$ and also have bounded directional derivative: $\|\partial h_j / \partial \hat{u}\|_\infty \leq 1$, where $\hat{u}$ is a unit vector in parameter space.

(c) *Ground-state subspace:* We consider the ground state $\rho(x)$ for the Hamiltonian $H(x)$ to be defined as $\rho(x) = \lim_{\beta \to \infty} e^{-\beta H(x)} / \operatorname{tr}\left(e^{-\beta H(x)}\right)$. This is equivalent to a uniform mixture over the eigenspace of $H(x)$ with the minimum eigenvalue.

(d) *Observable:* $O$ can be written as a sum of few-body observables $O = \sum_j O_j$, where each $O_j$ only acts on an $\mathcal{O}(1)$ number of sites. Hence, we can also write $O = \sum_{P \in S^{(\mathrm{geo})}} \alpha_P P$, where $P \in \{I, X, Y, Z\}^{\otimes n}$ and $S^{(\mathrm{geo})}$ is the set of all geometrically local Pauli observables. We focus on $O$ given as a sum of geometrically local observables $\sum_j O_j$, where each $O_j$ only acts on an $\mathcal{O}(1)$ number of sites in a ball of $\mathcal{O}(1)$ radius. Moreover, $O$ has $\|O\|_\infty = \mathcal{O}(1)$.

We also assume that the spectral gap of $H(x)$ is bounded from below by some constant $\gamma$ for all choices of parameters $x \in [-1, 1]^m$.

The ML algorithm is given a training data set $\{(x_\ell, y_\ell)\}_{\ell=1}^N$, where $x_\ell$ is sampled from some distribution $\mathcal{D}$ over the parameter space $[-1, 1]^m$ and $y_\ell$ approximates the ground state property: $|y_\ell - \operatorname{tr}(O\rho(x_\ell))| \leq \epsilon_2$. The goal is to learn some function $h^*(x)$ that achieves a low average prediction error

$$\mathbb{E}_{x \sim \mathcal{D}} |h^*(x) - \operatorname{tr}(O\rho(x))|^2 \leq \epsilon. \tag{A.1}$$

### A.1.1 ML Algorithm

The ML algorithm proposed in [2] requires several geometric definitions. We use $S^{(\mathrm{geo})}$ to denote the set of all geometrically local Pauli observables throughout.

Let $\delta_1, \delta_2, B > 0$ be efficiently-computable hyperparameters that we define later. Then, define the set $I_P$ of coordinates $c$ that parameterize some local term $h_{j(c)}$ that is close to a Pauli $P \in \{I, X, Y, Z\}^{\otimes n}$. Here, the distance between two observables $d_{\mathrm{obs}}$ is defined as the minimum distance between the qubits that the observables act on, where the distance between qubits is given by the geometry of the system, which we assume to be known. Formally, we define the set of local coordinates as

$$I_P \triangleq \{c \in \{1, \dots, m\} : d_{\mathrm{obs}}(h_{j(c)}, P) \leq \delta_1\}, \tag{A.2}$$

where $h_{j(c)}$ is the local term in the Hamiltonian $H(x)$ whose parameters $\vec{x}_{j(c)}$ include the variable $x_c$. The intuition behind this set of coordinates is that it indexes the parameters $x_c$ that influence the ground state property $\operatorname{tr}(P\rho(x))$ corresponding to this Pauli $P$. Using this intuition, because these parameters $x_c$ for $c \in I_P$ matter most for the property we are trying to learn (as [2] proves and we give the ideas for later), then we can define a new effective parameter space in which all other parameters are set to zero. Moreover, parameters $x_c$ for $c \in I_P$ can be discretized to lie on a lattice. This gives the following set $X_P$

$$X_P \triangleq \left\{ x \in [-1, 1]^m : \begin{array}{l} \text{if } c \notin I_P, x_c = 0 \\ \phantom{\text{if }} \text{if } c \in I_P, x_c \in \{0, \pm\delta_2, \pm 2\delta_2, \dots, \pm 1\} \end{array} \right\}. \tag{A.3}$$

We can also define a set $T_{x,P}$ for each vector $x \in X_P$ which is the set of parameters $x'$ that are close to $x$ for coordinates in $I_P$:

$$T_{x,P} \triangleq \left\{ x' \in [-1, 1]^m : -\frac{\delta_2}{2} < x_c - x'_c \leq \frac{\delta_2}{2}, \ \forall c \in I_P \right\}. \tag{A.4}$$

With these definitions in place, we set the hyperparameters as follows. Define $\delta_1$ as

$$\delta_1 \triangleq \max\left( C_{\max} \log^2(2C/\epsilon_1), C_4, C_5, \frac{\max(5900, \alpha, 7(d + 11), \theta)}{b} \right), \tag{A.5}$$

where $b, C_{\max}, C_4, C_5, \alpha, \theta, C$ are all constants. We refer to the supplementary information of [2] for a full description of these constants. Moreover, $\delta_2$ is given by

$$\delta_2 \triangleq \frac{1}{\left\lceil \frac{2\sqrt{C'|I_P|}}{\epsilon_1} \right\rceil}, \tag{A.6}$$

where $C'$ is a constant. Finally, we define an additional hyperparameter $B > 0$ as

$$B \triangleq 2^{\mathcal{O}(\mathrm{polylog}(1/\epsilon_1))}. \tag{A.7}$$

The ML algorithm from [2] utilizes these objects to encode the geometric locality of the system. The algorithm consists of two steps. First, it maps the parameter space $[-1, 1]^m$ to a high dimensional space $\mathbb{R}^{m_\phi}$ for

$$m_\phi \triangleq \sum_{P \in S^{(\mathrm{geo})}} |X_P| = \mathcal{O}(n) 2^{\mathcal{O}(\mathrm{polylog}(1/\epsilon_1))} \tag{A.8}$$

via a nonlinear feature map $\phi$. Second, it runs $\ell_1$-regularized linear regression (LASSO) over the feature space.

This first step encodes the geometry of the problem. In particular, the feature map is defined as follows, where each coordinate of $\phi(x)$ is indexed by $x' \in X_P$ and $P \in S^{(\mathrm{geo})}$

$$\phi(x)_{x',P} \triangleq \mathbb{1}[x \in T_{x',P}]. \tag{A.9}$$

In this way, the feature map $\phi(x)$ identifies the nearest lattice point to $x$. The idea is that one can approximate the ground state property well by only approximating it at these representative points and summing up. We make this intuition rigorous in the following section.

Following the feature mapping, our ML algorithm uses LASSO [62, 63, 64] to learn functions of the form $\{h(x) = \mathbf{w} \cdot \phi(x) : \|w\|_1 \leq B\}$. In particular, for a chosen hyperparameter $B > 0$, LASSO finds a coefficient vector $\mathbf{w}^*$ that solves the following optimization problem minimizing the training error subject to the constraint that $\|w\|_1 \leq B$

$$\min_{\substack{\mathbf{w} \in \mathbb{R}^{m_\phi} \\ \|w\|_1 \leq B}} \frac{1}{N} \sum_{\ell=1}^{N} |\mathbf{w} \cdot \phi(x_\ell) - y_\ell|^2. \tag{A.10}$$

We denote the learned function by $h^*(x) = \mathbf{w}^* \cdot \phi(x)$. Note that the learned function does not need to achieve the minimum training error, but can be some amount $\epsilon_3/2$ above it. For our purposes, we set $B = 2^{\mathcal{O}(\mathrm{polylog}(1/\epsilon_1))}$.

This algorithm obtains the following rigorous guarantee.

**Theorem 7** (Theorem 5 in [2]). *Let $1/e > \epsilon_1, \epsilon_2, \epsilon_3 > 0$ and $\delta > 0$. Given training data $\{(x_\ell, y_\ell)\}_{\ell=1}^{N}$ of size*

$$N = \log(n/\delta) 2^{\mathcal{O}(\log(1/\epsilon_3) + \mathrm{polylog}(1/\epsilon_1))}, \tag{A.11}$$

*where $x_\ell$ is sampled from $\mathcal{D}$ and $y_\ell$ is an estimator of $\mathrm{tr}(O\rho(x_\ell))$ such that $|y_\ell - \mathrm{tr}(O\rho(x_\ell))| \leq \epsilon_2$, the ML algorithm can produce $h^*(x)$ that achieves prediction error*

$$\mathbb{E}_{x \sim \mathcal{D}} |h^*(x) - \mathrm{tr}(O\rho(x))|^2 \leq (\epsilon_1 + \epsilon_2)^2 + \epsilon_3 \tag{A.12}$$

*with probability at least $1 - \delta$. The training time for constructing the hypothesis function $h$ and the prediction time for computing $h^*(x)$ are upper bounded by $\mathcal{O}(nN) = n \log(n/\delta) 2^{\mathcal{O}(\log(1/\epsilon_3) + \mathrm{polylog}(1/\epsilon_1))}$.*

### A.1.2 Proof Ideas

This rigorous guarantee is proven by first showing that the training error

$$\hat{R}(h) = \min_{\mathbf{w}} \frac{1}{N} \sum_{\ell=1}^{N} |h(x_\ell) - y_\ell|^2 \tag{A.13}$$

is small.

**Lemma 1** (Lemma 15 in [2]). *The function*

$$g(x) = \sum_{P \in S^{(\text{geo})}} \sum_{x' \in X_P} f_P(x') \mathbb{1}[x \in T_{x',P}] = \mathbf{w}' \cdot \phi(x), \tag{A.14}$$

*achieves training error*

$$\hat{R}(g) \leq (\epsilon_1 + \epsilon_2)^2. \tag{A.15}$$

The proof of this consists of three different steps. First, one can show that $\text{tr}(O\rho(x))$ can be approximated by a sum of smooth local functions, denoted as $f(x) = \sum_{P \in S^{(\text{geo})}} f_P(x)$, where $f_P(x) = \alpha_P \text{tr}(P\rho(\chi_P(x)))$ for $O = \sum_{P \in \{I,X,Y,Z\}^{\otimes n}} \alpha_P P$ and

$$\chi_P(x)_c = \begin{cases} x_c, & c \in I_P \\ 0 & c \notin I_P \end{cases} \tag{A.16}$$

for all $c \in \{1, \ldots, m\}$. In other words, parameters that parameterize local terms $h_j$ far away a Pauli $P$ ($x_c$ for $c \notin I_P$) do not contribute much to the ground state property, and thus we can simply set them to zero. Formally, this approximation is given in the following lemma.

**Lemma 2** (Corollary 2 in [2]). *Consider a class of local Hamiltonians $\{H(x) : x \in [-1,1]^m\}$ and an observable $O = \sum_{P \in \{I,X,Y,Z\}^{\otimes n}} \alpha_P P$ satisfying assumptions (a)-(d). There exists a constant $C > 0$ such that for any $1/e > \epsilon_1 > 0$,*

$$|\text{tr}(O\rho(x)) - f(x)| \leq C\epsilon_1 \left( \sum_{P \in S^{(\text{geo})}} |\alpha_P| \right), \tag{A.17}$$

*where $f(x) = \sum_{P \in S^{(\text{geo})}} f_P(x)$.*

Second, one can also show that this sum of local functions $f(x) = \sum_{P \in S^{(\text{geo})}} f_P(x)$ can in turn be approximated by a linear function over the feature space $g(x) = \mathbf{w}' \cdot \phi(x)$, where $\mathbf{w}'$ is a vector with entries indexed by $P \in S^{(\text{geo})}$ and $x' \in X_P$ given by $\mathbf{w}'_{x',P} = f_P(x')$.

**Lemma 3** (Corollary 3 in [2]). *For $g(x) = \mathbf{w}' \cdot \phi(x)$ and $f(x) = \sum_{P \in S^{(\text{geo})}} f_P(x)$, then writing an observable $O = \sum_{P \in \{I,X,Y,Z\}^{\otimes n}} \alpha_P P$, we have*

$$|g(x) - f(x)| < \epsilon_1 \left( \sum_{P \in S^{(\text{geo})}} |\alpha_P| \right) \tag{A.18}$$

*for any $x$.*

This tells us that the hypothesis functions of the ML algorithm indeed approximate the ground state properties well. The final piece needed is a norm inequality bounding the $\ell_1$-norm of the Pauli coefficients. This allows us to bound the terms involving $|\alpha_P|$ in Lemma 2 and Lemma 3. In particular, we have the following bound.

**Theorem 8** (Corollary 4 in [2]). *Let $O = \sum_{P \in \{I,X,Y,Z\}^{\otimes n}} \alpha_P P$ be an observable that can be written as a sum of geometrically local observables. Then,*

$$\sum_P |\alpha_P| = \mathcal{O}(1). \tag{A.19}$$

Given these results, Lemma 1 follows directly by triangle inequality and rescaling $\epsilon_1$ when using Lemma 2 and Lemma 3. Finally, to prove Theorem 7, it remains to bound the generalization error by $\epsilon_3$. This follows directly from known sample complexity guarantees for the LASSO algorithm [64], which learns $\ell_1$-regularized linear functions. In order to apply this known result, one needs to provide a regularization parameter, i.e., some $B > 0$ such that the ML algorithm learns functions of the form $h(x) = \mathbf{w} \cdot \phi(x)$ for $\|\mathbf{w}\|_1 \leq B$. To choose such a $B$, Lewis et al. bound the $\ell_1$-norm of $\mathbf{w}'$, where recall $\mathbf{w}'_{x',P} = f_P(x')$.

**Lemma 4** (Lemma 14 in [2]). *Let $\mathbf{w}'$ be the vector of coefficients defined by $\mathbf{w}'_{x',P} = f_P(x')$. Then,*

$$\|\mathbf{w}'\|_1 = \sum_{P \in S^{(\text{geo})}} \sum_{x' \in X_P} |f_P(x')| = 2^{\mathcal{O}(\text{polylog}(1/\epsilon_1))}. \tag{A.20}$$

Using $B = 2^{\mathcal{O}(\text{polylog}(1/\epsilon_1))}$ in the known guarantees for LASSO [64] gives Theorem 7.

## A.2 Deep learning with low-discrepancy sequences

In this section, we review results in classical deep learning theory for obtaining rigorous guarantees when learning from data sampled according to a low-discrepancy sequence (LDS) [55, 56, 76, 77, 78, 57, 79, 58, 80, 81]. For this discussion, we follow [54, 85].

We consider a *neural network* or *multi-layer perceptron model* as a composition of several layers of affine transformations and nonlinear activation functions. Namely, let $\sigma : \mathbb{R} \to \mathbb{R}$ be a nonlinear activation function. Then, a neural network with $L$ layers is defined as follows. Let $d_0, \ldots, d_L \in \mathbb{N}$ be the dimension (number of neurons or *width*) of each layer $k \in \{0, \ldots, L\}$. Here, the zeroth layer is the *input layer* and the $L$th layer is the *output layer*. At each layer $k \in \{0, \ldots, L-1\}$ except for the output layer, we define an affine transformation $W_k : \mathbb{R}^{d_k} \to \mathbb{R}^{d_{k+1}}$ by $W_k(x) = A_k x + b_k$ for a matrix of *weights* $A_k \in \mathbb{R}^{d_{k+1} \times d_k}$ and a vector of *biases* $b_k \in \mathbb{R}^{d_{k+1}}$. Then, a neural network is defined as

$$f_\theta(x) = (W_{L-1} \circ \sigma \circ \cdots \circ \sigma \circ W_0)(x), \tag{A.21}$$

where $\sigma$ is applied element-wise and $\theta = ((A_0, b_0), \ldots, (A_{L-1}, b_{L-1}))$. The *hidden layers* are the first $L-1$ layers. Here, $\theta$ are the trainable parameters of the neural network, which can be iteratively updated through training on data. A *deep neural network* is a neural network with at least three layers: $L \geq 3$. In this work, we consider the activation function

$$\sigma(x) = \tanh(x) = \frac{e^x - e^{-x}}{e^x + e^{-x}}. \tag{A.22}$$

We refer to such neural networks with this activation function as *tanh neural networks*.

Suppose a neural network $f_\theta$ aims to approximate some target function $f$, given training data $\{(x_\ell, f(x_\ell)\}_{\ell=1}^N$. Then, the training error is defined as

$$\hat{R}(\theta) = \frac{1}{N} \sum_{\ell=1}^N |f(x_\ell) - f_\theta(x_\ell)|^2. \tag{A.23}$$

The prediction error is then defined over the whole domain, including unseen data, as

$$R(\theta) = \mathbb{E}_{x \sim \mathcal{D}} |f(x) - f_\theta(x)|^2, \tag{A.24}$$

where the training data is sampled from some distribution $\mathcal{D}$. A canonical result in deep learning theory [66] is that the generalization error can be bounded by roughly

$$R(\theta) \lesssim \hat{R}(\theta) + \mathcal{O}\left(\frac{1}{\sqrt{N}}\right), \tag{A.25}$$

where $\lesssim$ indicates that we only state this result schematically. Importantly, this means that in order for the neural network $f_\theta$ to approximate $f$ with high accuracy, many training data points $N$ are needed, which is undesirable. In order to fix this issue, [54] combines ideas from deep learning with tools from quasi-Monte Carlo methods [55, 56, 57, 58] to achieve a generalization error bound of

$$R(\theta) \lesssim \hat{R}(\theta) + \tilde{\mathcal{O}}\left(\frac{1}{N}\right), \tag{A.26}$$

where $\tilde{\mathcal{O}}$ indicates that we are suppressing polylogarithmic factors. The key tool used here is low-discrepancy sequences [55, 56, 76, 77, 78, 57, 79, 58, 80, 81]. Intuitively, this is a collection of points that covers domain of the function $f$ in such a way that there are no large gaps, or discrepancies. By filling these gaps, one can ensure that the training data accurately represents the target function, more so than even uniformly random data. We leverage these ideas to obtain our rigorous guarantee on the sample complexity of a deep learning algorithm for predicting ground state properties. In the following, we formally define low-discrepancy sequences and a key inequality in quasi-Monte Carlo theory for obtaining our generalization bound.

First, we define the discrepancy of a sequence, which is a measure of uniformity.

**Definition 1** (Discrepancy [56]). *Let $\lambda$ be the Lebesgue measure, $N \in \mathbb{N}$. Let $x = \{x_\ell\}_{\ell=1}^N$ be a sequence of points with $x_\ell \in [0, 1]^d$ for all $\ell$. The* discrepancy *of the sequence $x$ is defined as*

$$D_N(d) = \sup_{J \in E} |R_N(J)|, \tag{A.27}$$

*where*

$$R_N(J) = \frac{1}{N} \sum_{\ell=1}^{N} \mathbb{1}\{x_\ell \in J\} - \lambda(J) \tag{A.28}$$

*for a Lebesgue-measurable set $J \subseteq [0,1]^d$. Also, E is the set of all rectangular subsets of $[0,1]^d$, i.e.,*

$$E = \left\{ \prod_{i=1}^{d} [a_i, b_i) : 0 \le a_i < b_i \le 1 \right\}. \tag{A.29}$$

Intuitively, one can consider the discrepancy as a measure of how well the sequence fills rectangular subsets of $[0,1]^d$. If the discrepancy is small, this means that the sequence fills these subsets well. We can similarly define the star-discrepancy, where the supremum is instead taken over rectangular subsets of $[0,1]^d$ such that one endpoint is 0.

**Definition 2** (Star-discrepancy [56])**.** *Let $\lambda$ be the Lebesgue measure, $N \in \mathbb{N}$. Let $x = \{x_\ell\}_{\ell=1}^{N}$ be a sequence of points with $x_\ell \in [0,1]^d$ for all $\ell$. The* star-discrepancy *of the sequence x is defined as*

$$D_N^*(d) = \sup_{J \in E^*} |R_N(J)|, \tag{A.30}$$

*where $R_N$ is defined in Equation (A.28) for a Lebesgue-measurable set $J \subseteq [0,1]^d$. Also, $E^*$ is the set of all rectangular subsets of $[0,1]^d$, i.e.,*

$$E^* = \left\{ \prod_{i=1}^{d} [0, b_i) : 0 < b_i \le 1 \right\}. \tag{A.31}$$

With these definitions, we can define low-discrepancy sequences.

**Definition 3** (Low-discrepancy sequence [56])**.** *A sequence of points $x = \{x_\ell\}_{\ell=1}^{N}$ with $x_\ell \in [0,1]^d$ for all $\ell$ is a* low-discrepancy sequence *if*

$$D_N^*(d) \le C \frac{(\log N)^d}{N}, \tag{A.32}$$

*where C is a constant that possibly depends on d but is independent of N.*

The value of the constant $C$ in this definition depends on the construction of the low-discrepancy sequence. Several constructions of low-discrepancy sequences are known [80, 77, 81, 57]. In this work, we consider Sobol sequences in base 2 [57]. For these sequences, we have the following guarantee

**Theorem 9** (Theorem 4.17 in [57])**.** *Let $N \in \mathbb{N}$. If $x = \{x_\ell\}_{\ell=1}^{N}$ is a Sobol sequence in base 2 with $x_\ell \in [0,1]^d$ for all $\ell$, then the star-discrepancy satisfies*

$$D_N^*(d) \le C(d) \frac{(\log N)^d}{N}, \tag{A.33}$$

*where $C(d)$ is a constant satisfying*

$$C(d) < \frac{1}{d!} \left( \frac{d}{\log(2d)} \right). \tag{A.34}$$

We state this result without proof and refer to [57] for details on this construction. Another important known discrepancy bound that we will use in Appendix C.3 is the following bound on the star-discrepancy of uniformly random points.

**Lemma 5** (Corollary 1 in [82])**.** *For any $d \ge 1$, $N \ge 1$ and $\delta \in (0,1)$ a (uniformly) randomly generated d-dimensional point set $(x_1, \ldots, x_N)$ satisfies*

$$D_N^*(d) \le 5.7 \sqrt{4.9 + \log\left(\frac{1}{\delta}\right)} \frac{\sqrt{d}}{\sqrt{N}} \tag{A.35}$$

*with probability at least $1 - \delta$.*

As discussed earlier, low-discrepancy sequences allow us to obtain better sample complexity guarantees for neural networks. The key result in quasi-Monte Carlo theory that enables this is the Koksma-Hlawka inequality [56]. In order to properly state it, we first need to define the Hardy-Krause variation. A full technical definition can be found in, e.g., [54], but for our purposes, it suffices to consider the following upper bound [86]. Let $f$ be a "sufficiently smooth" function. Then, its Hardy-Krause variation can be upper bounded by

$$V_{HK}(f) \leq \hat{V}_{HK} = \int_{[0,1]^d} \left| \frac{\partial^d f(y)}{\partial y_i \cdots \partial y_d} \right| dy + \sum_{i=1}^{d} \hat{V}_{HK}(f_1^{(i)}), \tag{A.36}$$

where $f_1^{(i)}$ is the restriction of the function $f$ to the boundary $y_i = 1$. If all of the mixed partial derivatives are continuous, then this inequality is actually an equality [56]. Now, we can state the Koksma-Hlawka inequality.

**Theorem 10** (Koksma-Hlawka inequality). *Let $f : [0,1]^d \to \mathbb{R}$ be a function whose mixed derivatives are absolutely integrable over its domain with bounded Hardy-Krause variation $V_{HK}(f) < \infty$. Let $x = \{x_\ell\}_{\ell=1}^N$ be a sequence of $N$ $d$-dimensional points in $[0,1]^d$ with star-discrepancy $D_N^*(d)$. Then*

$$\left| \int_{[0,1]^d} f(x) \, dx - \frac{1}{N} \sum_{\ell=1}^{N} f(x_\ell) \right| \leq V_{HK}(f) D_N^*(d). \tag{A.37}$$

This theorem is used in quasi-Monte Carlo methods to estimate the error of approximating an integral of a function $f$ by the empirical average of $f$ evaluated on a sequence of points. Notice that if the sequence $x$ is a low-discrepancy sequence, then by definition, we can upper bound the star-discrepancy $D_N^*$. Moreover, recalling the definitions of prediction error and training error (Equations (A.23) and (A.24)), one can see how this relates to our task of bounding the prediction error.

To generalize our results to a wider class of distributions, we need to extend these tools for arbitrary measures, rather than just the Lebesgue measure. First, we restate the definition of discrepancy and star-discrepancy [87].

**Definition 4** (General Discrepancy [88]). *Let $\mu$ be a normalized Borel measure on $[0,1]^d$. Let $x = \{x_\ell\}_{\ell=1}^N$ be a sequence of points with $x_\ell \in [0,1]^d$ for all $\ell$. The discrepancy with respect to $\mu$ of the sequence $x$ is defined as*

$$D_N(d; \mu) = \sup_{J \in E} |R_N(J; \mu)|, \tag{A.38}$$

*where*

$$R_N(J; \mu) = \frac{1}{N} \sum_{\ell=1}^{N} \mathbb{1}\{x_\ell \in J\} - \mu(J) \tag{A.39}$$

*for a Borel-measurable set $J \subseteq [0,1]^d$. Also, $E$ is the set of all rectangular subsets of $[0,1]^d$, i.e.,*

$$E = \left\{ \prod_{i=1}^{d} [a_i, b_i) : 0 \leq a_i < b_i \leq 1 \right\}. \tag{A.40}$$

**Definition 5** (General Star-Discrepancy [87]). *Let $\mu$ be a normalized Borel measure on $[0,1]^d$, and let $N \in \mathbb{N}$. Let $x = \{x_\ell\}_{\ell=1}^N$ be a sequence of points with $x_\ell \in [0,1]^d$ for all $\ell$. The star-discrepancy with respect to $\mu$ of the sequence $x$ is defined as*

$$D_N^*(d; \mu) = \sup_{J \in E^*} |R_N(J; \mu)|, \tag{A.41}$$

*where $R_N$ is defined in Equation (A.39) for a Borel-measurable set $J \subseteq [0,1]^d$. Also, $E^*$ is the set of all rectangular subsets of $[0,1]^d$, i.e.,*

$$E^* = \left\{ \prod_{i=1}^{d} [0, b_i) : 0 < b_i \leq 1 \right\}. \tag{A.42}$$

These definitions coincide with Definition 1 and Definition 2 when $\mu$ is the Lebesgue measure $\lambda$. Moreover, we can define general low-discrepancy sequences similarly to Definition 3 with respect this general star-discrepancy. There is also a generalized Koksma-Hlawka inequality [87], which we state below.

**Theorem 11** (Generalized Koksma-Hlawka inequality; Theorem 1 in [87]). *Let $f : [0, 1]^d \to \mathbb{R}$ be a measurable function whose mixed derivatives are absolutely integrable over its domain with bounded Hardy-Krause variation $V_{HK}(f) < \infty$. Let $\mu$ be a normalized Borel measure on $[0, 1]^d$, and let $x = \{x_\ell\}_{\ell=1}^N$ be a sequence of $N$ $d$-dimensional points in $[0, 1]^d$ with general star-discrepancy $D_N^*(d; \mu)$. Then,*

$$\left| \frac{1}{N} \sum_{\ell=1}^N f(x_\ell) - \int_{[0,1]^d} f(x) \, d\mu(x) \right| \leq V_{HK}(f) D_N^*(d; \mu). \tag{A.43}$$

# B  Constant sample complexity

In this section, we show that with a simple modification of the algorithm from [2], we can reduce the sample complexity to $\mathcal{O}(1)$ for a constant prediction error. We consider all of the same definitions/notation as in Appendix A.1. This section is similar to Section IV in the Supplementary Information of [2]. As in [2], our algorithm first maps the parameter space $[-1, 1]^m$ into a high-dimensional feature space $\mathbb{R}^{m_\phi}$ for $m_\phi$ given in Equation (A.8) via a feature map $\phi$. Our simple modification is to use the feature map defined by

$$\tilde{\phi}(x)_{x', P} \triangleq \text{sign}(\alpha_P) \sqrt{|\alpha_P|} \mathbb{1}\{x \in T_{x', P}\}, \tag{B.1}$$

where each coordinate of $\phi(x)$ is indexed by $P \in S^{(\text{geo})}, x' \in X_P$. Note that defining the feature map in this way requires knowledge of the observable $O = \sum_P \alpha_P P$ corresponding to the ground state property to be predicted. However, in practice, this is a natural assumption. The hypothesis class for our proposed ML algorithm consists of linear functions in this feature space, i.e., functions of the form $h(x) = \mathbf{w} \cdot \phi(x)$. Then, our algorithm learns these functions via ridge regression [65, 66]. For a chosen hyperparameter $\Lambda > 0$, ridge regression finds a vector $\mathbf{w}^*$ that solves the following optimization problem minimizing the training error subject to the constraint that $\|\mathbf{w}\|_2 \leq \Lambda$

$$\min_{\substack{\mathbf{w} \in \mathbb{R}^{m_\phi} \\ \|\mathbf{w}\|_2 \leq \Lambda}} \frac{1}{N} \sum_{\ell=1}^N |\mathbf{w} \cdot \tilde{\phi}(x_\ell) - y_\ell|^2, \tag{B.2}$$

where $y_\ell$ approximates $\text{tr}(O\rho(x_\ell))$. We denote the learned function by $h^*(x) = \mathbf{w}^* \cdot \tilde{\phi}(x)$. Note that the learned function does not need to achieve the minimum training error, but can be some amount say $\epsilon_3/2$ above it. For our purposes, we choose the hyperparameter to be $\Lambda = 2^{\mathcal{O}(\text{polylog}(1/\epsilon_1))}$, which we justify in the next section.

Note that there are two main differences from the algorithm in [2]. First, recall from Appendix A.1 that the feature map was previously defined as $\phi(x)_{x', P} = \mathbb{1}\{x \in T_{x', P}\}$ for $P \in S^{(\text{geo})}, x' \in X_P$. Second, instead of using LASSO ($\ell_1$-regularized regression), our proposed algorithm uses ridge regression.

With this algorithm, we obtain the following guarantee.

**Theorem 12** (Constant sample complexity; Detailed restatement of Theorem 4). *Let $1/e > \epsilon_1, \epsilon_2, \epsilon_3 > 0$ and $\delta > 0$. Given training data $\{(x_\ell, y_\ell)\}_{\ell=1}^N$ of size*

$$N = \log(1/\delta) 2^{\mathcal{O}(\log(1/\epsilon_3) + \text{polylog}(1/\epsilon_1))}, \tag{B.3}$$

*where $x_\ell$ is sampled from $\mathcal{D}$ and $y_\ell$ is an estimator of $\text{tr}(O\rho(x_\ell))$ such that $|y_\ell - \text{tr}(O\rho(x_\ell))| \leq \epsilon_2$, the ML algorithm can produce $h^*(x)$ that achieves prediction error*

$$\mathop{\mathbb{E}}_{x \sim \mathcal{D}} |h^*(x) - \text{tr}(O\rho(x))|^2 \leq (\epsilon_1 + \epsilon_2)^2 + \epsilon_3 \tag{B.4}$$

*with probability at least $1 - \delta$. The training time for constructing the hypothesis function $h^*$ and the prediction time for computing $h^*(x)$ are upper bounded by*

$$\mathcal{O}(n) \text{polylog}(1/\delta) 2^{\mathcal{O}(\log(1/\epsilon_3) + \text{polylog}(1/\epsilon_1))}. \tag{B.5}$$

Comparing to Theorem 7, notice that our sample complexity guarantee is completely independent of system size $n$.

The theorem in the main text corresponds to $\epsilon_1 = 0.2\epsilon, \epsilon_2 = \epsilon$, and $\epsilon_3 = 0.4\epsilon$. In this way, $(\epsilon_1 + \epsilon_2)^2 \le 1.44\epsilon^2 \le 0.53\epsilon$ and $(\epsilon_1 + \epsilon_2)^2 + \epsilon_3 \le \epsilon$.

So far, we have only considered the setting in which we learn a specific ground state property $\text{tr}(O\rho(x))$ for a fixed observable $O$. Because our training data is given in the form $\{(x_\ell, y_\ell)\}_{\ell=1}^N$, where $y_\ell$ approximates $\text{tr}(O\rho(x))$ for this fixed observable $O$, if we want to predict a new property for the same ground state $\rho(x)$, we would need to generate new training data. Thus, it may be more useful to learn a ground state representation, from which we could predict $\text{tr}(O\rho(x))$ for many different choices of observables $O$ without requiring new training data. In this case, suppose we are instead given training data $\{x_\ell, \sigma_T(\rho(x_\ell))\}_{\ell=1}^N$, where $\sigma_T(\rho(x_\ell))$ is a classical shadow representation [52, 68, 69, 70, 71] of the ground state $\rho(x_\ell)$. An immediate corollary of Theorem 12 is that we can predict ground state representations with the same sample complexity. This follows from the same proof as Corollary 5 in [2].

**Corollary 3** (Learning representations of ground states; detailed restatement of Corollary 1). *Let $1/e > \epsilon_1, \epsilon_2, \epsilon_3 > 0$ and $\delta > 0$. Given training data $\{(x_\ell, \sigma_T(\rho(x_\ell))\}_{\ell=1}^N$ of size*

$$N = \log(1/\delta)2^{\mathcal{O}(\log(1/\epsilon_3)+\text{polylog}(1/\epsilon_1))}, \tag{B.6}$$

*where $x_\ell$ is sampled from $\mathcal{D}$ and $\sigma_T(\rho(x_\ell))$ is the classical shadow representation of the ground state $\rho(x_\ell)$ using $T$ randomized Pauli measurements. For $T = \mathcal{O}(\log(nN/\delta)/\epsilon_2^2) = \tilde{\mathcal{O}}(\log(n/\delta)/\epsilon_2^2)$, the ML algorithm can produce a ground state representation $\hat{\rho}_{N,T}(x)$ that achieves*

$$\mathop{\mathbb{E}}_{x\sim\mathcal{D}} |\text{tr}(O\hat{\rho}_{N,T}(x)) - \text{tr}(O\rho(x))|^2 \le (\epsilon_1 + \epsilon_2)^2 + \epsilon_3 \tag{B.7}$$

*with probability at least $1 - \delta$, for any observable with eigenvalues between $-1$ and $1$ that can be written as a sum of geometrically local observables.*

We note that the number of measurements $T$ needed to generate the training data scales as $\log(n)$, but the amount of training data still remains constant with respect to system size. We do not consider the number of measurements as contributing to the sample complexity because in our setting, the ML algorithm is given this training data as input and does not generate it itself.

## B.1 Training error bound

To prove Theorem 12, we first derive a bound on the training error. Recall that the training error is defined as

$$\hat{R}(h) = \min_{\mathbf{w}} \frac{1}{N} \sum_{\ell=1}^N |h(x_\ell) - y_\ell|^2. \tag{B.8}$$

Define the vector $\tilde{\mathbf{w}}$ with entries indexed by $P \in S^{(\text{geo})}, x' \in X_P$ by

$$\tilde{\mathbf{w}}_{x',P} \triangleq \sqrt{|\alpha_P|} \, \text{tr}(P\rho(\chi_P(x))), \tag{B.9}$$

where $\chi_P(x)$ is defined in Equation (A.16). Then, notice that

$$\tilde{g}(x) \triangleq \tilde{\mathbf{w}} \cdot \tilde{\phi}(x) \tag{B.10}$$

$$= \sum_{P\in S^{(\text{geo})}} \sum_{x'\in X_P} \text{sign}(\alpha_P)|\alpha_P| \, \text{tr}(P\rho(\chi_P(x)))\mathbb{1}\{x \in T_{x',P}\} \tag{B.11}$$

$$= \sum_{P\in S^{(\text{geo})}} \sum_{x'\in X_P} \alpha_P \, \text{tr}(P\rho(\chi_P(x)))\mathbb{1}\{x \in T_{x',P}\} \tag{B.12}$$

$$= \mathbf{w}' \cdot \phi(x) \tag{B.13}$$

$$= g(x), \tag{B.14}$$

where $g(x) = \mathbf{w}' \cdot \phi(x)$ with $\mathbf{w}'_{x',P} = \alpha_P \, \text{tr}(P\rho(\chi_P(x)))$, $\phi(x)_{x',P} = \mathbb{1}\{x \in T_{x',P}\}$. By Lemma 1, we know that $g(x)$ approximates the ground state property with low training error, and thus, in turn, $\tilde{g}(x)$ also approximates the ground state property well. The existence of $\tilde{\mathbf{w}}$ such that $\tilde{g}(x) = \tilde{\mathbf{w}} \cdot \tilde{\phi}(x)$ guarantees that the function $h^*(x) = \mathbf{w}^* \cdot \tilde{\phi}(x)$ found by performing via ridge regression will also yield a small training error. More formally, we have the following guarantee

**Lemma 6** (Training error). *The function*

$$\tilde{g}(x) = \tilde{\mathbf{w}} \cdot \tilde{\phi}(x) = \sum_{P \in S^{(\text{geo})}} \sum_{x' \in X_P} \alpha_P \operatorname{tr}(P\rho(\chi_P(x))) \mathbb{1}\{x \in T_{x',P}\} \tag{B.15}$$

*achieves training error*

$$\hat{R}(\tilde{g}) \leq (\epsilon_1 + \epsilon_2)^2. \tag{B.16}$$

Since $\tilde{g}(x) = g(x)$, this follows directly from Lemma 1. Moreover, we can obtain an $\ell_2$-norm bound on $\tilde{\mathbf{w}}$. We can utilize this upper bound to choose the hyperparameter $\Lambda > 0$ such that $\|\mathbf{w}\|_2 \leq \Lambda$. Thus, we have the following lemma,

**Lemma 7** ($\ell_2$-Norm bound). *Let $\tilde{\mathbf{w}}$ be the vector of coefficients defined in Equation* (B.9). *Then, we have*

$$\|\tilde{\mathbf{w}}\|_2^2 = \sum_{P \in S^{(\text{geo})}} \sum_{x' \in X_P} |\alpha_P| |\operatorname{tr}(P\rho(\chi_P(x)))|^2 = 2^{\mathcal{O}(\text{polylog}(1/\epsilon_1))}. \tag{B.17}$$

*Proof.* This is a simple consequence of Lemma 4. Explicitly, we have

$$\|\tilde{\mathbf{w}}\|_2 = \sum_{P \in S^{(\text{geo})}} \sum_{x' \in X_P} |\alpha_P| |\operatorname{tr}(P\rho(\chi_P(x)))|^2 \tag{B.18}$$

$$\leq \sum_{P \in S^{(\text{geo})}} \sum_{x' \in X_P} |\alpha_P| |\operatorname{tr}(P\rho(\chi_P(x)))| \tag{B.19}$$

$$= 2^{\mathcal{O}(\text{polylog}(1/\epsilon_1))}, \tag{B.20}$$

where the second line follows because $\operatorname{tr}(P\rho(\chi_P(x))) \leq 1$ and the last line follows by Lemma 4. $\square$

This justifies our choice of $\Lambda = 2^{\mathcal{O}(\text{polylog}(1/\epsilon_1))}$. Now consider the learned function $h^*(x) = \mathbf{w}^* \cdot \tilde{\phi}(x)$, where $\mathbf{w}^*$ is found by minimizing the training error subject to the constraint that $\|\mathbf{w}\|_2 \leq \Lambda$. We do not require the learned function to achieve the minimum training error, but it can be some amount $\epsilon_3/2$ above it, i.e.,

$$\hat{R}(h^*) \leq \frac{\epsilon_3}{2} + \min_{\substack{\mathbf{w} \\ \|\mathbf{w}\|_2 \leq \Lambda}} \frac{1}{N} \sum_{\ell=1}^{N} |\mathbf{w} \cdot \tilde{\phi}(x_\ell) - y_\ell|^2. \tag{B.21}$$

Since we chose $\Lambda = 2^{\mathcal{O}(\text{polylog}(1/\epsilon_1))}$ and we showed in Lemma 7 that $\|\tilde{\mathbf{w}}\|_2 \leq \Lambda$, then the minimum training error is at most $\hat{R}(\tilde{g})$. We also know that this is bounded by $(\epsilon_1 + \epsilon_2)^2$ by Lemma 6. This then implies

$$\hat{R}(h^*) \leq \frac{\epsilon_3}{2} + \hat{R}(g) \leq (\epsilon_1 + \epsilon_2)^2 + \frac{\epsilon_3}{2}. \tag{B.22}$$

## B.2 Prediction error bound

To prove Theorem 12, it remains to bound the prediction error. We can use a standard result from machine learning theory on the prediction error of ridge regression algorithms [66, 64].

**Theorem 13** (Theorem 26.12 in [66]). *Suppose that $\mathcal{D}$ is a distribution over $\mathcal{X} \times \mathcal{Y}$ such that with probability 1 we have that $\|\mathbf{x}\|_2 \leq R$. Let $\mathcal{H} = \{\mathbf{x} \mapsto \mathbf{w} \cdot \mathbf{x} : \|\mathbf{w}\|_2 \leq \Lambda\}$ and let $\ell : \mathcal{H} \times Z \to \mathbb{R}$ be a loss function of the form $\ell(\mathbf{w}, (\mathbf{x}, y)) = \phi(\mathbf{w} \cdot \mathbf{x}, y)$ such that for all $y \in \mathcal{Y}, a \mapsto \phi(a, y)$ is a $\rho$-Lipschitz function and such that $\max_{a \in [-\Lambda R, \Lambda R]} |\phi(a, y)| \leq c$. Then, for any $\delta \in (0, 1)$, with probability of at least $1 - \delta$ over the choice of an i.i.d. sample of size $N$, for all $h \in \mathcal{H}$,*

$$R(h) \leq \hat{R}_S(h) + \frac{2\rho\Lambda R}{\sqrt{N}} + c\sqrt{\frac{2\log(2/\delta)}{N}}. \tag{B.23}$$

Here, $R(h)$ denotes the prediction error for the hypothesis $h$. With this, we can complete the proof of Theorem 12.

*Proof of Theorem 12.* First, let us reframe the theorem in our setting. Consider the input space $\mathcal{X}$ to be the parameter space $[-1,1]^m$ and our input variable is $\mathbf{x} = \tilde{\phi}(x)$. Since the observables we consider have spectral norm at most 1, the output space fulfils $\mathcal{Y} \subseteq [-1,1]$. The hypothesis set is $\mathcal{H} = \{x \mapsto \mathbf{w} \cdot \tilde{\phi}(x) : \|\mathbf{w}\|_2 \leq \Lambda\}$, where in the previous section, we set $\Lambda = 2^{\mathcal{O}(\text{polylog}(1/\epsilon_1))}$.

It remains to check the conditions of the theorem. We begin by showing that $\|\mathbf{x}\|_2 \leq R$ for some $R > 0$. We have the following computation:

$$\|\mathbf{x}\|_2 = \tilde{\phi}(x) \cdot \tilde{\phi}(x) = \left\|\tilde{\phi}(x)\right\|_2^2 \tag{B.24}$$

$$= \sum_{P \in S^{(\text{geo})}} \sum_{x' \in X_P} |\text{sign}(\alpha_P)\sqrt{|\alpha_P|}\mathbb{1}\{x \in T_{x',P}\}|^2 \tag{B.25}$$

$$= \sum_{P \in S^{(\text{geo})}} \sum_{x' \in X_P} |\alpha_P|\mathbb{1}\{x \in T_{x',P}\} \tag{B.26}$$

$$= \sum_{P \in S^{(\text{geo})}} |\alpha_P| \tag{B.27}$$

$$= \mathcal{O}(1), \tag{B.28}$$

where the second to last line follows because for a given $P$, $x \in T_{x',P}$ for exactly one $x' \in X_P$. This is shown in Corollary 3 of [2]. Also, the last line follows by Theorem 8. Thus, we can take $R = \mathcal{O}(1)$.

Finally, note that $\ell(\mathbf{w},(\mathbf{x},y)) = |\mathbf{w} \cdot \mathbf{x} - y| = \phi(\mathbf{w} \cdot \mathbf{x}, y)$. Therefore, $\phi(a,y)$ is a 1-Lipschitz function and fulfils

$$\max_{a \in [-\Lambda R, \Lambda R]} |\phi(a,y)| = \max_{a \in [-\Lambda R, \Lambda R]} |a - y| \leq \Lambda R + 1. \tag{B.29}$$

Thus, we can consider $\rho = 1$ and $c = \mathcal{O}(1) \cdot 2^{\mathcal{O}(\text{polylog}(1/\epsilon_1))} + 1$.

By Equation (B.22), we know that the learned model $h^*(x) = \mathbf{w}^* \cdot \tilde{\phi}(x)$ achieves

$$\hat{R}(h^*) \leq (\epsilon_1 + \epsilon_2)^2 + \frac{\epsilon_3}{2}. \tag{B.30}$$

Plugging in $R = \mathcal{O}(1)$, $\rho = 1$, $\Lambda = 2^{\mathcal{O}(\text{polylog}(1/\epsilon_1))}$ and $c = \mathcal{O}(1) \cdot 2^{\mathcal{O}(\text{polylog}(1/\epsilon_1))} + 1$ into Theorem 13, we have

$$R(h^*) \leq (\epsilon_1 + \epsilon_2)^2 + \frac{\epsilon_3}{2} \tag{B.31}$$

$$+ \frac{1}{\sqrt{N}} \left(2\mathcal{O}(1) \cdot 2^{\mathcal{O}(\text{polylog}(1/\epsilon_1))} + \left(\mathcal{O}(1) \cdot 2^{\mathcal{O}(\text{polylog}(1/\epsilon_1))} + 1\right) \sqrt{2\log(2/\delta)}\right) \tag{B.32}$$

with probability at least $1 - \delta$. In order to bound the prediction error by $(\epsilon_1 + \epsilon_2)^2 + \epsilon_3$, we need $N$ to be large enough such that

$$\frac{1}{\sqrt{N}} \left(2\mathcal{O}(1) \cdot 2^{\mathcal{O}(\text{polylog}(1/\epsilon_1))} + \left(\mathcal{O}(1) \cdot 2^{\mathcal{O}(\text{polylog}(1/\epsilon_1))} + 1\right) \sqrt{2\log(2/\delta)}\right) \leq \frac{\epsilon_3}{2}. \tag{B.33}$$

Solving for $N$ in this inequality and simplifying we have

$$N = \frac{4}{\epsilon_3^2} 2^{\mathcal{O}(\text{polylog}(1/\epsilon_1))}(1 + \sqrt{\log(1/\delta)})^2 = 2^{\mathcal{O}(\log(1/\epsilon_3) + \text{polylog}(1/\epsilon_1))} \log(1/\delta). \tag{B.34}$$

Thus, for this $N$, we can guarantee that $R(h^*) \leq (\epsilon_1 + \epsilon_2)^2 + \epsilon_3$, as claimed. □

On another note, when considering a scenario with a fixed number of parameters $m = \mathcal{O}(1)$, much like the setting in [51], the expression derived from the result in Lemma 10 exhibits polynomial dependence on $\epsilon$. One can incorporate the constant number of parameters by setting $\tilde{m} = m$. Thus, we recover the exact ground state properties $\text{tr}(P\rho(x))$ in $f_P$ and the approximation error resulting from applying Lemma 2 vanishes completely. Furthermore, we can slightly adapt the proof of Lemma 7 and obtain

$$\|\tilde{\mathbf{w}}\|_2^2 = \sum_{P \in S^{(\text{geo})}} \sum_{x' \in X_P} |\alpha_P| |\text{tr}(P\rho(\chi_P(x)))| = \max_{P \in S^{(\text{geo})}} |X_P| \sum_{Q \in S^{(\text{geo})}} |\alpha_P| = \mathcal{O}(\epsilon^{-m}), \tag{B.35}$$

where the last step is performed similarly as in the proof of Lemma 4.

### B.3 Computational time for training and prediction

It remains to analyze the computational time for the ML algorithm's training and prediction.

*Proof of computational time in Theorem 12.* The training time is dominated by the time required for ridge regression over the feature space defined by the feature map $\phi$. Recall that the optimization problem under considerations is

$$\min_{\substack{\mathbf{w} \in \mathbb{R}^{m_\phi} \\ \|\mathbf{w}\|_2 \leq \Lambda}} \frac{1}{N} \sum_{\ell=1}^{N} |\mathbf{w} \cdot \tilde{\phi}(x_\ell) - y_\ell|^2. \tag{B.36}$$

One can show that this is a convex optimization problem so that we can solve its equivalent dual problem instead. This dual optimization problem is given by

$$\max_{\alpha \in \mathbb{R}^N} -\alpha^\intercal (K + \lambda I)\alpha + 2\alpha \cdot Y, \tag{B.37}$$

where the kernel matrix is $K = X^\intercal X$, for the feature matrix $X \in \mathbb{R}^{m_\phi \times N}$ defined by $X = (\tilde{\phi}(x_1) \cdots \tilde{\phi}(x_N))$ and the response vector $Y = (y_1, \ldots, y_N)^\intercal$. If $\kappa$ is the maximum time it takes to compute a kernel entry $K(x, x') = \tilde{\phi}(x) \cdot \tilde{\phi}(x')$, then one can show that the time to solve this dual problem is $\mathcal{O}(\kappa N^2 + N^3)$. Moreover, prediction can be executed in $\mathcal{O}(\kappa N)$. For more details in this analysis, we refer the reader to, e.g., Section 11.3.2 of [64].

In our case, $\kappa = \mathcal{O}(m_\phi)$ since $\tilde{\phi}(x) \in \mathbb{R}^{m_\phi}$ and the kernel is simply the dot product of two of these vectors. By Equation (A.8), we know that

$$m_\phi = \mathcal{O}(n)2^{\mathcal{O}(\mathrm{polylog}(1/\epsilon_1))} \tag{B.38}$$

so that $\kappa = \mathcal{O}(n)2^{\mathcal{O}(\mathrm{polylog}(1/\epsilon_1))}$. Moreover, by Theorem 12,

$$N = \log(1/\delta)2^{\mathcal{O}(\log(1/\epsilon_3) + \mathrm{polylog}(1/\epsilon_1))}. \tag{B.39}$$

Plugging this into the time required to solve the dual problem for kernel ridge regression, we have

$$\mathcal{O}(\kappa N^2 + N^3) = \mathcal{O}(n)\mathrm{polylog}(1/\delta)2^{\mathcal{O}(\log(1/\epsilon_3) + \mathrm{polylog}(1/\epsilon_1))}. \tag{B.40}$$

Moreover, the prediction time is given by

$$\mathcal{O}(\kappa N) = \mathcal{O}(n)\mathrm{polylog}(1/\delta)2^{\mathcal{O}(\log(1/\epsilon_3) + \mathrm{polylog}(1/\epsilon_1))}. \tag{B.41}$$

$\square$

## C  Rigorous guarantees for neural networks

In this section, we derive a rigorous guarantee on the sample complexity of a deep-learning based model for predicting ground state properties. Similarly to the previous sections, let $1/e > \epsilon_1, \epsilon_2, \epsilon_3 > 0$ throughout. One can think of $\epsilon_1$ as the approximation error caused by our neural network model not exactly capturing the ground state property; $\epsilon_2$ represents the noise in the training data; $\epsilon_3$ corresponds to the generalization error.

Recall again the setup, where we consider a family of $n$-qubit Hamiltonians $H(x) = \sum_{j=1}^{L} h_j(\vec{x}_j)$ parameterized by an $m$-dimensional vector $x \in [-1, 1]^m$, which satisfies the assumptions (a)-(c) in Appendix A.1. Let $\rho(x)$ denote the ground state of $H(x)$. We consider the task of predicting ground state properties $\mathrm{tr}(O\rho(x))$ for some observable $O$ that satisfies assumption (d) in Appendix A.1, where we are given training data $\{(x_\ell, y_\ell)\}_{\ell=1}^{N}$ with $y_\ell \approx \mathrm{tr}(O\rho(x_\ell))$. In particular, suppose $|y_\ell - \mathrm{tr}(O\rho(x_\ell))| \leq \epsilon_2$. Furthermore, we also assume that all mixed partial derivatives of order $\tilde{m} \triangleq |I_P|$ of $h_j$ are bounded as

$$\left\| \frac{\partial^{\tilde{m}}}{\partial x_1 \partial x_2 \ldots \partial x_{\tilde{m}}} h_j(x) \right\|_\infty \leq 1, \tag{C.1}$$

where $I_P$ is the set of local coordinates defined in Equation (A.2). Here, we denote the number of local coordinates by $\tilde{m} = |I_P|$ for ease of notation. This is similar in spirit to assumption (b), in

which we assume that the local terms have bounded directional derivatives: $\|\partial h_j / \partial \hat{u}\|_\infty \leq 1$, where $\hat{u}$ is a unit vector in parameter space.

Let $S^{(\text{geo})}$ denote the set of geometrically local Pauli observables. Our deep neural network model consists of $|S^{(\text{geo})}| = \mathcal{O}(n)$ "local" multi-layer perceptron models (defined in generally in Appendix A.2) with two hidden layers and $\tanh$ activation functions. Their outputs are combined through a linear layer without activation function. Formally, our model is defined as follows.

**Definition 6** (Deep neural network model). *The neural network model is given by a function* $f^{\Theta,w} : [-1,1]^m \to \mathbb{R}$ *defined by*

$$f^{\Theta,w}(x) = \sum_{P \in S^{(\text{geo})}} w_P f_P^{\theta_P}(x), \tag{C.2}$$

*where the "local models"* $f_P^{\theta_P} : [-1,1]^{\tilde{m}} \to \mathbb{R}$ *are given by*

$$f_P^{\theta_P}(x) = (W_{\text{out}} \circ \tanh \circ W_{\text{hidden}} \circ \tanh \circ W_{\text{in}} \circ \tau^{-1})(x), \tag{C.3}$$

*with* $\tau^{-1}(x) = (x+1)/2$ *and* $\theta_P = [(W_{\text{in}}, b_{\text{in}}), (W_{\text{hidden}}, b_{\text{hidden}}), (W_{\text{out}}, b_{\text{out}})]$. *Here,* $W_{\text{in}} \in \mathbb{R}^{\tilde{m} \times W}$, $b_{\text{in}} \in \mathbb{R}^W$, $W_{\text{hidden}} \in \mathbb{R}^{W \times W}$, $b_{\text{hidden}} \in \mathbb{R}^W$, $W_{\text{out}} \in \mathbb{R}^{W \times 1}$ *and* $b_{\text{out}} \in \mathbb{R}$, *where* $W$ *denotes the width of the hidden layers. The weights are given by* $\Theta = \{\theta_P : P \in S^{(\text{geo})}\}$ *in the local models and* $w \in \mathbb{R}$ *in the last layer. Furthermore, we denote the individual parameters by* $\Theta_i \in \mathbb{R}$.

Using this model, we can establish an objective function that we aim to minimize during the training process. Specifically, this objective function comprises the mean square error along with a lasso penalty applied to the weights $w$ in the final layer.

**Definition 7** (Training objective). *Let* $f^{\Theta,w}$ *be a neural network model as in Definition 6. Let* $\{(x_\ell, y_\ell)\}_{\ell=1}^N$ *be the training data set and* $\lambda > 0$ *be some regularization parameter that may depend on* $\epsilon_1, \epsilon_2 > 0$. *The training objective is given by*

$$\frac{1}{N} \sum_{\ell=1}^N |f^{\Theta,w}(x_\ell) - y_\ell|^2 + \lambda \|w\|_1. \tag{C.4}$$

Our proposed ML algorithm then operates as in Algorithm 1.

---

**Algorithm 1:** Deep learning-based prediction of ground state properties

---

Sample $N$ low-discrepancy points $\{x_\ell\}_{\ell=1}^N$;
Collect training labels $\{y_\ell\}_{\ell=1}^N$, where $y_\ell \approx \text{tr}(O\rho(x_\ell))$;
**Data:** $\{(x_\ell, y_\ell)\}_{\ell=1}^N$;
Fix $|I_P|$;
Initialize model architecture according to Definition 6 with appropriate hyperparameter $\delta_1$, width $W$ as in Theorem 16 and weights $\Theta, w$ using an appropriate initialization method (e.g., Xavier initialization [72]);
Train with respect to the objective in Definition 7 with appropriate hyperparameter $\lambda > 0$ using a quasi-Monte Carlo training algorithm, e.g., Adam [73] until convergence;
Obtain locally optimal parameters $\Theta^*, w^*$;
**Result:** Classical representation $f^{\Theta^*, w^*}$;

---

After training our model using Algorithm 1, we obtain the following rigorous guarantee.

**Theorem 14** (Neural network sample complexity guarantee). *Let* $1/e > \epsilon_1, \epsilon_2, \epsilon_3 > 0$. *Let* $f^{\Theta^*, w^*} : [-1,1]^m \to \mathbb{R}$ *be a neural network model produced from Algorithm 1 trained on data* $\{(x_\ell, y_\ell)\}_{\ell=1}^N$ *of size*

$$N = 2^{\mathcal{O}(\text{polylog}(1/\epsilon_1) + \text{polylog}(1/\epsilon_3))}, \tag{C.5}$$

*where the* $x_\ell$*'s form a low-discrepancy Sobol sequence and* $|y_\ell - \text{tr}(O\rho(x_\ell))| \leq \epsilon_2$. *Suppose that* $f^{\Theta^*, w^*}$ *achieves a training error of at most* $((\epsilon_1 + \epsilon_2)^2 + \epsilon_3)/2$. *Additionally, suppose that all parameters* $\Theta_i^*$ *of* $f^{\Theta^*, w^*}$ *satisfy* $|\Theta_i^*| \leq W_{\max}$, *for some* $W_{\max} > 0$ *that is independent of the system size* $n$. *Then the neural network* $f^{\Theta^*, w^*}$ *achieves prediction error*

$$\mathop{\mathbb{E}}_{x \sim U[-1,1]^m} |f^{\Theta^*, w^*}(x) - \text{tr}(O\rho(x))|^2 \leq 2(\epsilon_1 + \epsilon_2)^2 + \epsilon_3, \tag{C.6}$$

*where* $x \sim U[-1,1]^m$ *denotes sampling* $x$ *from a uniform distribution over* $[-1,1]^m$.

We prove this theorem in the next two sections (Appendices C.1 and C.2). As a corollary of this, we obtain the theorem stated in the main text. We discuss the assumptions that the distribution $\mathcal{D}$ must satisfy in depth and prove the corollary in Appendix C.3. The theorem in the main text (Theorem 5) corresponds to $\epsilon_1 = \epsilon_3 = 0.1\epsilon$ and $\epsilon_2 = \epsilon$. Hence, $2(\epsilon_1 + \epsilon_2)^2 \leq 2.44\epsilon^2 \leq 0.9\epsilon$ for $1/e > \epsilon > 0$ and so $2(\epsilon_1 + \epsilon_2)^2 + \epsilon_3 \leq \epsilon$.

**Corollary 4** (Neural network sample complexity guarantee; detailed restatement of Theorem 5). *Let $1/e > \epsilon_1, \epsilon_2, \epsilon_3 > 0$, $\mathcal{D}$ a distribution with PDF $g$ satisfying the following properties: $g$ has full support and is continuously differentiable on $[-1, 1]^m$. Moreover, $g$ is of the form*

$$g(x) = \prod_{j=1}^{L} g_j(\vec{x}_j). \tag{C.7}$$

*Let $f^{\Theta^*, w^*} : [-1, 1]^m \to \mathbb{R}$ be a neural network model produced from Algorithm 1 trained on data $\{(x_\ell, y_\ell)\}_{\ell=1}^{N}$ of size*

$$N = \log(1/\delta) 2^{\mathcal{O}(\mathrm{polylog}(1/\epsilon_1) + \mathrm{polylog}(1/\epsilon_3))}, \tag{C.8}$$

*where the $x_\ell \sim \mathcal{D}$ and $|y_\ell - \mathrm{tr}(O\rho(x_\ell))| \leq \epsilon_2$. Suppose that $f^{\Theta^*, w^*}$ achieves a training error of at most $((\epsilon_1 + \epsilon_2)^2 + \epsilon_3)/2$. Additionally, suppose that all parameters $\Theta_i^*$ of $f^{\Theta^*, w^*}$ satisfy $|\Theta_i^*| \leq W_{\max}$, for some $W_{\max} > 0$ that is independent of the system size $n$. Then the neural network $f^{\Theta^*, w^*}$ achieves prediction error*

$$\mathop{\mathbb{E}}_{x \sim \mathcal{D}} |f^{\Theta^*, w^*}(x) - \mathrm{tr}(O\rho(x))|^2 \leq 2(\epsilon_1 + \epsilon_2)^2 + \epsilon_3, \tag{C.9}$$

*with probability at least $1 - \delta$.*

Moreover, while we can show the existence of suitable parameters that achieve a low training error, quantified by our training objective in Definition 7, in general, we cannot prove that Algorithm 1 converges to parameters close to this optimum because our training objective is not convex. Thus, to obtain the guarantee in Theorem 14, we need to assume that a low training error is indeed achieved by Algorithm 1. This is commonly satisfied in practice.

Similar to Corollary 3, if we are instead given training data $\{x_\ell, \sigma_T(\rho(x_\ell))\}_{\ell=1}^{N}$, where $\sigma_T(\rho(x_\ell))$ is a classical shadow representation [52, 68, 69, 70, 71] of the ground state $\rho(x_\ell)$, then an immediate corollary of Theorem 14 is that we can predict ground state representations with the same sample complexity. This follows from the same proof as Corollary 5 in [2].

**Corollary 5** (Learning representations of ground states with neural networks; detailed restatement of Corollary 2). *Let $1/e > \epsilon_1, \epsilon_2, \epsilon_3 > 0$ and $\delta > 0$. Given training data $\{(x_\ell, \sigma_T(\rho(x_\ell)))\}_{\ell=1}^{N}$ of size*

$$N = \log(1/\delta) 2^{\mathcal{O}(\mathrm{polylog}(1/\epsilon_3) + \mathrm{polylog}(1/\epsilon_1))}, \tag{C.10}$$

*where $x_\ell$ is sampled from a distribution $\mathcal{D}$ satisfying the same assumptions as Corollary 4 and $\sigma_T(\rho(x_\ell))$ is the classical shadow representation of the ground state $\rho(x_\ell)$ using $T$ randomized Pauli measurements. For $T = \mathcal{O}(\log(nN/\delta)/\epsilon_2^2) = \tilde{\mathcal{O}}(\log(n/\delta)/\epsilon_2^2)$, the ML algorithm can produce a ground state representation $\hat{\rho}_{N,T}(x)$ that achieves*

$$\mathop{\mathbb{E}}_{x \sim \mathcal{D}} |\mathrm{tr}(O\hat{\rho}_{N,T}(x)) - \mathrm{tr}(O\rho(x))|^2 \leq 2(\epsilon_1 + \epsilon_2)^2 + \epsilon_3 \tag{C.11}$$

*with probability at least $1 - \delta$, for any observable with eigenvalues between $-1$ and $1$ that can be written as a sum of geometrically local observables.*

We review low-discrepancy sequences and techniques in quasi-Monte Carlo theory in Appendix A.2, which we use in our proof. To prove Theorem 14, we first show that there exists weights $\Theta', w'$ such that our proposed neural network $f^{\Theta', w'}$ achieves a low training error, i.e., it approximates the ground state properties $\mathrm{tr}(O\rho(x))$ well. We show this using results in classical deep learning theory about approximating arbitrary functions with neural networks [53]. Then, we use the Koksma-Hlawka inequality (Theorem 10) to bound the prediction error of our model, similarly to [54]. As we want to derive a bound which is independent of the size of the physical system, our approach requires some additional steps. Since the dimension of the input domain of our model depends on the size of the physical system, we cannot treat it as constant as in [54]. Therefore, we bound the prediction error with respect to local functions, whose domain size is independent of the system size.

Moreover, recall that the Koksma-Hlawka inequality produces a bound in terms of the star-discrepancy (Definition 2) and the Hardy-Krause variation. The star-discrepancy can be bounded by considering low-discrepancy sequences (Definition 3), and the Hardy-Krause variation can be bounded by Equation (A.36). We derive explicit bounds for the Hardy-Krause variation of the ground state properties $\mathrm{tr}(O\rho(x))$, using tools from the spectral flow formalism [59, 60, 61]. To obtain Corollary 4, we follow a similar proof but use results relating the discrepancy with respect to the Lebesgue measure to the discrepancy with respect to arbitrary measures and bounds on the discrepancy of uniformly random points (Appendix A.2).

In Appendix C.1, we prove that our model approximates the ground state properties well. Then, in Appendix C.2, we use the Koksma-Hlawka inequality to bound the prediction error of our model. Technical results explicitly bounding the mixed partial derivatives of the ground state properties are found in Appendix C.4. We use these in Appendix C.2 to bound the Hardy-Krause variation. We then generalize this to data sampled from different distributions, as in Corollary 4, in Appendix C.3.

## C.1 Approximation of ground state properties by neural networks

In this section, we prove that when choosing the number of parameters and width of the model appropriately, there exists a parameter set for which the deep neural network model approximates the ground state properties well. This shows the existence of a neural network with low training error. The proof is a direct application of the main result from [53], which proves that tanh neural networks can approximate sufficiently smooth functions, in combination with the bounds on the mixed derivative of $\mathrm{tr}(P\rho(x))$ we derived in Appendix C.4.

We consider the local functions defined as in Appendix A.1.2. Namely, define $f(x) = \sum_{P \in S^{(\mathrm{geo})}} \alpha_P f_P(x)$, where $f_P(x) = \mathrm{tr}(P\rho(\chi_P(x)))$ for $O = \sum_{P \in \{I,X,Y,Z\}^{\otimes n}} \alpha_P P$ and

$$\chi_P(x)_c = \begin{cases} x_c, & c \in I_P \\ 0 & c \notin I_P \end{cases} \tag{C.12}$$

for all $c \in \{1, \ldots, m\}$, for $I_P$ defined in Equation (A.2). Note that here, we slightly alter the definition from Appendix A.1.2, where we do not include the coefficient $\alpha_P$ in the definition of $f_P(x)$. Because all parameters with coordinates not in $I_P$ are set to $0$, we can view $f_P$ as a function taking inputs in $[-1, 1]^{\tilde{m}}$, where recall that we use $\tilde{m} = |I_P|$ to denote the number of local coordinates.

We show that there exists a neural network that can approximate these local functions $f_P$ well. In order to apply the result from [53] to approximate $\mathrm{tr}(P\rho(\chi_P(x)))$, we need to transform its inputs, such that it becomes a map $[0, 1]^{\tilde{m}} \to \mathbb{R}$. Therefore, we introduce an appropriate function $\tau(x) = 2x - 1$. To avoid confusion when considering the different domains $[-1, 1]^m$ versus $[0, 1]^m$, if an input $x \in [-1, 1]^m$, we simply denote it by $x$. If $x \in [0, 1]^m$, we denote it by $\bar{x}$. We similarly use this notation for other domain dimensions.

In the following lemma, we use $W^{k,\infty}(\Omega)$ for $\Omega \subseteq \mathbb{R}^m$ to denote the Sobolev space

$$W^{k,\infty}(\Omega) \triangleq \left\{ f \in L^\infty(\Omega) : \frac{\partial^{|\alpha|} f}{\partial x_1^{\alpha_1} \cdots \partial x_m^{\alpha_m}} \in L^\infty(\Omega) \text{ for all } \alpha \in \mathbb{N}_0^m \text{ with } |\alpha| \leq k \right\} \tag{C.13}$$

so that the Sobolev norm is defined as

$$\|f\|_{W^{k,\infty}(\Omega)} \triangleq \max_{|\alpha|=s} \left\| \frac{\partial^{|\alpha|} f}{\partial x_1^{\alpha_1} \cdots \partial x_m^{\alpha_m}} \right\|_{L^\infty(\Omega)}. \tag{C.14}$$

for $\alpha \in \mathbb{N}_0^m$ and the $L^\infty$-norm is defined by

$$\|f\|_{L^\infty(\Omega)} = \sup_{x \in \Omega} \|f\|. \tag{C.15}$$

**Lemma 8** (Existence of approximating neural network). *Let $\epsilon_1 > 0$, $s, M \in \mathbb{N}$. Let $\|H(x)\|_{W^{s,\infty}([-1,1]^m)} \leq 1$. Define functions $f_P : [0, 1]^{\tilde{m}} \to \mathbb{R}$ as $f_P(\tau(\bar{x})) = \mathrm{tr}(P\rho(\chi_P(\tau(\bar{x}))))$, where $\tau(\bar{x}) = 2\bar{x} - 1$. Then, there exist neural networks $\hat{f}_P^M$, such that*

$$\left\| f_P \circ \tau - \hat{f}_P^M \right\|_{L^\infty([0,1]^{\tilde{m}})} \leq \epsilon_1 \tag{C.16}$$

*with at most $\epsilon_1^{-\frac{\tilde{m}+1}{s}} 2^{\mathcal{O}(\tilde{m} \log(\tilde{m}))}$ parameters. Furthermore, the weights scale as $2^{\mathrm{poly}(\log(1/\epsilon_1), \tilde{m}, s)}$.*

To prove this, we utilize the main result from [53], which states that a neural network with $\tanh$ activation functions can approximate effectively any function.

**Theorem 15** (Theorem 5.1 in [53]). *Let $d, s \in \mathbb{N}$, $R > 0$, $d > 0$ and $f \in W^{s,\infty}([0,1]^d)$. There exist constants $\mathcal{C}(d,k,s,f)$, $N_0(d) > 0$, such that for every $N \in \mathbb{N}$ with $N > N_0(d)$ there exists a $\tanh$ neural network $\hat{f}^N$ with two hidden layers, one of width at most $3\lceil \frac{s}{2}\rceil |P_{s-1,d+1}| + d(N-1)$ (where $|P_{n,d}| = \binom{n+d-1}{n}$) and another of width at most $3\lceil \frac{d+2}{2}\rceil |P_{d+1,d+1}|N^d$ (or $3\lceil \frac{s}{2}\rceil + N - 1$ and $6N$ for $d = 1$), such that,*

$$\|f - \hat{f}^N\|_{L^\infty([0,1]^d)} \leq (1 + \delta)\frac{\mathcal{C}(d, 0, s, f)}{N^s}, \tag{C.17}$$

*and for $k = 1, \ldots, s - 1$,*

$$\|f - \hat{f}^N\|_{W^{k,\infty}([0,1]^d)} \leq 3^d(1+\delta)(2(k+1))^{3k}\max\left\{R^k, \ln^k\left(\beta N^{s+d+2}\right)\right\}\frac{\mathcal{C}(d,k,s,f)}{N^{s-k}}, \tag{C.18}$$

*where we define*

$$\beta = \frac{k^3 2^d \sqrt{d}\max\{1, \|f\|_{W^{k,\infty}([0,1]^d)}^{1/2}\}}{\delta \min\{1, \sqrt{\mathcal{C}(d,k,s,f)}\}}. \tag{C.19}$$

*If $f \in C^s([0,1]^d)$, then it holds that*

$$\mathcal{C}(d,k,s,f) = \max_{0 \leq l \leq k}\frac{1}{(s-l)!}\left(\frac{3d}{2}\right)^{s-l}|f|_{W^{s,\infty}([0,1]^d)}, \qquad N_0(d) = \frac{3d}{2}, \tag{C.20}$$

*and else it holds that*

$$\mathcal{C}(d,k,s,f) = \max_{0 \leq l \leq k}\frac{\pi^{1/4}\sqrt{s}}{(s-l-1)!}\left(5d^2\right)^{s-l}|f|_{W^{s,\infty}([0,1]^d)}, \qquad N_0(d) = 5d^2. \tag{C.21}$$

*In addition, the weights of $\hat{f}^N$ scale as $\mathcal{O}\left(\mathcal{C}^{-s/2}N^{d(d+s^2+k^2)/2}(s(s+2))^{3s(s+2)}\right)$.*

The proof of Lemma 8 follows by an application of Theorem 15.

*Proof of Lemma 8.* We directly apply Theorem 15, with the input space dimension $\tilde{m}$, where $\tilde{m}$ is the number of local parameters $\tilde{m} = |I_P|$. By Corollary 8, then we have

$$\left\|f_P \circ \tau - \hat{f}_P^M\right\|_{L^\infty([0,1]^{\tilde{m}})} \leq \frac{(1+\delta)}{s!}\left(\frac{3\tilde{m}Cs^2}{2M}\right)^s. \tag{C.22}$$

We want to show that this is bounded above by $\epsilon_1$. By rearranging, we find that this holds when

$$\epsilon_1^{-\frac{1}{s}}\left(\frac{(1+\delta)}{s!}\right)^{\frac{1}{s}}\frac{3}{2}\tilde{m}Cs^2 \leq M. \tag{C.23}$$

Note that by composing with $f_P$ with $\tau$, we acquire an extra factor of $2^s$, which can be considered a component of $C$. Using $M = \mathcal{O}(\epsilon_1^{-\frac{1}{s}}\tilde{m}s^2)$, this results in the widths of the two layers being

$$3\left\lceil\frac{s}{2}\right\rceil\binom{s+\tilde{m}}{\tilde{m}+1} + \tilde{m}(M-1) \text{ and } \left\lceil\frac{\tilde{m}+2}{2}\right\rceil\binom{2\tilde{m}+2}{d+1}M^{\tilde{m}}, \tag{C.24}$$

and therefore at most

$$(c_1\tilde{m})^{c_2\tilde{m}}\epsilon_1^{-\frac{\tilde{m}+1}{s}} = 2^{\mathcal{O}(\tilde{m}(\log(\tilde{m})+\log(1/\epsilon_1)/s))} \tag{C.25}$$

trainable weights in the neural network. The constants $c_1$ and $c_2$ are independent of $\tilde{m}$, but may depend on $s$. By Theorem 15, the weights scale as

$$\mathcal{O}\left(\mathcal{C}^{-s/2}\left(\epsilon_1^{-\frac{1}{s}}\tilde{m}s^2\right)^{\tilde{m}(\tilde{m}+s^2+k^2)/2}(s(s+2))^{3s(s+2)}\right) = \epsilon_1^{-\frac{\tilde{m}+1}{s}}2^{\mathcal{O}(\tilde{m}\log(\tilde{m}))}. \tag{C.26}$$

$\square$

Although the dependence on $s$ is not relevant for our final statement, it is important to comment on the effect of the smoothness of $H(x)$. The result states that the dependence of the required parameters with respect to $1/\epsilon_1$ improves with the highest degree for which all mixed derivatives of $H(x)$ are bounded. When $H(x)$ is analytic, $s$ can be chosen to be very large so that the number of parameters in the neural network almost scales as $\mathcal{O}(\tilde{m}s^{\log(\tilde{m})})$. The constant scales rather poorly with $s$; however, this effect is only be visible for very small $\epsilon_1$.

Using Lemma 8, we can show that there exist parameters such that the complete model approximates $\text{tr}(O\rho(x))$ and obtains a small training objective (defined in Definition 7). The theorem (Theorem 6) in the main text corresponds to $\epsilon_1 = 0.2\epsilon, \epsilon_2 = \epsilon$ so that $(\epsilon_1 + \epsilon_2)^2 \leq 1.44\epsilon^2 \leq 0.53\epsilon \leq \epsilon$.

**Theorem 16** (Detailed restatement of Theorem 6). *For any $1/e > \epsilon_1, \epsilon_2 > 0$ and appropriate width $W$, there exist weights $\Theta', w'$ such that the neural network model $f^{\Theta',w'}$ satisfies*

$$|f^{\Theta',w'}(x) - \text{tr}(O\rho(x))| \leq \epsilon_1 \tag{C.27}$$

*for any $x \in [-1,1]^m$. In particular, for any collection of $N$ training data points $\{(x_\ell, y_\ell)\}_{\ell=1}^N$ with $|y_\ell - \text{tr}(O\rho(x_\ell))| \leq \epsilon_2$, we have*

$$\frac{1}{N} \sum_{\ell=1}^N |f^{\Theta',w'}(x_\ell) - y_\ell|^2 + \lambda \|w'\|_1 \leq (\epsilon_1 + \epsilon_2)^2 \tag{C.28}$$

*for a suitable choice of regularization parameter $\lambda = \mathcal{O}(\epsilon_1^2)$. Moreover, each parameter $\Theta_i$ of the network has a magnitude of $|\Theta_i| = 2^{\mathcal{O}(\text{polylog}(1/\epsilon_1))}$.*

*Proof.* Write $O = \sum_P \alpha_P \text{tr}(P\rho(x))$. By Theorem 8, let $D = \mathcal{O}(1)$ be a constant such that

$$\sum_P |\alpha_P| \leq D. \tag{C.29}$$

For every Pauli $P$, then by Lemma 8, there exist weights $\theta'_P$ such that a neural network $\bar{f}_P^{\theta'}$ : $[0,1]^{\tilde{m}} \to \mathbb{R}$ with two hidden layers as in Definition 6 approximates the local functions $f_P(\tau(\bar{x})) = \text{tr}(P\rho(\chi_P(\tau(\bar{x}))))$, where $\tau(\bar{x}) = 2\bar{x} - 1$, up to an error $\epsilon_1/(4D)$, when the width of their hidden layers is chosen as $W = \epsilon_1^{-\frac{\tilde{m}+1}{s}} 2^{\mathcal{O}(\tilde{m}\log(\tilde{m}))}$, where the number of local coordinates is given by $\tilde{m} = |I_P|$ and by the smoothness assumption Item (d), $s \geq 1$. In other words, we have

$$\left| \bar{f}_P^{\theta'_P}(\bar{x}) - \text{tr}(P\rho(\chi_P(\tau(\bar{x})))) \right| \leq \frac{\epsilon_1}{4D}, \tag{C.30}$$

where $\bar{x} \in [0,1]^{\tilde{m}}$. Because $\tau$ is simply a coordinate transformation, then we also obtain

$$\left| f_P^{\theta'_P}(x) - \text{tr}(P\rho(\chi_P(x))) \right| \leq \frac{\epsilon_1}{4D}, \tag{C.31}$$

for $x \in [-1,1]^{\tilde{m}}$.

Furthermore, by Lemma 2 in Appendix A.1.2, the sum of the local functions $f(x) = \sum_P \alpha_P f_P(x)$ approximates the ground state property $\text{tr}(O\rho(x)) = \sum_P \alpha_P \text{tr}(P\rho(x))$ well. In particular, combining Lemma 2 with Theorem 8, we have

$$\left| \sum_P \alpha_P \text{tr}(P\rho(\chi_P(x))) - \sum_P \alpha_P \text{tr}(P\rho(x)) \right| \leq \frac{\epsilon_1}{4}. \tag{C.32}$$

This holds when choosing the local radius $\delta_1$ (defined in Equation (A.5)) to be $\delta_1 = 4C \log^2(1/\epsilon_1)$ for some constant $C$. This implies that $\tilde{m} = |I_P| = \mathcal{O}(\text{polylog}(1/\epsilon_1))$ (e.g., Equation (S33) of [2]). Thus, for the model $f^{\Theta',w'}$ with architecture defined in Definition 6 and weights $w'_P = \alpha_P$ and $\Theta' = \{\theta'_P\}_P$, we have

$$|f^{\Theta',w'}(x) - \text{tr}(O\rho(x))| \tag{C.33}$$

$$= \left| \sum_P \alpha_P f_P^{\theta'_P}(x) - \sum_P \text{tr}(P\rho(\chi_P(x))) + \sum_P \text{tr}(P\rho(\chi_P(x))) - \sum_P \text{tr}(P\rho(x)) \right| \tag{C.34}$$

$$\leq \sum_P |\alpha_P| \frac{\epsilon_1}{4D} + \left| \sum_P \text{tr}(P\rho(\chi_P(x))) - \sum_P \text{tr}(P\rho(x)) \right| \tag{C.35}$$

$$\leq \frac{\epsilon_1}{4} + \left| \sum_P \text{tr}(P\rho(\chi_P(x))) - \sum_P \text{tr}(P\rho(x)) \right| \tag{C.36}$$

$$\leq \frac{\epsilon_1}{2}. \tag{C.37}$$

Moreover, by definition of the training data, we have $|y_\ell - \text{tr}(O\rho(x_\ell))| \leq \epsilon_2$. Thus, by triangle inequality and choosing regularization parameter $\lambda = \epsilon_1^2/(2D)$, we have

$$\frac{1}{N} \sum_{\ell=1}^N |f^{\Theta',w'}(x_\ell) - y_\ell|^2 + \lambda \|w'\|_1 \tag{C.38}$$

$$= \frac{1}{N} \sum_{\ell=1}^N |f^{\Theta',w'}(x_\ell) - \text{tr}(O\rho(x_\ell)) + \text{tr}(O\rho(x_\ell)) - y_\ell|^2 + \lambda \|w'\|_1 \tag{C.39}$$

$$\leq (\epsilon_1/2 + \epsilon_2)^2 + \lambda \|w'\|_1 \tag{C.40}$$

$$\leq \left( \frac{\epsilon_1}{2} + \epsilon_2 \right)^2 + \frac{\epsilon_1^2}{2} \tag{C.41}$$

$$\leq (\epsilon_1 + \epsilon_2)^2. \tag{C.42}$$

Finally, plugging in $\tilde{m} = \mathcal{O}(\text{polylog}(1/\epsilon_1))$, by Lemma 8, then we obtain $|\Theta_i| = 2^{\mathcal{O}(\text{polylog}(1/\epsilon_1))}$, as required. $\qquad\square$

## C.2 Prediction error bound

In this section, we derive our result on the prediction error to complete the proof of Theorem 14. The central result we use is the Koksma-Hlawka inequality (Theorem 10) from quasi-Monte Carlo theory, which produces a bound in terms of the star-discrepancy (Definition 2) and the Hardy-Krause variation (Equation (A.36)). We review these tools in Appendix A.2. To bound the star-discrepancy, we consider a specific low-discrepancy sequence with guarantees described in Appendix A.2. The Hardy-Krause variation can be bounded by considering the mixed derivatives of our target function $\text{tr}(O\rho(x))$ and our neural network model. We relegate the bounds on the mixed derivatives of $\text{tr}(O\rho(x))$ to Appendix C.4, as the discussion is fairly technical. To bound the mixed derivatives of our model, we consider the following lemma.

**Lemma 9** (Bound on mixed derivatives of neural network). *Let $k, d \in \mathbb{N}$. Let $\hat{f} : [-1, 1]^d \to \mathbb{R}$ be a tanh neural network with two hidden layers of width $W \geq d$ and maximal weight $W_{\max}$. Then*

$$\|\hat{f}\|_{W^{k,\infty}([-1,1]^d)} = 2^{\mathcal{O}(k^2 \log(k) + k \log(dWW_{\max}))}. \tag{C.43}$$

*Proof.* Recall that a tanh deep neural network with two hidden layers is defined as a function $\hat{f} : [-1, 1]^d \to \mathbb{R}$ such that

$$\hat{f}(x) = (W_{\text{out}} \circ \tanh \circ W_{\text{hidden}} \circ \tanh \circ W_{\text{in}})(x), \tag{C.44}$$

where the activation function $\tanh$ is applied element-wise. Note that this result holds for any $\tanh$ neural network with two hidden layers, where this neural network does not necessarily have to be the same model as Definition 6.

Let $W_L \in \{W_{\text{in}}, W_{\text{hidden}}, W_{\text{out}}\}$ denote the layers of the neural network that perform an affine transformation for $L \in \{\text{in}, \text{hidden}, \text{out}\}$. We can also use $L \in \{0, 1, 2\}$, where 0 corresponds to in, 1 corresponds to hidden, and 2 corresponds to out. Let $d_L$ denote the width (number of input neurons) in each layer, where we define $d_0 = d_{\text{in}}, d_1 = d_2 = W, d_3 = d_{\text{out}}$. In this way, $W_L : \mathbb{R}^{d_L} \to \mathbb{R}^{d_{L+1}}$ for $L \in \{0, 1, 2\}$. Explicitly, we have $W_{\text{in}} : \mathbb{R}^{d_{\text{in}}} \to \mathbb{R}^W, W_{\text{hidden}} : \mathbb{R}^W \to \mathbb{R}^W, W_{\text{out}} : \mathbb{R}^W \to \mathbb{R}^{d_{\text{out}}}$.

Since $W_L$ performs an affine transformation, we can write it has $W_L(x) = (f_1(x), \ldots, f_{d_{L+1}}(x))$, where $x \in \mathbb{R}^{d_L}$ and $f_i$ are linear functions $f_i(x) = w_i^\mathsf{T} x + b_i$ for $w_i \in \mathbb{R}^{d_L}, b_i \in \mathbb{R}$. For these linear

layers, we observe for any function $g : \mathbb{R}^{d_g} \to \mathbb{R}^{d_L}$ with input dimension $d_g$ and for $L \in \{0, 1, 2\}$, we have

$$\max_{1 \leq i \leq d_{L+1}} \|(W_L \circ g)_i\|_{W^{k,\infty}([-1,1]^d)} = \max_{1 \leq i \leq d_{L+1}} \|f_i(g(x))\|_{W^{k,\infty}([-1,1]^d)} \tag{C.45}$$

$$\leq \sum_{i=1}^{d_{L+1}} \|w_i^T g(x) + b_i\|_{W^{k,\infty}([-1,1]^d)} \tag{C.46}$$

$$\leq \max_{1 \leq j \leq d_g} \|W_L\|_1 \|g(x)_j\|_{W^{k,\infty}([-1,1]^d)}, \tag{C.47}$$

where we use the notation

$$\|W_L\|_1 \triangleq \sum_{i=1}^{d_{L+1}} \left( |b_i| + \sum_{j=1}^{d_L} |w_{i,j}| \right), \tag{C.48}$$

where $w$ is a matrix with rows given by the vectors $w_i^T$, $w_i \in \mathbb{R}^{d_L}$. To show this inequality, we used Hölder's inequality in the last step. With this, by factoring out one layer $W_L$ at a time, we can bound the Sobolev norm of $\hat{f}$. In particular, we have

$$\|\hat{f}\|_{W^{k,\infty}([-1,1]^d)} \tag{C.49}$$
$$= \|(W_{\text{out}} \circ \tanh \circ W_{\text{hidden}} \circ \tanh \circ W_{\text{in}})(x)\|_{W^{k,\infty}([-1,1]^d)} \tag{C.50}$$
$$\leq \|W_{\text{out}}\|_1 \max_{1 \leq j \leq W} \|(\tanh \circ W_{\text{hidden}} \circ \tanh \circ W_{\text{in}})_j\|_{W^{k,\infty}([-1,1]^d)} \tag{C.51}$$
$$\leq \|W_{\text{out}}\|_1 16(e^2 k^4 d^2)^k (2k)^{k(k+1)} \max_{1 \leq j \leq W} \|(W_{\text{hidden}} \circ \tanh \circ W_{\text{in}})_j\|_{W^{k,\infty}([-1,1]^d)}^k \tag{C.52}$$
$$\leq \|W_{\text{out}}\|_1 \|W_{\text{hidden}}\|_1^k \cdot 16(e^2 k^4 d^2)^k (2k)^{k(k+1)} \max_{1 \leq j \leq W} \|(\tanh \circ W_{\text{in}})_j\|_{W^{k,\infty}([-1,1]^d)}^k \tag{C.53}$$
$$\leq \|W_{\text{out}}\|_1 \|W_{\text{hidden}}\|_1^k \cdot 16^2 (e^2 k^4 d^2)^{2k} (2k)^{2k(k+1)} \|W_{\text{in}}\|_1^k. \tag{C.54}$$

In the second line, we used Equation (C.47). In the third line, we used the two following inequalities:

$$\left| \frac{d^m}{dx^m} \tanh(x) \right| \leq (2m)^{m+1} \min\{\exp(-2x), \exp(2x)\} \tag{C.55}$$

for all $x \in \mathbb{R}, m \in \mathbb{N}$ (see Lemma A.4 in [53]), and

$$\|g \circ f\|_{W^{n,\infty}} \leq 16(e^2 n^4 m d^2)^n \|g\|_{W^{n,\infty}} \max_{1 \leq i \leq m} \|(f)_i\|_{W^{n,\infty}}^n \tag{C.56}$$

for any functions $f \in C^n(\Omega_1; \Omega_2)$ and $g \in C^n(\Omega_2; \mathbb{R})$ defined on $\Omega_1 \subset \mathbb{R}^d$, $\Omega_2 \subset \mathbb{R}^m$ with $d, m, n \in \mathbb{N}$ (see Lemma A.7 in [53]). In the fourth and fifth lines, we used Equation (C.47) and these inequalities again. Furthermore, we used that our inputs are absolutely bounded by 1 in the last step.

We can further bound this term using that $W_{\text{max}}$ is the maximal weight of $\hat{f}$ and the width of the hidden layers is lower bounded by $W \geq d$.

$$\|\hat{f}\|_{W^{k,\infty}([-1,1]^d)} \leq 16^2 (e^2 k^4 d^2)^{2k} (2k)^{2k(k+1)} W_{\text{max}}^{2k+1} W^{3k+1} d^k = 2^{\mathcal{O}(k(k \log(k) + \log(dW W_{\text{max}})))}. \tag{C.57}$$

$\square$

Now we have all the necessary tools in order to derive a bound on the generalization error for our neural network model of the form given in Definition 6. In our proof, we first bound the prediction error in terms of functions with $2\tilde{m}$-dimensional domain and on which we can directly apply the Koksma-Hlawka inequality. Then, we use the previous result to obtain an explicit bound. Due to the regularity of the parameters $\alpha_P$ and our model parameters $w_P$, this prediction error bound is independent of the system size $n$.

Before stating the formal result bounding the prediction error, we introduce some notation. We define the prediction error of a neural network $f^{\Theta, w}$ with weights given by $\Theta, w$ as

$$R(\Theta) \triangleq \mathop{\mathbb{E}}_{x \sim U[-1,1]^m} |f^{\Theta, w}(x) - \text{tr}(O\rho(x))|^2, \tag{C.58}$$

where in our case, $x \sim U[-1,1]^m$ denotes $x$ sampled from a uniform distribution over $[-1,1]^m$. We suppress $w$ in the notation to avoid cluttering. For a neural network $f^{\Theta,w}$ generated from training on some data $\{(x_\ell, y_\ell)\}_{\ell=1}^N$, we can define the training error as

$$\hat{R}(\Theta) \triangleq \frac{1}{N} \sum_{\ell=1}^N |f^{\Theta,w}(x_\ell) - y_\ell|^2. \tag{C.59}$$

Moreover, as in our analysis in Appendix C.1, we rely on an approximation of the ground state property $\mathrm{tr}(O\rho(x))$ by a sum of smooth local functions $\sum_P \alpha_P f_P(x)$ (Lemma 2). Namely, combining Lemma 2 and Theorem 8, we have that for $\epsilon_1 > 0$, then choosing $\delta_1 > 0$ as in Equation (A.5), i.e., $\delta_1 = \mathcal{O}(\log^2(1/\epsilon_1))$,

$$\left| \sum_P \alpha_P f_P(x) - \mathrm{tr}(O\rho(x)) \right| \leq \frac{\epsilon_1}{2} \tag{C.60}$$

Note that here, again, we slightly alter the definition from Appendix A.1.2, where we do not include the coefficient $\alpha_P$ in the definition of $f_P(x)$. With these definitions, we have the following guarantee on the prediction error.

**Lemma 10** (Prediction error bound). *Let $1/e > \epsilon_1, \epsilon_2 > 0$. Consider a tanh neural network $f^{\Theta,w} : [-1,1]^m \to \mathbb{R}$ with architecture defined in Definition 7 with weights $\Theta_i \leq W_{\max}$ for some $W_{\max} > 0$ independent of the system size $n$ and weights $w$ in the last layer. Suppose we train $f^{\Theta,w}$ on data $\{(x_\ell, y_\ell)\}_{\ell=1}^N$ of size $N$, where the $x_\ell$'s form a low-discrepancy sequence with star-discrepancy $D_N^*$ and $|y_\ell - \mathrm{tr}(O\rho(x_\ell))| \leq \epsilon_2$. Then, we have*

$$R(\Theta) \leq \hat{R}(\Theta) + \frac{\epsilon_1^2}{2} + \epsilon_2^2 + (\|w\|_1 + \|w\|_1^2) D_N^* \cdot 2^{\mathcal{O}(\mathrm{polylog}(WW_{\max}/\epsilon_1))}. \tag{C.61}$$

*where $\tilde{m} = |I_P| = \mathcal{O}(\log^2(1/\epsilon_1))$ for $I_P$ defined in Equation (A.2).*

*Proof.* Recall the definition of our neural network model in Definition 6. In particular, our model is given by a function $f^{\Theta,w} : [-1,1]^m \to \mathbb{R}$ defined by

$$f^{\Theta,w}(x) = \sum_{P \in S^{(\mathrm{geo})}} w_P f_P^{\theta_P}(x), \tag{C.62}$$

where we refer to $f_P^{\theta_P} : [-1,1]^{\tilde{m}} \to \mathbb{R}$ as the local models. For $f_P(x) = \mathrm{tr}(P\rho(\chi_P(x)))$ as considered in Equation (C.60), we can define the following quantities. Define the training error with respect to this local approximation by

$$\hat{R}_{\mathrm{loc}}(\Theta) \triangleq \frac{1}{N} \sum_{\ell=1}^N \left| f^{\Theta,w}(x_\ell) - \sum_P \alpha_P f_P(x_\ell) \right|^2 \tag{C.63}$$

Also define the prediction error with respect to the local approximation as

$$R_{\mathrm{loc}}(\Theta) \triangleq \mathop{\mathbb{E}}_{x \sim U[-1,1]^m} \left| f^{\Theta,w}(x) - \sum_P \alpha_P f_P(x) \right|^2, \tag{C.64}$$

where $x \sim U[-1,1]^m$ means that $x$ is sampled according to the uniform distribution.

By Lemma 2, for our choice of $\delta_1$, we have Equation (C.60):

$$\left| \sum_P \alpha_P f_P(x) - \mathrm{tr}(O\rho(x)) \right| \leq \frac{\epsilon_1}{2}. \tag{C.65}$$

By the triangle inequality, we can bound the prediction error as

$$R(\Theta) = \mathop{\mathbb{E}}_{x \sim U[-1,1]^m} \left| f^{\Theta,w}(x) - \sum_P \alpha_P f_P(x) + \sum_P \alpha_P f_P(x) - \mathrm{tr}(O\rho(x)) \right|^2 \leq R_{\mathrm{loc}}(\Theta) + \frac{\epsilon_1^2}{4}. \tag{C.66}$$

By applying the reverse triangle inequality, we can further bound this as

$$R(\Theta) \leq \frac{\epsilon_1^2}{4} + \hat{R}_{\text{loc}}(\Theta) + |R_{\text{loc}}(\Theta) - \hat{R}_{\text{loc}}(\Theta)| \tag{C.67}$$

$$= \frac{\epsilon_1^2}{4} + \hat{R}_{\text{loc}}(\Theta) \tag{C.68}$$

$$+ \left| \underset{x \sim U[-1,1]^m}{\mathbb{E}} \left| f^{\Theta,w}(x) - \sum_P \alpha_P f_P(x) \right|^2 - \frac{1}{N} \sum_{\ell=1}^{N} \left| f^{\Theta,w}(x_\ell) - \alpha_P f_P(x_\ell) \right|^2 \right| \tag{C.69}$$

$$= \frac{\epsilon_1^2}{4} + \hat{R}_{\text{loc}}(\Theta) \tag{C.70}$$

$$+ \left| \underset{x \sim U[-1,1]^m}{\mathbb{E}} \left( \sum_P w_P f_P^{\theta_P}(x) - \alpha_P f_P(x) \right)^2 - \frac{1}{N} \sum_{\ell=1}^{N} \left( \sum_P w_P f_P^{\theta_P}(x_\ell) - \alpha_P f_P(x_\ell) \right)^2 \right| \tag{C.71}$$

We can expand the term in the expectation/sum as follows

$$\left( \sum_P w_P f_P^{\theta_P}(x) - \alpha_P f_P(x) \right)^2 \tag{C.72}$$

$$= \left( \sum_P w_P f_P^{\theta_P}(x) \right)^2 - 2 \left( \sum_P w_P f_P^{\theta_P}(x) \right) \left( \sum_P \alpha_P f_P(x) \right) + \left( \sum_P \alpha_P f_P(x) \right)^2 \tag{C.73}$$

$$= \sum_{P_1,P_2} w_{P_1} f_{P_1}^{\theta_{P_1}}(x) w_{P_2} f_{P_2}^{\theta_{P_2}}(x) - 2 w_{P_1} f_{P_1}^{\theta_{P_1}}(x) \alpha_{P_2} f_{P_2}(x) + \alpha_{P_1} f_{P_1}(x) \alpha_{P_2} f_{P_2}(x). \tag{C.74}$$

Plugging this into the absolute value term in Equation (C.71) and upper bounding it with the triangle inequality, we have

$$|R_{\text{loc}}(\Theta) - \hat{R}_{\text{loc}}(\Theta)| \leq \sum_{P_1,P_2} |w_{P_1}||w_{P_2}| \left| \underset{x \sim U[-1,1]^m}{\mathbb{E}} [f_{P_1}^{\theta_{P_1}}(x) f_{P_2}^{\theta_{P_2}}(x)] - \frac{1}{N} \sum_{\ell=1}^{N} f_{P_1}^{\theta_{P_1}}(x_\ell) f_{P_2}^{\theta_{P_2}}(x_\ell) \right|$$

$$+ 2|w_{P_1}||\alpha_{P_2}| \left| \underset{x \sim U[-1,1]^m}{\mathbb{E}} [f_{P_1}^{\theta_{P_1}}(x) f_{P_2}(x)] - \frac{1}{N} \sum_{\ell=1}^{N} f_{P_1}^{\theta_{P_1}}(x_\ell) f_{P_2}(x_\ell) \right|$$

$$+ |\alpha_{P_1}||\alpha_{P_2}| \left| \underset{x \sim U[-1,1]^m}{\mathbb{E}} [f_{P_1}(x) f_{P_2}(x)] - \frac{1}{N} \sum_{\ell=1}^{N} f_{P_1}(x_\ell) f_{P_2}(x_\ell) \right|. \tag{C.75}$$

Notice that in the expectation over $[-1,1]^m$, we can replace this by an expectation over the set of local parameters, i.e., the parameters with coordinates in $I_{P_1} \cup I_{P_2}$, which we denote by $S_{P_1,P_2}$. This is because the functions in the expectations are local functions that only depend on these local parameters. The dimension of the domain we integrate over thus becomes independent of the system size $n$, as $|S_{P_1,P_2}| \leq 2\tilde{m} = 2|I_P|$.

We can now bound this term further using the Koksma-Hlawka inequality (Theorem 10). We apply a simple variable transformation $\tau(x) = 2x - 1$ so that the domain of $f_P \circ \tau$ becomes $[0,1]^{\tilde{m}}$. Furthermore, we denote the domain associated with $S_{P_1,P_2}$ by $\Omega_{P_1,P_2} \triangleq [0,1]^{|S_{P_1,P_2}|}$. Starting with the first term in Equation (C.75), we obtain

$$\left| \underset{x \sim U[0,1]^m}{\mathbb{E}} [f_{P_1}^{\theta_{P_1}}(\tau(\bar{x})) f_{P_2}^{\theta_{P_2}}(\tau(\bar{x}))] - \frac{1}{N} \sum_{\ell=1}^{N} f_{P_1}^{\theta_{P_1}}(\tau(\bar{x}_\ell)) f_{P_2}^{\theta_{P_2}}(\tau(\bar{x}_\ell)) \right| \tag{C.76}$$

$$= \left| \underset{x \sim U(\Omega_{P_1,P_2})}{\mathbb{E}} [f_{P_1}^{\theta_{P_1}}(\tau(\bar{x})) f_{P_2}^{\theta_{P_2}}(\tau(\bar{x}))] - \frac{1}{N} \sum_{\ell=1}^{N} f_{P_1}^{\theta_{P_1}}(\tau(\bar{x}_\ell)) f_{P_2}^{\theta_{P_2}}(\tau(\bar{x}_\ell)) \right| \tag{C.77}$$

$$= \left| \int_{S_{P_1,P_2}} f_{P_1}^{\theta_{P_1}}(\tau(\bar{x})) f_{P_2}^{\theta_{P_2}}(\tau(\bar{x})) dx - \frac{1}{N} \sum_{\ell=1}^{N} f_{P_1}^{\theta_{P_1}}(\tau(\bar{x}_\ell)) f_{P_2}^{\theta_{P_2}}(\tau(\bar{x}_\ell)) \right| \tag{C.78}$$

$$\leq D_N^*(2\tilde{m}) V_{HK}\left( (f_{P_1}^{\theta_{P_1}} \cdot f_{P_2}^{\theta_{P_2}}) \circ \tau \right), \tag{C.79}$$

where $\bar{x}_\ell = \tau^{-1}(x_\ell)$, such that Equation (C.76) matches the expression referenced in Equation (C.75). Note that we also applied in the last step that the star-discrepancy is increasing with respect to the dimension of the sequence. By application of the chain rule and the Cauchy-Schwartz inequality in the definition of the Hardy-Krause variation (Equation (A.36)), it is easy to see that

$$V_{HK}\left( (f_{P_1}^{\theta_{P_1}} \cdot f_{P_2}^{\theta_{P_2}}) \circ \tau \right) \leq 2^{2\tilde{m}} V_{HK}(f_{P_1}^{\theta_{P_1}} \cdot f_{P_2}^{\theta_{P_2}}). \tag{C.80}$$

For all subsets $S' \subseteq S_{P_1,P_2}$, applying the product rule yields

$$\left| \frac{\partial^{|S'|}}{\partial x_{S'}} (f_{P_1}^{\theta_{P_1}} \cdot f_{P_2}^{\theta_{P_2}}) \right| \leq \sum_{A \subseteq S'} \left| \frac{\partial^{|A|}}{\partial x_A} f_{P_1}^{\theta_{P_1}} \right| \left| \frac{\partial^{|S'\setminus A|}}{\partial x_{S'\setminus A}} f_{P_2}^{\theta_{P_2}} \right| = 2^{\mathcal{O}(\tilde{m} \log(WW_{\max}) + \tilde{m}^2 \log(\tilde{m}))}, \tag{C.81}$$

where the last equality follows from applying Lemma 9 from Appendix C.4 with $d = k = 2\tilde{m}$ and $|\{A : A \subseteq S'\}| = 2^{2\tilde{m}}$. Here, we are using the notation $\frac{\partial^{|B|}}{\partial x_B}$ to denote the mixed derivative with respect to all parameters $x_i \in B$ for some set $B$. Thus, applying Lemma 9 again, we obtain

$$V_{HK}(f_{P_1}^{\theta_{P_1}} \cdot f_{P_2}^{\theta_{P_2}}) \leq \sum_{S' \subseteq S_{P_1,P_2}} \left| \frac{\partial^{|S'|}}{\partial x_{S'}} (f_{P_1}^{\theta_{P_1}} \cdot f_{P_2}^{\theta_{P_2}}) \right| = 2^{\mathcal{O}(\tilde{m} \log(WW_{\max}) + \tilde{m}^2 \log(\tilde{m}))}. \tag{C.82}$$

Thus, putting it all together, we see that

$$\left| \mathbb{E}_{x \sim U[-1,1]^m} [f_{P_1}^{\theta_{P_1}}(x) f_{P_2}^{\theta_{P_2}}(x)] - \frac{1}{N} \sum_{\ell=1}^{N} f_{P_1}^{\theta_{P_1}}(x_\ell) f_{P_2}^{\theta_{P_2}}(x_\ell) \right| \tag{C.83}$$

$$\leq 2^{2\tilde{m}} D_N^*(2\tilde{m}) 2^{\mathcal{O}(\tilde{m} \log(WW_{\max}) + \tilde{m}^2 \log(\tilde{m}))}. \tag{C.84}$$

The remaining terms in Equation (C.75) can be bounded similarly using Lemma 22 from Appendix C.4. This lemma is applicable to $f_P$ because the derivatives are with respect to the local parameters. In this way, we can upper bound Equation (C.75) by

$$|R_{\mathrm{loc}}(\Theta) - \hat{R}_{\mathrm{loc}}(\Theta)| \leq \sum_{P_1,P_2} \Big( (|w_{P_1}||w_{P_2}| + |w_{P_1}||\alpha_{P_2}|) 2^{\mathcal{O}(\tilde{m} \log(WW_{\max}) + \tilde{m}^2 \log(\tilde{m}))} \tag{C.85}$$

$$+ |\alpha_{P_1}||\alpha_{P_2}| 2^{\mathcal{O}(\tilde{m} \log(\tilde{m}))} \Big) D_N^*(2\tilde{m}). \tag{C.86}$$

Plugging this back in to Equation (C.67), we have

$$R(\Theta) \leq \frac{\epsilon_1^2}{4} + \hat{R}_{\mathrm{loc}}(\Theta) + \sum_{P_1,P_2} \Big( (|w_{P_1}||w_{P_2}| + |w_{P_1}||\alpha_{P_2}|) 2^{\mathcal{O}(\tilde{m} \log(WW_{\max}) + \tilde{m}^2 \log(\tilde{m}))} \tag{C.87}$$

$$+ |\alpha_{P_1}||\alpha_{P_2}| 2^{\mathcal{O}(\tilde{m} \log(\tilde{m}))} \Big) D_N^*(2\tilde{m}). \tag{C.88}$$

Lastly, we can bound $\hat{R}_{\mathrm{loc}}(\Theta) \leq \epsilon_1^2/4 + \epsilon_2^2 + \hat{R}(\Theta)$ in the same way as in Equation (C.66):

$$\hat{R}_{\mathrm{loc}}(\Theta) = \frac{1}{N} \sum_{\ell=1}^{N} \left| f^{\Theta,w}(x_\ell) - \sum_P \alpha_P f_P(\chi_P(x_\ell)) \right|^2 \tag{C.89}$$

$$\leq \frac{1}{N} \sum_{\ell=1}^{N} |f^{\Theta,w}(x_\ell) - y_\ell|^2 + |y_\ell - \mathrm{tr}(O\rho(x_\ell))|^2 + \left| \mathrm{tr}(O\rho(x_\ell)) - \sum_P \alpha_P f_P(\chi_P(x_\ell)) \right|^2 \tag{C.90}$$

$$\leq \hat{R}(\Theta) + \epsilon_2^2 + \frac{\epsilon_1^2}{4}. \tag{C.91}$$

Inserting $\tilde{m} = |I_P| = \mathcal{O}\left(\text{polylog}\left(1/\epsilon_1\right)\right)$ and using that $\sum_P |\alpha_P| = \mathcal{O}(1)$ (Theorem 8) yields

$$R(\Theta) \leq \hat{R}(\Theta) + \frac{\epsilon_1^2}{2} + \epsilon_2^2 + (\|w\|_1 + \|w\|_1^2)D_N^*(2\tilde{m})2^{\mathcal{O}(\text{polylog}(WW_{\max}/\epsilon_1))}. \tag{C.92}$$

$\square$

Using the previous result and the results from low-discrepancy theory (see Appendix A.2 for a review), we can now show that Algorithm 1 will, under mild assumptions for training, output a model, which yields low prediction error. Thus, using Lemma 10, we can easily prove Theorem 14.

*Proof of Theorem 14.* By Theorem 9, we know that for Sobol sequences in base 2 with points in $[0,1]^d$, the star-discrepancy is bounded by

$$D_N^*(d) \leq C(d)\frac{\log(N)^d}{N}, \tag{C.93}$$

where $C(d)$ is a constant such that

$$C(d) < \frac{1}{d!}\left(\frac{d}{\log(2d)}\right). \tag{C.94}$$

Since $C(d) = o(1)$, there exists a constant $C$, such that $C \geq C(d)$ for all $d > 0$. In our case, $d = 2\tilde{m}$, so we have

$$D_N^*(2\tilde{m}) \leq C\frac{\log(N)^{2\tilde{m}}}{N}. \tag{C.95}$$

Using the assumption that the training objective is not larger than $((\epsilon_1 + \epsilon_2)^2 + \epsilon_3)/2$, by Lemma 10, we have

$$R(\Theta^*) = \mathop{\mathbb{E}}_{x \sim U[-1,1]^m} |f^{\Theta^*,w^*}(x) - \text{tr}(O\rho(x))|^2 \tag{C.96}$$

$$\leq \frac{\epsilon_1^2}{2} + \epsilon_2^2 + \frac{(\epsilon_1 + \epsilon_2)^2 + \epsilon_3}{2} + C'\frac{\log(N)^{\text{polylog}(1/\epsilon_1)}2^{\mathcal{O}(\text{polylog}(W_{\max}/\epsilon_1))}}{N} \tag{C.97}$$

$$\leq 2(\epsilon_1 + \epsilon_2)^2 + \frac{\epsilon_3}{2} + C'\frac{\log(N)^{\text{polylog}(1/\epsilon_1)}2^{\mathcal{O}(\text{polylog}(W_{\max}/\epsilon_1))}}{N}, \tag{C.98}$$

where $C'$ is a constant. We also used here that $\tilde{m} = |I_P| = \mathcal{O}(\text{polylog}(1/\epsilon_1))$. Since the training data has size $N = \mathcal{O}\left(2^{\text{polylog}(1/\epsilon_1)+\text{polylog}(1/\epsilon_3)}\right)$, $W_{\max}$ can be chosen with respect to $\epsilon_1, \epsilon_3$ and independent of the system size $n$ such that

$$C'\frac{\log(N)^{\text{polylog}(1/\epsilon_1)}2^{\mathcal{O}(\text{polylog}(W_{\max}/\epsilon_1))}}{N} \leq \frac{\epsilon_3}{4}. \tag{C.99}$$

In this way, we obtain

$$R(\Theta^*) \leq 2(\epsilon_1 + \epsilon_2)^2 + \epsilon_3. \tag{C.100}$$

$\square$

Since the training objective from Definition 7 is non-convex, we cannot guarantee that our algorithm converges to a neural network with low training error. However, the assumptions made in Theorem 14 are rather mild in practice. Small training errors are a well-known phenomenon in deep learning and usually come at the expense of a larger prediction error, which is referred to as *overfitting*. Overfitting may arise due to excessive model complexity [89], i.e. too many trainable parameters. This is reflected by *Lemma* 10, since the generalization error increases with the width $W$ of the layers. The major challenge in practice lies in finding an appropriate balance between achieving a small training objective and model complexity, rather than only the latter. Furthermore, when the inputs are regularized, the weights usually remain small during training when initialized properly. This was for example observed in [53].

Finally, it is worth noting that in a scenario with a constant number of parameters $m = \mathcal{O}(1)$, similar to the setup in [51], the expression derived from the outcome in Lemma 10 exhibits nearly linear dependence on $\epsilon$. When incorporating the constant number of parameters by setting $\tilde{m} = m$, we

recover the exact ground state properties $\text{tr}(P\rho(x))$ in $f_P$. Thus, $\epsilon_1$ in Lemma 10 becomes 0. Hence, the ability of LDS training to overcome the curse of dimensionality can unfold its full potential, since the domain dimension becomes independent of $\epsilon$ and expression Equation (C.88) reduces to a constant multiplied by $D_N^*(2m)$. By Equation (C.95), we obtain $R(\Theta) = \mathcal{O}(\epsilon^{-(1+\delta)})$ for any $\delta > 0$ and $\epsilon$ small enough, when the conditions of Theorem 5 are fulfilled.

## C.3   Prediction on general distributions

In this section, we generalize our results to hold for a wider class of distributions. Recall that our rigorous guarantee proven so far (Theorem 14) holds when the training data is generated according to a low-discrepancy sequence and the prediction error is measured with respect to the uniform distribution. We want to extend this result for different choices of both training and prediction error distributions. Notice that our prediction error bound (Lemma 10) is the only place that requires these assumptions on the distributions. Thus, in this section, we establish bounds on the expected prediction error for a more general family of distributions. We consider the following two cases.

1. The training data is generated according to a general low-discrepancy sequence (in the sense of Definitions 4 and 5), and the prediction error is measured with respect to some distribution $\mathcal{D}$.

2. The training data consists of independently and identically distributed (i.i.d.) random samples according to a distribution $\mathcal{D}$, and the prediction error is measured with respect to the same distribution $\mathcal{D}$.

There are some conditions on the distributions that we discuss shortly. In Case 1, suppose for example that we want to provide rigorous guarantees on the prediction error when the parameters $x \in [-1,1]^m$ are sampled from a standard normal distribution (restricted to $[-1,1]^m$ and normalized appropriately). As normally distributed test samples are more densely populated around the mean and more sparse around the boundary of the input domain, we need to predict more accurately around the mean than close to the boundary. When using a uniform low-discrepancy sequence for training, as in Algorithm 1, the predictive capabilities of our model are not exploited properly. To remedy this, we consider the training data to form a general low-discrepancy sequence, where it is low-discrepancy with respect to a normal distribution. We can relate this general low-discrepancy sequence to an LDS with respect to the *Lebesgue measure*, which are the sequences considered in Appendix C.2, via the probability integral transform (see, e.g., [90]). We sometimes refer to LDS with respect to the Lebesgue measure as *uniform* low-discrepancy sequences. Formally, for any random variable $X$, which follows some probability distribution $P(X \geq x) \triangleq F_X(x)$, the random variable $Y = F_X(X)$ follows a uniform distribution. It turns out that the same transformation on LDS produces LDS with respect to other measures than the Lebesgue measure, as illustrated in Figure 3. Moreover, under some assumptions on the distribution, we can bound the discrepancy with respect to other measures in terms of the discrepancy with respect to the Lebesgue measure, which we know how to bound as in Appendix C.2.

In the following, we formalize this argument and adapt it to our problem setting. We refer the reader to Appendix A.2 to review the necessary concepts of generalized (star-)discrepancy, the Koksma-Hlawka inequality, and related results. Then, we demonstrate that a generalization of Lemma 10 and Theorem 14 can be achieved by incorporating these findings with slight adjustments to the proofs.

In Case 2, we consider training data sampled i.i.d. from some distribution $\mathcal{D}$ and prediction error measured with respect to the same distribution $\mathcal{D}$. To obtain a rigorous guarantee on the prediction error in this case, we leverage a probabilistic bound on the discrepancy of uniformly random points from [82]. Utilizing the previously established framework from Case 1, we can bound the discrepancy of points sampled from $\mathcal{D}$ in terms of the discrepancy of uniformly random points. This allows us to establish similar guarantees for Case 2.

Before proving each of these cases, we set up our probabilistic framework and define the Borel measure with respect to which our low-discrepancy sequence is defined. Let $g \triangleq \text{PDF}(\mathcal{D})$ be the probability density function (PDF) of the data distribution and let $G \triangleq \text{CDF}(\mathcal{D})$ be the corresponding cumulative distribution function (CDF). In the following, assume that the PDF $g$ satisfies the following properties.

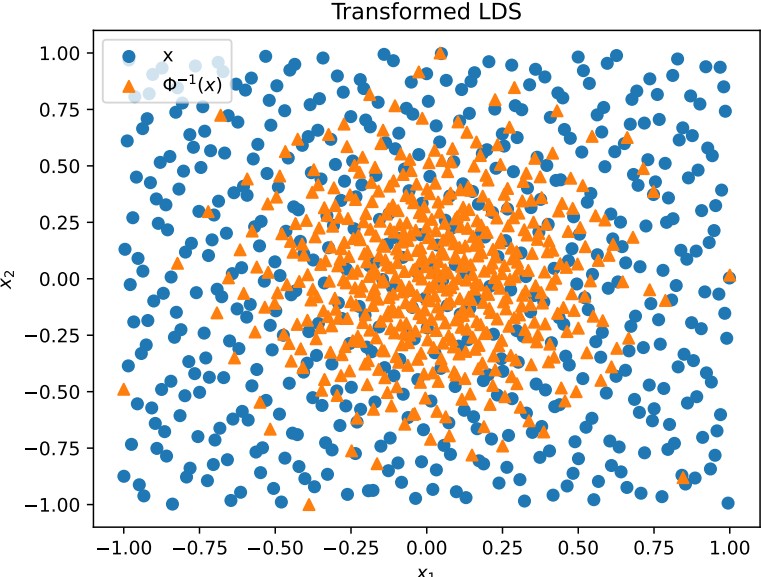

Figure 3: **Transformed low-discrepancy sequences.** The blue circles correspond to two-dimensional uniform Sobol points $x$. The orange triangles indicate the corresponding Sobol points with respect to the CDF of the standard normal distribution, denoted by $\Phi$. The latter forms a low-discrepancy-sequence with respect to the Borel measure $\mu = \Phi$.

(a) *Strict positivity:* $g$ has full support on $[-1, 1]^m$, i.e., $g(x) > 0$ if $x \in [-1, 1]^m$ and $g(x) = 0$ otherwise.

(b) *Continuity:* $g(x)$ is continuously differentiable on $[-1, 1]^m$.

(c) *Component-wise independence*: The (random) variables $\vec{x}_i, \vec{x}_j$ upon which different local terms $h_i(\vec{x}_i), h_j(\vec{x}_j)$ of the Hamiltonian depend on, are independent. Hence, the PDF $g$ is of the form

$$g(x) = \prod_{j=1}^{L} g_j(\vec{x}_j) \tag{C.101}$$

for PDFs $g_j$.

We implicitly assume that $g$ also satisfies all properties of a probability density function. It should be noted that Assumptions (a), (b) could technically be relaxed. We expand more on this later. Notice that if $g : [-1, 1]^m \to \mathbb{R}$ meets these requirements, the same holds for $\bar{g} \triangleq g \circ \tau : [0, 1]^m \to \mathbb{R}$. Here, we use the notation from the previous section, where a bar denotes that we are working in the domain $[0, 1]^m$ as opposed to $[-1, 1]^m$, and $\tau(\bar{(x)}) = 2\bar{x} - 1$. Since the available results hold on $[0, 1]^m$, we will mostly work with $\bar{g}$ and the corresponding CDF $\bar{G}$.

We continue to set up the necessary notation to formally state our prediction error bound for Case 1. Let $S_{P_1, P_2}, \Omega_{P_1, P_2}$ be as in the proof of Lemma 10. Namely, let $S_{P_1, P_2}$ be the parameters with coordinates in $I_{P_1} \cup I_{P_2}$, where $I_P$ is defined in Equation (A.2), and let $\Omega_{P_1, P_2} = [0, 1]^{|S_{P_1, P_2}|}$. Additionally, define $\mu_{P_1, P_2} \triangleq \prod_{j \in S_{P_1, P_2}} \bar{G}_j(\vec{x}_j)$ as the probability measure of the marginal for all (random) variables with indices in $S_{P_1, P_2}$. Due to Assumption (c), $\mu_{P_1, P_2}$ depends on at most $2\tilde{m}$ variables. Furthermore, we define

$$\mu^* \triangleq \underset{\mu_{P_i, P_j}}{\arg\max} \, D_N(|S_{P_i, P_j}|; \mu_{P_i, P_j}), \tag{C.102}$$

and denote by $S^*$ the corresponding coordinate set. $S^*$ forms the domain of $\mu^*$, and we use $d^*$ to denote the dimension of the domain.

In both Case 1 and Case 2, the idea is to define a transformation $F$ that maps random variables with an arbitrary distribution to uniformly random variables. Namely, we construct a mapping $\phi$ such that

$$\mathbb{E}_{x \sim \mathcal{D}}[u(x)] = \mathbb{E}_{x \sim U[-1,1]^m}[u(\phi(x))] \tag{C.103}$$

for any function $u$. In the following, we introduce the transform $F \triangleq \phi^{-1}$, as has been introduced in [91, 88, 92]. $F$ can nicely be characterized using $\bar{g}$ and $\bar{G}$, and assumptions on $F$ are easy to verify for a given data distribution. In fact, if $F$ satisfies a Lipschitz condition, then known results [88] bound the discrepancy with respect to an arbitrary measure in terms of the discrepancy with respect to the Lebesgue measure, i.e., we can directly upper-bound $D_N(d^*; \mu^*)$ in terms of $D_N(d^*)$. Our prediction error bound for more general distributions follows from this result and the results from Appendix C.2.

Let $g^*$ be defined such that $d\mu^*(x) = g^*(x)dx$. Also, let $A, B \subseteq S^*$ be such that $A \cap B = \emptyset$ and $C = S^* \setminus (A \cup B)$. Then, we define the conditional marginal PDF as

$$g^*(x_A | X_B = x_B) \triangleq \frac{\int_{[0,1]^{|C|}} g^*(x) dx_C}{\int_{[0,1]^{|A|+|C|}} \int_0^{(x_B)_1} \cdots \int_0^{(x_B)_{|B|}} g^*(x) dx} \tag{C.104}$$

and the corresponding CDF as

$$G^*(X_A = x_A | X_B = x_B) \triangleq \int_0^{(x_A)_1} \cdots \int_0^{(x_A)_{|A|}} g^*(x_A | x_B) dx_A. \tag{C.105}$$

For convenience, we refer to the indices of $x$ in $S^*$ via $x_1, x_2, \ldots, x_{d^*}$. We can do this without loss of generality by permuting the order of the parameters. Using these definitions, we can now define the reverse transformation as $F : [0,1]^{d^*} \to [0,1]^{d^*}$, where the indices of $F$ are given by

$$F_j(x) \triangleq G^*(X_j = x_j | X_1 = x_1, \ldots, X_{j-1} = x_{j-1}). \tag{C.106}$$

If random variables are distributed as $X \sim G^*$, then $F(X) \sim U[-1,1]^{d^*}$ (or equivalently $U[0,1]^{d^*}$ under the variable transformation $\tau(x) = 2x - 1$), since $X_1, X_2 | X_1, \ldots, X_{d^*} | X_1, \ldots, X_{d^*-1}$ are independent and

$$\prod_j F_j(X) = G^*(X). \tag{C.107}$$

Finally, with this notation set up, we can formally state our result for Case 1.

**Corollary 6** (Neural network sample complexity guarantee; generalization of Theorem 14)**.** *Let $1/e > \epsilon_1, \epsilon_2, \epsilon_3 > 0$, and let $\mathcal{D}$ be a distribution with PDF $g$ fulfilling assumptions (a)-(c) and $F$ according to Equation* (C.106)*. Let $f^{\Theta^*, w^*} : [-1, 1]^m \to \mathbb{R}$ be a neural network model produced from Algorithm 1 trained on data $\{(\hat{x}_\ell, \hat{y}_\ell)\}_{\ell=1}^N$ of size*

$$N = 2^{\mathcal{O}(\text{polylog}(1/\epsilon_1) + \text{polylog}(1/\epsilon_3))}, \tag{C.108}$$

*where the $x_\ell$'s form a low-discrepancy Sobol sequence, $\hat{x}_\ell = F^{-1}(x_\ell)$ and $|\hat{y}_\ell - \text{tr}(O\rho(\hat{x}_\ell))| \leq \epsilon_2$. Suppose that $f^{\Theta^*, w^*}$ achieves a training error of at most $((\epsilon_1 + \epsilon_2)^2 + \epsilon_3)/2$. Additionally, suppose that all parameters $\Theta_i^*$ of $f^{\Theta^*, w^*}$ satisfy $|\Theta_i^*| \leq W_{\max}$, for some $W_{\max} > 0$ that is independent of the system size $n$. Then the neural network $f^{\Theta^*, w^*}$ achieves prediction error*

$$\mathbb{E}_{x \sim \mathcal{D}} |f^{\Theta^*, w^*}(x) - \text{tr}(O\rho(x))|^2 \leq 2(\epsilon_1 + \epsilon_2)^2 + \epsilon_3. \tag{C.109}$$

Similarly, we also have a guarantee for Case 2, which is the version we state in the main text and the beginning of this appendix.

**Corollary 7** (Neural network sample complexity guarantee; generalization of Corollary 6 for random data)**.** *Let $1/e > \epsilon_1, \epsilon_2, \epsilon_3 > 0$, $\mathcal{D}$ a distribution with PDF $g$ satisfying assumptions (a)-(c). Let $f^{\Theta^*, w^*} : [-1, 1]^m \to \mathbb{R}$ be a neural network model produced from Algorithm 1 trained on data $\{(x_\ell, y_\ell)\}_{\ell=1}^N$ of size*

$$N = \log(1/\delta) 2^{\mathcal{O}(\text{polylog}(1/\epsilon_1) + \text{polylog}(1/\epsilon_3))}, \tag{C.110}$$

where the $x_\ell \sim \mathcal{D}$ and $|y_\ell - \mathrm{tr}(O\rho(x_\ell))| \leq \epsilon_2$. *Suppose that* $f^{\Theta^*, w^*}$ *achieves a training error of at most* $((\epsilon_1 + \epsilon_2)^2 + \epsilon_3)/2$. *Additionally, suppose that all parameters* $\Theta_i^*$ *of* $f^{\Theta^*, w^*}$ *satisfy* $|\Theta_i^*| \leq W_{\max}$, *for some* $W_{\max} > 0$ *that is independent of the system size* $n$. *Then the neural network* $f^{\Theta^*, w^*}$ *achieves prediction error*

$$\mathop{\mathbb{E}}_{x \sim \mathcal{D}} |f^{\Theta^*, w^*}(x) - \mathrm{tr}(O\rho(x))|^2 \leq 2(\epsilon_1 + \epsilon_2)^2 + \epsilon_3, \tag{C.111}$$

*with probability at least* $1 - \delta$.

We first prove Corollary 6 similarly to how we proved Theorem 14. In particular, we can prove a generalized version of Lemma 10, where we are given a low-discrepancy sequence with respect to $\mu^*$ and wish to bound the prediction error with respect to $\mathcal{D}$, as in Case 1. Define the prediction error of a neural network $f^{\Theta, w}$ with weights given by $\Theta, w$ with respect to a distribution $\mathcal{D}$ as

$$R_{\mathcal{D}}(\Theta) \triangleq \mathop{\mathbb{E}}_{x \sim \mathcal{D}} |f^{\Theta, w}(x) - \mathrm{tr}(O\rho(x))|^2 = \int_{[-1,1]^m} |f^{\Theta, w}(x) - \mathrm{tr}(O\rho(x))|^2 \, dG(x), \tag{C.112}$$

where $x \sim \mathcal{D}$ denotes $x$ sampled from the distribution $\mathcal{D}$ over $[-1, 1]^m$ and $dG(x) = g(x)dx$. Again, we suppress $w$ in the notation to avoid cluttering. Then, we have the following lemma.

**Lemma 11** (Generalized prediction error bound). *Let* $1/e > \epsilon_1, \epsilon_2 > 0$. *Consider a tanh neural network* $f^{\Theta, w} : [-1, 1]^m \to \mathbb{R}$ *with architecture defined in Definition 7 with weights* $\Theta_i \leq W_{\max}$ *for some* $W_{\max} > 0$ *independent of the system size* $n$ *and weights* $w$ *in the last layer. Assume that* $G$ *satisfies assumptions (a)-(c) and* $|y_\ell - \mathrm{tr}(O\rho(x_\ell))| \leq \epsilon_2$. *Furthermore, suppose we train* $f^{\Theta, w}$ *on data* $\{(x_\ell, y_\ell)\}_{\ell=1}^N$ *of size* $N$, *where the* $\tau^{-1}(x_\ell)$'s *from a set with star-discrepancy at most* $D_N^*(d; \mu^*)$ *in each dimension* $d$. *Then, we have*

$$R_{\mathcal{D}}(\Theta) \leq \hat{R}(\Theta) + \frac{\epsilon_1^2}{2} + \epsilon_2^2 + (\|w\|_1 + \|w\|_1^2) D_N^*(d^*; \mu^*) \cdot 2^{\mathcal{O}(\mathrm{polylog}(WW_{\max}/\epsilon_1))}. \tag{C.113}$$

*Moreover, if there exist constants* $b_1, b_2$ *such that* $D_N^*(d) \leq b_1 \sqrt{b_2 + \log(1/\delta)} \sqrt{\frac{d}{N}}$ *with probability at least* $1 - \delta$, *then there exists a constant* $\tilde{b}_1$ *such that*

$$R_{\mathcal{D}}(\Theta) \leq \hat{R}(\Theta) + \frac{\epsilon_1^2}{2} + \epsilon_2^2 + (\|w\|_1 + \|w\|_1^2) \tilde{b}_1 \sqrt{1 + \log(1/\delta)} \sqrt{\frac{\tilde{m}}{N}} \cdot 2^{\mathcal{O}(\mathrm{polylog}(WW_{\max}/\epsilon_1))} \tag{C.114}$$

*with probability at least* $1 - \delta$.

This lemma can be proven in the same fashion as Lemma 10 with two minor adjustments. One change is the use of a generalization of the Koksma-Hlawka inequality, which we discuss in Theorem 11 in Appendix A.2. To prove the second part of Lemma 11, we need a technical lemma (Lemma 14) to handle the probability of failure in the bound on the star-discrepancy. We relegate this to the end of the section, as it is mainly a technicality.

*Proof.* We proceed in the same way as in Lemma 10, replacing $\mathcal{R}(\Theta)$ with $\mathcal{R}_{\mathcal{D}}(\Theta)$ and replacing $\mathcal{R}_{\mathrm{loc}}(\Theta)$ with

$$R_{\mathrm{loc}, \mathcal{D}}(\Theta) \triangleq \mathop{\mathbb{E}}_{x \sim \mathcal{D}} \left| f^{\Theta, w}(x) - \sum_P \alpha_P f_P(x) \right|^2. \tag{C.115}$$

We follow the proof of Lemma 10 until Equation (C.75). This gives us

$$R_{\mathcal{D}}(\Theta) \leq \frac{\epsilon_1^2}{4} + \hat{R}_{\mathrm{loc}}(\Theta) + |R_{\mathrm{loc}, \mathcal{D}}(\Theta) - \hat{R}_{\mathrm{loc}}(\Theta)|, \tag{C.116}$$

where recall that

$$\hat{R}_{\mathrm{loc}}(\Theta) = \frac{1}{N} \sum_{\ell=1}^N \left| f^{\Theta, w}(x_\ell) - \sum_P \alpha_P f_P(x_\ell) \right|^2, \tag{C.117}$$

as in Equation (C.63). Moreover, we also have the adjusted version of Equation (C.75)

$$|R_{\mathrm{loc}, \mathcal{D}}(\Theta) - \hat{R}_{\mathrm{loc}}(\Theta)| \leq \sum_{P_1, P_2} |w_{P_1}||w_{P_2}| \left| \mathop{\mathbb{E}}_{x \sim \mathcal{D}} [f_{P_1}^{\theta_{P_1}}(x) f_{P_2}^{\theta_{P_2}}(x)] - \frac{1}{N} \sum_{\ell=1}^N f_{P_1}^{\theta_{P_1}}(x_\ell) f_{P_2}^{\theta_{P_2}}(x_\ell) \right|$$

$$+ 2|w_{P_1}||\alpha_{P_2}| \left| \mathop{\mathbb{E}}_{x\sim\mathcal{D}}[f_{P_1}^{\theta_{P_1}}(x)f_{P_2}(x)] - \frac{1}{N}\sum_{\ell=1}^{N} f_{P_1}^{\theta_{P_1}}(x_\ell)f_{P_2}(x_\ell) \right|$$

$$+ |\alpha_{P_1}||\alpha_{P_2}| \left| \mathop{\mathbb{E}}_{x\sim\mathcal{D}}[f_{P_1}(x)f_{P_2}(x)] - \frac{1}{N}\sum_{\ell=1}^{N} f_{P_1}(x_\ell)f_{P_2}(x_\ell) \right|. \qquad \text{(C.118)}$$

To bound the first term, we use the generalized Koksma-Hlawka inequality (Theorem 11) to obtain

$$\left| \mathop{\mathbb{E}}_{x\sim\mathcal{D}}[f_{P_1}^{\theta_{P_1}}(x)f_{P_2}^{\theta_{P_2}}(x)] - \frac{1}{N}\sum_{\ell=1}^{N} f_{P_1}^{\theta_{P_1}}(x_\ell)f_{P_2}^{\theta_{P_2}}(x_\ell) \right| \qquad \text{(C.119)}$$

$$= \left| \int_{[-1,1]^m} |f^{\Theta,w}(x) - \mathrm{tr}(O\rho(x))|^2 \, dG(x) - \frac{1}{N}\sum_{\ell=1}^{N} f_{P_1}^{\theta_{P_1}}(x_\ell)f_{P_2}^{\theta_{P_2}}(x_\ell) \right| \qquad \text{(C.120)}$$

$$= \left| \int_{[0,1]^m} f_{P_1}^{\theta_{P_1}}(\tau(\bar{x}))f_{P_2}^{\theta_{P_2}}(\tau(\bar{x}))\prod_{i=1}^{L}\bar{g}_i(\vec{\bar{x}}_i)d\bar{x} - \frac{1}{N}\sum_{\ell=1}^{N} f_{P_1}^{\theta_{P_1}}(\tau(\bar{x}_\ell))f_{P_2}^{\theta_{P_2}}(\tau(\bar{x}_\ell)) \right| \qquad \text{(C.121)}$$

$$= \left| \int_{\Omega_{P_1,P_2}} f_{P_1}^{\theta_{P_1}}(\tau(\bar{x}))f_{P_2}^{\theta_{P_2}}(\tau(\bar{x}))d\mu_{P_1,P_2}(\bar{x}) - \frac{1}{N}\sum_{\ell=1}^{N} f_{P_1}^{\theta_{P_1}}(\tau(\bar{x}_\ell))f_{P_2}^{\theta_{P_2}}(\tau(\bar{x}_\ell)) \right| \qquad \text{(C.122)}$$

$$\leq D_N^*(d^*;\mu^*)V_{HK}\left( (f_{P_1}^{\theta_{P_1}} \cdot f_{P_2}^{\theta_{P_2}})\circ\tau \right). \qquad \text{(C.123)}$$

Here, in the first equality, we use Assumption (c) on the structure of the PDF $g$ and use the variable transformation $\tau(x) = 2x - 1$. In the second equality, similarly to Lemma 10, we notice that because the functions in the expectation are local functions that only depend on parameters in $S_{P_1,P_2}$, we can replace the expectation over the whole domain $[0,1]^m$ with an expectation over just the domain $\Omega_{P_1,P_2}$. This step also crucially uses Assumption (c), where the factorization of the PDF $g$ due to independence is needed. The last line uses the generalized Koksma-Hlawka inequality (Theorem 11). The remainder of the proof follows in the same way as Lemma 10.

The second part of the statement is a direct consequence of Lemma 14. Specifically, following the proof of Lemma 10, we use Lemma 14 to bound the term in Equation (C.88). This is necessary because the upper bound on the star-discrepancy only holds probabilistically, so we must show that we can still use this upper bound on the sum of several star-discrepancy terms. This is more complicated than a simple union bound, and we relegate the proof and statement of Lemma 14 to the end of this section. $\qquad\square$

In addition to this lemma, we also need a result from [88], adapted to our definitions above. In the following, we use $D_N(\omega; d)$ to denote the discrepancy with respect to the Lebesgue measure of a specific sequence $\omega$ of length $N$ and dimension $d$, as in Definition 1. Similarly, we use $D_N(\omega; d; \mu)$ to denote the discrepancy with respect to a measure $\mu$ of a specific sequence $\omega$ of length $N$ and dimension $d$, as in Definition 4.

**Lemma 12** (Theorem 2 in [88]). *Let $\omega = \{x_\ell\}_{\ell=1}^{N}$ be an arbitrary sequence on the open $d$-dimensional unit cube with discrepancy $D_N(\omega, d)$, and let $\hat{\omega} = \{\hat{x}_\ell\}_{\ell=1}^{N}$ be the sequence defined by $F\hat{x}_\ell = x_\ell$, where $F$ is defined in Equation (C.106). Moreover, let $g$ be a strictly positive, $d$-times continuously differentiable PDF, such that $g(x) \geq m > 0$ for all $x$. Let $G$ be the corresponding probability measure (i.e., CDF). Furthermore, let $F$ satisfy*

$$\|F(x) - F(y)\| \leq K\|x - y\|. \qquad \text{(C.124)}$$

*Then, the discrepancy of $\hat{\omega}$ with respect to $G$ is bounded as*

$$D_N(\hat{\omega}; d; G) \leq c\left(D_N(\omega; d)\right)^{\frac{1}{d}}, \qquad \text{(C.125)}$$

*where $c = 2d \cdot 3^d (K+1)^{d-1}$.*

The authors in [88] note that the assumption on $F$ in Equation (C.124) is certainly fulfilled when $g$ is continuously differentiable. This is where Assumption (b) is used, where technically, we only require this Lipschitz condition on $F$. With Lemma 11 and Lemma 12, we are ready to prove Corollary 6.

*Proof of Corollary 6.* We proceed similarly as in the proof of Theorem 14 but this time using Lemma 11 instead of Lemma 10. First, we bound the nonuniform discrepancy of our training inputs $\hat{\omega} = \{\hat{x}_\ell\}_{\ell=1}^N$. Recall that $\hat{x}_\ell = F^{-1}(x_\ell)$ for $x_\ell$ generated according to a low-discrepancy Sobol sequence (i.e., low-discrepancy with respect to the Lebesgue measure). By definition of $F$, then $\hat{\omega}$ has star-discrepancy $D_N^*(\hat{\omega}; d^*; \mu^*)$. By Assumption (b), we can apply Lemma 12 to obtain

$$D_N^*(\hat{\omega}; d^*; \mu^*) \leq D_N(\hat{\omega}; d^*; \mu^*) \leq c\,(D_N(\omega; d^*))^{\frac{1}{d^*}} \leq 2^{d^*} c\,(D_N^*(\omega; d^*))^{\frac{1}{d^*}}. \tag{C.126}$$

Here, the first inequality follows because $D_N^*(d) \leq D_N(d)$, and the second follows by Lemma 12. Finally, the last inequality follows from $D_N(d) \leq 2^d D_N^*(d)$ (see, e.g., [93]). Because $d^* \leq 2\tilde{m}$, we can proceed as in the proof of Theorem 14 using the bound above.

By Theorem 9, we know that for Sobol sequences in base 2 with points in $[0, 1]^{d^*}$, the star-discrepancy is bounded by

$$D_N^*(d^*) \leq C(d^*)\frac{\log(N)^{d^*}}{N}, \tag{C.127}$$

where $C(d)$ is a constant such that

$$C(d) < \frac{1}{d!}\left(\frac{d}{\log(2d)}\right). \tag{C.128}$$

Since $C(d) = o(1)$, there exists a constant $C$, such that $C \geq C(d)$ for all $d > 0$. Using the assumption that the training objective is not larger than $((\epsilon_1 + \epsilon_2)^2 + \epsilon_3)/2$, by Lemma 11, we have

$$R_\mathcal{D}(\Theta^*) = \mathop{\mathbb{E}}_{x\sim\mathcal{D}} |f^{\Theta^*, w^*}(x) - \text{tr}(O\rho(x))|^2 \tag{C.129}$$

$$\leq \frac{\epsilon_1^2}{2} + \epsilon_2^2 + \frac{(\epsilon_1 + \epsilon_2)^2 + \epsilon_3}{2} + C'\frac{\log(N)2^{\mathcal{O}(\text{polylog}(W_{\max}/\epsilon_1))}}{N^{1/d^*}} \tag{C.130}$$

$$\leq 2(\epsilon_1 + \epsilon_2)^2 + \frac{\epsilon_3}{2} + C'\frac{\log(N)2^{\mathcal{O}(\text{polylog}(W_{\max}/\epsilon_1))}}{N^{1/\text{polylog}(1/\epsilon_1)}}, \tag{C.131}$$

where $C'$ is a constant. We also used here that $d^* \leq 2\tilde{m}$ and $\tilde{m} = |I_P| = \mathcal{O}(\text{polylog}(1/\epsilon_1))$. Since the training data has size $N = \mathcal{O}\left(2^{\text{polylog}(1/\epsilon_1)+\text{polylog}(1/\epsilon_3)}\right)$, $W_{\max}$ can be chosen with respect to $\epsilon_1, \epsilon_3$ and independent of the system size $n$ such that

$$C''\frac{\log(N)2^{\mathcal{O}(\text{polylog}(W_{\max}/\epsilon_1))}}{N^{1/\text{polylog}(1/\epsilon_1)}} \leq \frac{\epsilon_3}{4} \tag{C.132}$$

for some constant $C''$. In this way, we obtain

$$R_\mathcal{D}(\Theta^*) \leq 2(\epsilon_1 + \epsilon_2)^2 + \epsilon_3. \tag{C.133}$$

$\square$

Finally, we prove Case 2, where the training data is sampled i.i.d. according to a distribution $\mathcal{D}$ and the prediction error is also measured with respect to $\mathcal{D}$. Note that we can drop the assumption that $F^{-1}$ is efficiently computable for this case. The key result we need for this is a bound on the star-discrepancy for uniformly random points (Lemma 5).

*Proof of Corollary 7.* Let $\hat{x}_\ell = Fx_\ell$, where $F$ is as in Equation (C.106). As stated in [91, 92], $F$ transforms random variables with distribution $\mathcal{D}$ into standard uniform random variables. Hence, similarly to the proof of Corollary 6, if $\omega = \{x_\ell\}_{\ell=1}^N$ has star-discrepancy $D_N^*(\omega; d^*; \mu^*)$, we can bound it with respect to the discrepancy of $\hat{\omega} = \{\hat{x}_\ell\}_{\ell=1}^N$:

$$D_N^*(\omega; d^*; \mu^*) \leq D_N(\omega; d^*; \mu^*) \leq c\,(D_N(\hat{\omega}; d^*))^{\frac{1}{d^*}} \leq 2^{d^*} c\,(D_N(\hat{\omega}; d^*))^{\frac{1}{d^*}}. \tag{C.134}$$

Here, again we use $D_N^*(d) \leq D_N(d) \leq 2^d D_N^*(d)$ and Lemma 12. Then, by Lemma 5, for uniformly random points, i.e., $\hat{\omega} = \{\hat{x}_\ell\}_{\ell=1}^N$, we have

$$D_N^*(d) \leq 5.7\sqrt{4.9 + \log(1/\delta)}\sqrt{\frac{d}{N}} \tag{C.135}$$

with probability at least $1 - \delta$. The rest of the proof follows in the same way as Corollary 6, but using the above discrepancy bound. Note that because this discrepancy bound only holds probabilistically, we need to use the second part of Lemma 11. $\square$

Our prediction error bounds for Case 1 and Case 2 (in Corollaries 6 and 7, respectively) look rather similar. However, one can verify that Corollary 6 only requires about square root of the number of samples Corollary 7 uses to achieve a certain risk bound with small enough $\epsilon$, but this advantage is hidden in the polylogarithmic factors in the exponent. Hence, low-discrepancy data yields better theoretical guarantees. However, they did not yield an improvement in our numerical experiments, as discussed in Appendix D. The size $N$ of the training set seems to be very large for low-discrepancy data to have practical effects. In the main text, we present Corollary 7 because it is the more general theoretical statement.

In fact, one can also improve the polylogarithmic factors in Corollary 6 by imposing stronger assumptions on the distribution. In particular, the result in Lemma 12 seems surprisingly weak at first glance. Multidimensional transformations do, however, constitute a major challenge, since they generally do not preserve properties such as lines remaining straight or parallel. The boxes over which one optimizes in order to compute the discrepancy can thus change severely in shape, which can strongly alter the discrepancy and makes it difficult to analyze. This can result in a rather poor scaling in terms of the discrepancy with respect to the Lebesgue measure and thus $N$. However, when $g$ fulfills additional assumptions, we obtain a much better dependence on $N$ by directly applying the Koksma-Hlawka inequality (with respect to the Lebesgue measure) to $f \circ \phi$. Unsurprisingly, this is possible when the mixed derivative of $F^{-1} = \phi$ is bounded on $[0, 1]$. This follows, when $g$'s mixed derivative is bounded [88]. We restate this result below.

**Lemma 13** (Theorem 1 in [88]). *Let $\omega = \{x_\ell\}_{\ell=1}^N$ of size $N$ be an arbitrary sequence on the open $d$-dimensional unit cube with discrepancy $D_N(\omega, d)$ and $\hat{\omega} = \{\hat{x}_\ell\}_{\ell=1}^N$ the sequence defined by $F\hat{x}_\ell = x_\ell$, where $F$ is defined in Equation (C.106). Moreover, let $g$ be a strictly positive, $d$-times continuously differentiable PDF, such that $g(x) \geq m > 0$ for all $x$. Let $G$ be the corresponding probability measure (i.e., CDF). Furthermore, let $F$ satisfy*

$$\frac{\partial^{|A|} F_j}{\partial x_A} \leq M \qquad 1 \leq j \leq d, \quad A \subseteq \{1, \ldots, d\}. \tag{C.136}$$

*Then*

$$\left| \int_{[0,1^d]} f(\hat{x}) dG(\hat{x}) - \frac{1}{N} \sum_{\ell=1}^N f(\hat{x}_\ell) \right| \leq d! \left( \frac{M}{m} \right)^{2d-1} D_N^*(\omega; d) V_{HK}(f). \tag{C.137}$$

On a high level, the proof works via the observation that the Jacobian of $F$ has $g$ as its determinant, which is strictly positive. Since $F \circ \phi$ is the identity, one can write the Jacobian of $F \circ \phi$ as a linear system of equations with the derivatives of $\phi$ as solution. Using Cramer's rule and the assumption on $F$, one can upper bound the derivative of $\phi$. Applying this iteratively, one can show via induction that the mixed derivatives of $\phi$ are also bounded when the mixed derivatives of $F$ are bounded. Note that (as stated in [88]) when $\mathcal{D}$ is a product of independent distributions, i.e. $g(x) = \prod_{i=1}^m g_i(x_i)$ and fulfills assumptions (a)-(c), the conditions for Lemma 13 are also fulfilled. It is important to emphasize that the additional assumption used in Lemma 13 yield a much better dependence of $\epsilon$ on $N$. However, this improvement is hidden in the polylogarithmic factors.

We dedicate the last part of this section to the proof the following statement, which we used in the proof of Lemma 11. At a high level, this shows that when we have a probabilistic upper bound on the star-discrepancy, we can still upper bound a sum of star-discrepancies with high probability.

**Lemma 14.** *Suppose there exist constants $b_1, b_2$ such that $D_N^*(d) \leq b_1 \sqrt{b_2 + \log(1/\delta)} \sqrt{\frac{d}{N}}$ with probability $1 - \delta$. Then, there exists a constant $\tilde{b}_1$, such that for any $t > 0$*

$$\Pr\left( \sum_{P_1, P_2 \in S^{(geo)}} (c_1 |\alpha_{P_1}||\alpha_{P_2}| + c_2(|w_{P_1}||\alpha_{P_2}| + |w_{P_1}||w_{P_2}|)) D_N^*(x_{P_1, P_2}) \geq t \right) \tag{C.138}$$

$$\leq \exp\left( -\frac{Nt^2}{\tilde{b}_1(c_1\|\alpha\|_1^2 + c_2(\|w\|_1\|\alpha\|_1 + \|w\|_1^2))^2} \right), \tag{C.139}$$

*where recall $S_{P_1, P_2}$ is the set of parameters with coordinates in $I_{P_1} \cup I_{P_2}$ (Equation (A.2)), $c_1 = 2^{\mathcal{O}(\tilde{m}\log(\tilde{m}))}$ and $c_2 = 2^{\mathcal{O}(\tilde{m}\log(WW_{\max}) + \tilde{m}^2\log(\tilde{m}))}$. Thus $D_N^*(x_{P_1, P_2})$ denotes the star-discrepancy of this set of parameters in the training data.*

First, we introduce two useful tools for the proof.

**Theorem 17** (Azuma's Inequality for Martingales with Subgaussian Tails; Adapted from Theorem 2 in [94])**.** *Let $Z_1, Z_2, \ldots, Z_n$ be a martingale difference sequence with respect to a sequence $X_1, X_2, \ldots, X_n$, and suppose there are constants $b > 1$, $c_1, \ldots, c_n > 0$, such that for any $j$ and any $t > 0$ it holds that*

$$\Pr(Z_j > t | X_1, \ldots, X_{j-1}) \leq b \exp\left(-\frac{t^2}{c_j^2}\right). \tag{C.140}$$

*Then, it holds that*

$$\Pr\left(\sum_{j=1}^n Z_j > t\right) \leq \exp\left(-\frac{t^2}{28b \sum_{j=1}^n c_j^2}\right). \tag{C.141}$$

*Proof of Theorem 17.* Following the steps of the proof of Theorem 2 in [94], but taking the sum over $Z_j$ instead of the empirical average, we obtain for any $s > 0$

$$\Pr\left(\sum_{j=1}^n Z_j > t\right) \leq e^{-st} e^{7bc_n^2 s^2} \mathbb{E}\left[\prod_{j=1}^n e^{sZ_j} \middle| X_1, \ldots, X_{n-1}\right] \leq e^{-st+7bs^2 \sum_{j=1}^n c_j^2}. \tag{C.142}$$

We refer to [94] for further details of this calculation. Choosing $s = t/\left(14b \sum_{j=1}^n c_j^2\right)$, the expression above equals $e^{-t^2/\left(28b \sum_{j=1}^n c_j^2\right)}$, and we get the claim:

$$\Pr\left(\sum_{j=1}^n Z_j > t\right) \leq \exp\left(-\frac{t^2}{28b \sum_{j=1}^n c_j^2}\right). \tag{C.143}$$

$\square$

We also need the following two small lemmas.

**Lemma 15.** *Let $\delta > 0$. Let $X : \Omega_X \to \mathcal{X}$ and $Y : \Omega_Y \to \mathcal{Y}$ be independent random variables and $f : \mathcal{X} \times \mathcal{Y} \to \mathbb{R}_+$ be a function such that $\Pr_{XY}(f(X, Y) \geq t) \leq \delta$ for $t > 0$. Then,*

$$\mathbb{E}[f(X, Y)|X] \leq \frac{t}{2} \tag{C.144}$$

*with probability at least $1 - 2\delta$.*

*Proof.* First, we show that $\Pr(f(X, Y) \geq t | X) \geq 1/2$ with high probability. Then, we show that this implies the claim by Markov's inequality.

First, suppose for the sake of contradiction that $\Pr\left(\mathbb{E}[\mathbb{1}\{f(X, Y) \geq t\}|X] \geq \frac{1}{2}\right) > 2\delta$. Using the independence of $X$ and $Y$ applying Markov's inequality, we obtain

$$\Pr(f(X, Y) \geq t) = \mathbb{E}[\mathbb{E}[\mathbb{1}\{f(X, Y) \geq t\}|X]] \geq \frac{1}{2} \Pr\left(\mathbb{E}[\mathbb{1}\{f(X, Y) \geq t\}|X] \geq \frac{1}{2}\right) > 2\delta \cdot \frac{1}{2} = \delta, \tag{C.145}$$

which contradicts our initial assumption. Therefore, with probability at most $2\delta$ (w.r.t. $X$), $\Pr(f(X, Y) \geq t | X) \geq \frac{1}{2}$ and hence

$$\frac{1}{2} \leq \Pr(f(X, Y) \geq t | X) \leq \frac{\mathbb{E}[f(X, Y)|X]}{t} \tag{C.146}$$

by Markov's inequality. The result follows immediately.

$\square$

**Lemma 16.** *Let $j \in [m]$ be a coordinate of the parameters. Then, $|\{P \in S^{(\mathrm{geo})} : j \in I_P\}| = \tilde{m}$, where $\tilde{m} = \mathcal{O}(|I_P|)$.*

*Proof.* Recall that

$$I_P = \{c \in \{1, \ldots, m\} : d_{\mathrm{obs}}(h_{j(c)}, P) \leq \delta_1\}. \tag{C.147}$$

Fixing $c$ instead of $P$ also results in a set of geometrically local terms in a radius $\delta_1$ around a geometrically local term. Hence, the size of the set $\{P \in S^{(\mathrm{geo})} : j \in I_P\}$ also scales as $|I_P|$, which is at most $\tilde{m}$. □

Now we are able to provide a partial proof to Lemma 14.

**Lemma 17.** *Let $S_{P_1, P_2}$ be the set of parameters with coordinates in $I_{P_1} \cup I_{P_2}$ and let $x_{P_1, P_2} \triangleq \{\{x \in S_{P_1, P_2}\}_\ell\}_{\ell=1}^N$ denote the training data set only for these local parameters. If there exist constants $b_1, b_2$ such that $D_N^*(d) \leq b_1 \sqrt{b_2 + \log(1/\delta)} \sqrt{\frac{d}{N}}$ with probability at least $1 - \delta$, then*

$$\Pr\left(\sum_{P_2 \in S^{(\mathrm{geo})}} |\alpha_{P_2}| D_N^*(x_{P_1, P_2}) \geq t\right) \leq \exp\left(-\frac{Nt^2}{224\|\alpha\|_2^2(\tilde{m})^2 \exp(b_1)b_2}\right) \tag{C.148}$$

*for any $P_1 \in S^{(\mathrm{geo})}$ and any $t > 0$.*

*Proof.* Let $P_1 \in S^{(\mathrm{geo})}$. Define

$$X_j \triangleq \mathbb{E}\left[\sum_{P_2 \in S^{(\mathrm{geo})}} |\alpha_{P_2}| D_N^*(x_{P_1, P_2}) \middle| Y_{j-1}, \ldots, Y_0\right], \tag{C.149}$$

where we omit the dependence $x_{P_1, P_2} = x_{P_1, P_2}(Y_1, \ldots, Y_m)$ and $Y_j \triangleq \{(x_j)_\ell\}_{\ell=1}^N$ and $x_j$ parameterize $h_j$. We consider all increments, which are not contained in $I_{P_1}$. Hence, with slight abuse of notation, let index $j = 0$ refer to all coordinates in $I_{P_1}$ and $Y_0 \triangleq \{\{(x_j) : j \in I_{P_1}\}_\ell\}_{\ell=1}^N$. Furthermore, let

$$X_0 \triangleq \mathbb{E}\left[\sum_{P_2 \in S^{(\mathrm{geo})}} |\alpha_{P_2}| D_N^*(x_{P_1, P_2}) \middle| Y_0\right] \tag{C.150}$$

and $Z_1 \triangleq X_1 - X_0$. Clearly, $X_0, \ldots, X_m$ is a martingale sequence and $Z_1, \ldots, Z_m$ the respective martingale difference sequence. Furthermore, note that $X_m = \sum_{P_2 \in S^{(\mathrm{geo})}} |\alpha_{P_2}| D_N^*(x_{P_1, P_2})$ and by definition of $Y_0$, $j \notin I_{P_1}$ for all $j > 0$. Now, since $D_N^*(x_{P_1, P_2}) \geq 0$ and $|\alpha_{P_2}| D_N^*(x_{P_1, P_2})$ cancel out if $j \notin I_{P_2}$,

$$Z_j \leq \mathbb{E}\left[\sum_{P_2 \in S^{(\mathrm{geo})} : j \in I_{P_2} \setminus I_{P_1}} |\alpha_{P_2}| D_N^*(x_{P_1, P_2}) \middle| Y_{j-1}, \ldots, Y_0\right] \tag{C.151}$$

$$= \sum_{P_2 \in S^{(\mathrm{geo})} : j \in I_{P_2} \setminus I_{P_1}} |\alpha_{P_2}| \mathbb{E}\left[D_N^*(x_{P_1, P_2}) \middle| Y_{j-1}, \ldots, Y_0\right] \tag{C.152}$$

$$= \sum_{P_2 \in S^{(\mathrm{geo})} : j \in I_{P_2} \setminus I_{P_1}} |\alpha_{P_2}| \mathbb{E}\left[D_N^*(x_{P_1, P_2}) \middle| (Y_k)_{k < j, k \in I_{P_1}}\right]. \tag{C.153}$$

Then, for any $t > 0$, we have

$$\Pr(Z_j \geq t) \leq \Pr\left(\sum_{P_2 \in S^{(\mathrm{geo})} : j \in I_{P_2} \setminus I_{P_1}} |\alpha_{P_2}| \mathbb{E}\left[D_N^*(x_{P_1, P_2}) \middle| (Y_k)_{k < j, k \in I_{P_1} \cup I_{P_2}}\right] \geq t\right) \tag{C.154}$$

$$\leq \sum_{P_2 \in S^{(\mathrm{geo})} : j \in I_{P_2}} \Pr\left(\mathbb{E}\left[D_N^*(x_{P_1, P_2}) \middle| (Y_k)_{k < j, k \in I_{P_1} \cup I_{P_2}}\right] \geq \frac{t}{|\alpha_{P_2}|}\right) \tag{C.155}$$

$$\leq 2 \sum_{P_2 \in S^{(\mathrm{geo})} : j \in I_{P_2}} \Pr\left(D_N^*(x_{P_1, P_2}) \geq \frac{t}{2|\alpha_{P_2}|}\right) \tag{C.156}$$

$$\leq 2 \sum_{P_2 \in S^{(\text{geo})}: j \in I_{P_2}} \exp(b_1) \exp\left(-\frac{Nt^2}{4|\alpha_{P_2}|^2 b_2 \tilde{m}}\right) \tag{C.157}$$

$$\leq 2|\{P_2 \in S^{(\text{geo})} : j \in I_{P_2}\}| \exp(b_1) \exp\left(-\frac{Nt^2}{4b_2\tilde{m} \sum_{P_2 \in S^{(\text{geo})}: j \in I_{P_2}} |\alpha_{P_2}|^2}\right) \tag{C.158}$$

$$\leq 2\tilde{m} \exp(b_1) \exp\left(-\frac{Nt^2}{4b_2\tilde{m} \sum_{P_2 \in S^{(\text{geo})}: j \in I_{P_2}} |\alpha_{P_2}|^2}\right), \tag{C.159}$$

In the second line, we use a union bound. In the third line, we use Lemma 15 with $X = (Y_k)_{k<j, k \in I_{P_1} \cup I_{P_2}}$ and $Y = (Y_0, \ldots, Y_{j-1})$. In the fourth line, we use a rearrangement of the probabilistic upper bound on the star-discrepancy. In the last line, we use the definition of $\tilde{m}$. Now, by Theorem 17, for any $t > 0$

$$\Pr\left(\sum_{P_2 \in S^{(\text{geo})}} |\alpha_{P_2}| D_N^*(x_{P_1,P_2}) \geq t\right) \leq \exp\left(-\frac{t^2}{28b' \sum_{i=1}^m c_i^2}\right) \tag{C.160}$$

with $b' = 2\tilde{m} \exp(b_1)$ and $c_i^2 = \frac{4b_2\tilde{m}}{N} \sum_{P_2 \in S^{(\text{geo})}: i \in I_{P_2}} |\alpha_{P_2}|^2$. Furthermore,

$$\sum_{i=1}^m \sum_{P_2 \in S^{(\text{geo})}: i \in I_{P_2}} |\alpha_{P_2}|^2 = \sum_{P_2 \in S^{(\text{geo})}} |\alpha_{P_2}|^2 \sum_{i=1}^m \mathbb{1}\{i \in I_{P_2}\} = \tilde{m} \sum_{P_2 \in S^{(\text{geo})}} |\alpha_{P_2}|^2 = \tilde{m}\|\alpha\|_2^2. \tag{C.161}$$

The result follows from this. $\qquad\square$

Now, we are finally able to prove Lemma 14.

*Proof of Lemma 14.* We need to bound the weighted sum of the star-discrepancies we consider. This requires an extra step, since the star-discrepancy may vary among the sequences in the sum. Note that simply applying the union bound would result in a $\log(n)$-factor. Luckily, only the sum needs to be small, rather than all individual terms needing to be small at once. Recall that we use $S_{P_1,P_2}$ to denote the set of parameters with coordinates in $I_{P_1} \cup I_{P_2}$. In the following, we use $D_N^*(x_{P_1,P_2})$ to denote the star-discrepancy of $\{x \in S_{P_1,P_2}\}_{\ell=1}^N$, i.e., the training data points restricted to local parameters in $S_{P_1,P_2}$. We aim to apply Theorem 17 to $\sum_{P_1,P_2 \in S^{(\text{geo})}} |\alpha_{P_1}||\alpha_{P_2}| D_N^*(x_{P_1,P_2})$, $\sum_{P_1,P_2} |w_{P_1}||\alpha_{P_2}| D_N^*(x_{P_1,P_2})$ and $\sum_{P_1,P_2 \in S^{(\text{geo})}} |w_{P_1}||w_{P_2}| D_N^*(x_{P_1,P_2}))$.

For illustrative purposes, we only consider the first term for now. We proceed similarly to the proof of Lemma 17. Define

$$X_j \triangleq \mathbb{E}\left[\sum_{P_1,P_2 \in S^{(\text{geo})}} |\alpha_{P_1}||\alpha_{P_2}| D_N^*(x_{P_1,P_2}) \,\bigg|\, Y_{j-1}, \ldots, Y_1\right], \tag{C.162}$$

where the expectation is with respect to the inputs $Y_j \triangleq \{(x_j)_\ell\}_{\ell=1}^N$ and $x_j$ parametrize $h_j$. Furthermore, let

$$X_0 \triangleq \mathbb{E}\left[\sum_{P_1,P_2 \in S^{(\text{geo})}} |\alpha_{P_1}||\alpha_{P_2}| D_N^*(x_{S_{P_1,P_2}})\right] \tag{C.163}$$

and $Z_1 \triangleq X_1 - X_0$. Clearly, $X_0, \ldots, X_m$ is a martingale sequence and $Z_1, \ldots, Z_m$ the respective martingale difference sequence. Furthermore, $X_m = \sum_{P_1,P_2 \in S^{(\text{geo})}} |\alpha_{P_1}||\alpha_{P_2}| D_N^*(x_{P_1,P_2})$. Now, since $D_N^*(x_{P_1,P_2}) \geq 0$ and $|\alpha_{P_1}||\alpha_{P_2}| D_N^*(x_{P_1,P_2})$ cancel out if $j \notin I_{P_1} \cup I_{P_2}$,

$$Z_j \geq \mathbb{E}\left[\sum_{P_1,P_2 \in S^{(\text{geo})}: j \in I_{P_1} \cup I_{P_2}} |\alpha_{P_1}||\alpha_{P_2}| D_N^*(x_{P_1,P_2}) \,\bigg|\, Y_{j-1}, \ldots, Y_1\right] \tag{C.164}$$

$$= \sum_{P_1,P_2 \in S^{(\text{geo})}: j \in I_{P_1} \cup I_{P_2}} |\alpha_{P_1}||\alpha_{P_2}| \mathbb{E}\left[D_N^*(x_{P_1,P_2}) \,\bigg|\, Y_{j-1}, \ldots, Y_1\right] \tag{C.165}$$

$$= 2 \sum_{P_1 \in S^{(\text{geo})} : j \in I_{P_1}} \sum_{P_2 \in S^{(\text{geo})}} |\alpha_{P_1}| |\alpha_{P_2}| \, \mathbb{E}\left[ D_N^*(x_{P_1,P_2}) \Big| Y_{j-1}, \ldots, Y_1 \right]. \tag{C.166}$$

In the last step, we used the observation that for $j$ to be contained in $I_{P_1} \cup I_{P_2}$, it has to be contained in at least one of the two sets. Hence, we can enumerate the admissible coordinate sets by fixing $P_1$, such that $I_{P_1}$ contains $j$ and combine it with all $I_{P_2}$. The factor two arises from doing the same with $P_1$ when fixing $P_2$.

Now, for any $t > 0$, we have

$$\Pr(Z_j \geq t) \leq \Pr\left( 2 \sum_{P_1 \in S^{(\text{geo})} : j \in I_{P_1}} \sum_{P_2 \in S^{(\text{geo})}} |\alpha_{P_1}| |\alpha_{P_2}| \, \mathbb{E}\left[ D_N^*(x_{P_1,P_2}) \Big| Y_{j-1}, \ldots, Y_1 \right] \geq t \right) \tag{C.167}$$

$$\leq 2 \sum_{P_1 \in S^{(\text{geo})} : j \in I_{P_1}} \Pr\left( \sum_{P_2 \in S^{(\text{geo})}} |\alpha_{P_2}| D_N^*(x_{P_1,P_2}) \geq \frac{t}{4|\alpha_{P_1}|} \right) \tag{C.168}$$

$$\leq 2 \sum_{P_1 \in S^{(\text{geo})} : j \in I_{P_1}} \exp\left( -\frac{Nt^2}{16 \cdot 224 |\alpha_{P_1}|^2 \|\alpha\|_2^2 (\tilde{m})^2 \exp(b_1) b_2} \right) \tag{C.169}$$

$$\leq 2 |\{ P_1 \in S^{(\text{geo})} : j \in I_{P_1} \}| \exp\left( -\frac{Nt^2}{16 \cdot 224 \|\alpha\|_2^2 (\tilde{m})^2 \exp(b_1) b_2 \sum_{P_1 \in S^{(\text{geo})} : j \in I_{P_1}} |\alpha_{P_1}|^2} \right) \tag{C.170}$$

$$\leq 2\tilde{m} \exp\left( -\frac{Nt^2}{16 \cdot 224 \|\alpha\|_2^2 (\tilde{m})^2 \exp(b_1) b_2 \sum_{P_2 \in S^{(\text{geo})} : j \in I_{P_1}} |\alpha_{P_1}|^2} \right). \tag{C.171}$$

In second line, we use the union bound and Lemma 15 with $X = (Y_k)_{k<j, k \in I_{P_1} \cup I_{P_2}}$ and $Y = (Y_k)_{k>j, k \in I_{P_1} \cup I_{P_2}}$. In the third line, we use Lemma 17. In the last line, we use the definition of $\tilde{m}$. Applying Theorem 17 and bounding $\sum_j c_j^2$ exactly as in the proof of Lemma 17 yields

$$\Pr\left( \sum_{P_1, P_2 \in S^{(\text{geo})}} |\alpha_{P_1}| |\alpha_{P_2}| D_N^*(x_{P_1,P_2}) \geq t \right) \leq \exp\left( -\frac{Nt^2}{\tilde{b}_1 \|\alpha\|_2^4} \right). \tag{C.172}$$

One can similarly repeat this argument for the remaining terms of

$$\sum_{P_1, P_2 \in S^{(\text{geo})}} (c_1 |\alpha_{P_1}| |\alpha_{P_2}| + c_2 (|w_{P_1}| |\alpha_{P_2}| + |w_{P_1}| |w_{P_2}|)) D_N^*(x_{P_1,P_2})). \tag{C.173}$$

Using that $\|\alpha\|_2^2 \leq \|\alpha\|_1^2$ and solving for the appropriate $\delta$ yields the desired result. $\qquad \square$

### C.4 Bound on the mixed derivatives

Let $O = \sum_{P \in \{I,X,Y,Z\}^{\otimes n}} \alpha_P P$ be an observable that can be written as a sum of geometrically local observables. In the following, we derive an expression for the mixed partial derivatives of $\mathrm{tr}(P\rho(x))$, using tools from the spectral flow formalism [59, 60, 61]. This allows us to bound the Hardy-Krause variation (Equation (A.36)) in Appendix C.2. Let the spectral gap of $H(x)$ be lower bounded by some constant $\gamma$ for all choices of parameters $x \in [-1,1]^m$. Then, by the spectral flow formalism [59, 60, 61], the directional derivative of a ground state of $H(x)$ in the direction defined by the parameter unit vector $\hat{u}$ is given by

$$\frac{\partial}{\partial \hat{u}} \rho(x) = \hat{u} \cdot \nabla_x \rho(x) = i[D_{\hat{u}}(x), \rho(x)] \tag{C.174}$$

where

$$D_{\hat{u}}(x) = \int_{-\infty}^{+\infty} W_\gamma(t) e^{itH(x)} \frac{\partial H}{\partial \hat{u}}(x) e^{-itH(x)} dt, \tag{C.175}$$

and $W_\gamma(t)$ is defined by

$$|W_\gamma(t)| \leq \begin{cases} \frac{1}{2} & 0 \leq \gamma|t| \leq \theta, \\ 35e^2(\gamma|t|)^4 e^{-\frac{2}{7}\frac{\gamma|t|}{\log^2(\gamma|t|)}} & \gamma|t| > \theta. \end{cases} \tag{C.176}$$

The parameter $\theta$ is chosen to be the largest real solution of $35e^2(\gamma|t|)^4 \exp\left(-\frac{2}{7}\frac{\gamma|t|}{\log^2(\gamma|t|)}\right) = 1/2$.

This allows to us to obtain an expression of the first order derivative of $\rho(x)$ with respect to some parameter $x_k$. Consider the unit vector $\hat{u} = \hat{e}_k \triangleq (0, \dots 0, 1, 0, \dots 0)^T$, where the 1 is in the $k$th position. Then, the directional derivative in the direction given by $e_k$ is

$$\frac{\partial}{\partial \hat{e}_k}\rho(x) = \hat{e}_k \cdot \nabla_x \rho(x) = \frac{\partial}{\partial x_k}\rho(x) = i[D_{\hat{e}_k}(x), \rho(x)]. \tag{C.177}$$

Hence, we obtain

$$\frac{\partial}{\partial x_1}\operatorname{tr}(P\rho(x)) = \operatorname{tr}\left(P\frac{\partial}{\partial x_1}\rho(x)\right) = i\operatorname{tr}(P[D_{\hat{e}_1}(x), \rho(x)]) = i\operatorname{tr}([P, D_{\hat{e}_1}(x)]\rho(x)). \tag{C.178}$$

In order to compute the mixed derivative of second order, we now apply the product rule to this expression, which yields

$$\frac{\partial^2}{\partial x_1 \partial x_2}\alpha_P\operatorname{tr}(P\rho(x)) = \frac{\partial}{\partial x_2}i\alpha_P\operatorname{tr}([P, D_{\hat{e}_1}(x)]\rho(x)) \tag{C.179}$$

$$= i\alpha_P\left(\operatorname{tr}\left(\left[P, \frac{\partial}{\partial x_2}D_{\hat{e}_1}(x)\right]\rho(x)\right) - \operatorname{tr}\left([P, D_{\hat{e}_1}(x)]\frac{\partial}{\partial x_2}\rho(x)\right)\right) \tag{C.180}$$

$$= \alpha_P\operatorname{tr}\left(i\left[P, \frac{\partial}{\partial x_2}D_{\hat{e}_1}(x)\right]\rho(x)\right) - \operatorname{tr}([[P, D_{\hat{e}_1}(x)], D_{\hat{e}_2}(x)]\rho(x)). \tag{C.181}$$

Note that the terms of this expression can be treated similarly as the first partial derivative. For each additional partial derivative, we obtain terms consisting of the product with nested commutators with $\rho(x)$ under the trace. The nested commutators contain $D_{\hat{e}_j}(x)$ or partial derivatives of it, for which we will later derive an explicit form. Hence, we can apply the same scheme until we arrive at the $k$-th partial derivative. In order to formalize this statement, we need to introduce some additional notation.

Throughout the rest of this section, we use the notation $\frac{\partial^{|B|}}{\partial x_B}$ to denote the mixed derivative with respect to all parameters $x_i \in B$ for some set $B$.

**Definition 8.** *Let $k \in \mathbb{N}$. Let $A \subseteq [k]$ be a set of size $|A| = m$. Define an ordering $l_1 < l_2 < \cdots < l_m$ over the elements $l_1, l_2, \dots, l_m \in A$ of $A$. Then, we define*

$$\bigodot_{l\in A} \partial^{B_l} l \triangleq i^m \operatorname{tr}\left(\left[\left[\cdots\left[\left[P, \frac{\partial^{|B_{l_1}|}}{\partial x_{B_{l_1}}}D_{\hat{e}_{l_1}}(x)\right], \frac{\partial^{|B_{l_2}|}}{\partial x_{B_{l_2}}}D_{\hat{e}_{l_2}}(x)\right]\cdots\right], \frac{\partial^{|B_{l_m}|}}{\partial x_{B_{l_m}}}D_{\hat{e}_{l_m}}(x)\right]\rho(x)\right), \tag{C.182}$$

*where $B_j \subset [k]$. We refer to the nested commutators under the trace as* summands. *The set $A$ and collection $\{B_l\}_{l\in A} \triangleq \mathcal{B}_A$ satisfy the following conditions:*

1. *Each summand contains $D_{\hat{e}_1}(x)$.*

2. *The sets $A, B_1, \dots, B_m$ satisfy $A \cup \bigcup_{j=1}^m B_j = [k]$ and $A \cap B_j = \emptyset$, $B_i \cap B_j = \emptyset$.*

3. *For each $(B_l, l)$ pair, it holds that $i > l$ for all $i \in B_l$.*

This notation gives a compact way of expressing the terms of the mixed derivative and allows us to address each mixed derivative of the terms $D_{\hat{e}_j}$ individually. Each term $\bigodot_{l\in A} \partial^{B_l} l$ contains the product of $m+1$ matrix-valued functions (including $\rho(x)$, which depend on $x$. The set $A$ denotes the partial derivatives on the factor $\rho(x)$, which have been differentiated using Equation (C.174) when

applying the product rule. We will address the partial derivatives on $D_{\hat{e}_j}$ later in this section, when we derive an upper bound for $\bigodot_{l \in A} \partial^{B_l} l$.

The first condition underlines that the first partial derivative on $\rho(x)$ is necessarily computed via Equation (C.174) and thus contained in each term. The second condition reflects that each partial derivative operates on exactly one factor in each summand when applying the product rule. The third condition arises from the order, by which the partial derivatives are computed. For example, when we apply $\frac{\partial}{\partial x'_j}$ after $\frac{\partial}{\partial x_j}$, the $\frac{\partial}{\partial x_j} D_{\hat{e}'_j}$ can not occur in any term, since no term contained $D_{\hat{e}'_j}$ when the partial derivatives $\frac{\partial}{\partial x_j}$ were computed.

We can show that the mixed partial derivatives of $\alpha_P \operatorname{tr}(P\rho(x))$ can be written in terms of $\bigodot_{l \in A} \partial^{B_l} l$.

**Lemma 18** (Mixed derivative). *Let $\mathcal{A}_k = \{A \subseteq [k] : 1 \in A\}$ and $\mathcal{B}_A$ be as in Definition 8. The mixed derivative of the ground state property $\operatorname{tr}(P\rho(x))$ is given by*

$$\frac{\partial^k}{\partial x_1 \ldots \partial x_k} \alpha_P \operatorname{tr}(P\rho(x)) = \alpha_P \sum_{A \in \mathcal{A}_k} \sum_{(B_1, \ldots, B_{|A|}) \in \mathcal{B}_A} \bigodot_{l \in A} \partial^{B_l} l. \tag{C.183}$$

*Proof.* We proceed via induction. First, we verify that $\frac{\partial}{\partial x_k} \bigodot_{l \in A} \partial^{B_l} l$ with $A \in \mathcal{A}_{k-1}$ and $A \cup \bigcup_{j=1}^m B_j = [k-1]$ yield summands which fulfill the criteria for summands of the $k$-th partial derivative stated in Definition 8. Then, we show that each summand of the $k$-th derivative stems from a unique summand from the $(k-1)$-th partial derivative.

For the first part, let $|A| = m$. Furthermore, let

$$I_s(j) = \begin{cases} \{j\} & \text{if } s = k \\ \emptyset & \text{else} \end{cases}. \tag{C.184}$$

Then,

$$\frac{\partial}{\partial x_k} \bigodot_{l \in A} \partial^{B_l} l = \sum_{j=1}^m \bigodot_{l_j \in A} \partial^{B_{l_j} \cup I_j(l)} l + \bigodot_{l \in A \cup \{k\}} \partial^{B_l} l, \tag{C.185}$$

where each summand fulfills the properties, since $k > l$ for all $l \in A$.

Next, let $A' \in \mathcal{A}_k$ and let $\mathcal{B}_{A'}$ be the corresponding collection. Then, it is easy to see that, if $k \in B_{l_j}$, it stems from the summand $\bigodot_{l \in A'} \partial^{B_l} l$ with $B_{l_1}, \ldots, B_{l_j} \setminus \{k\}, \ldots, B_{l_m}$. If $k \in A'$, it stems from $\bigodot_{l \in A \setminus \{k\}} \partial^{B_l} l$. $\qquad\square$

With this expression in hand, we can move forward to bounding the mixed derivative, which we need in order to bound the Hardy-Krause variation. First, we will upper bound the number of terms in $\bigodot_{l \in A} \partial^{B_l} l$. Then, we derive an upper bound on the individual terms. For the first step, we exploit the conditions on the sets defining the mixed derivative. When we drop the third requirement in definition Definition 8, $|\mathcal{B}_A|$ corresponds to the number of ways of assigning $k - m$ distinct balls to $m$ bins. Thus, we obtain the following result on the number of terms $\bigodot_{l \in A} \partial^{B_l} l$ in the $k$-th mixed derivative.

**Lemma 19.** *Let $\mathcal{A}$ denote the set of all subsets of $[k]$. Then, the number of summands in the expression $\bigodot_{l \in A} \partial^{B_l} l$ is upper bounded by*

$$|\mathcal{B}_A| \le \sum_{s=1}^k \binom{k}{s} s^{k-s}. \tag{C.186}$$

*Proof.* There are $|\mathcal{A}| = \sum_{s=1}^k \binom{k}{s}$ different subsets of $[k]$. For each set $A$ with $|A| = s$, there are $s$ sets $B_l$. Dropping the third condition in Definition 8, we observe that each of the $k - s$ elements in $[k] \setminus A$ can be in any of the $s$ sets. Thus, we obtain the claimed upper bound. $\qquad\square$

In the next step, we aim to bound each individual term of the mixed derivative $\bigodot_{l \in A} \partial^{B_l} l$. Therefore, a crucial step is to bound the spectral norm of each factor. We first derive a preliminary result on the

mixed derivatives of the factors in $D_{\hat{e}_j}(x)$, which depend on $x$. This can be done using Duhamel's Formula for the derivative of the exponential map on $e^{H(x)}$, where we exploit that we only compute the derivative with respect to one parameter at a time, such that we can treat $H(x)$ as a function, which only depends on one parameter.

**Theorem 18** (Derivative of the exponential map; Theorem 3a in [95]). *Let $A(t) : \mathbb{R} \to \mathbb{C}^{n \times n}$. Then,*

$$\frac{d}{dt} e^{A(t)} = \int_0^1 e^{(1-s)A(t)} \left( \frac{dA(t)}{dt} \right) e^{sA(t)} ds. \tag{C.187}$$

**Lemma 20.** *Let $k \in [n]$, $B \subseteq [n] \setminus \{k\}$, such that $\left\| \frac{\partial^{|C|} h_j}{\partial x_C} \right\|_\infty \leq 1 \ \forall C \subseteq B \cup \{k\}$. Then*

$$\left\| \frac{\partial^{|B|}}{\partial x_B} \left( e^{itH(x)} \left( \frac{\partial h_j}{\partial x_k} \right) e^{-itH(x)} \right) \right\|_\infty \leq 2^{|B|+1} (|B| + 1)^{|B|+1} \tag{C.188}$$

*Proof.* By Theorem 18, the mixed derivative equals the sum of terms of the form

$$T = \int_0^1 \cdots \int_0^1 \prod_l f_l(s_l) ds_l \ldots ds_1, \tag{C.189}$$

where $f_l(s_l)$ can be any of $e^{(1-s_l)iH(x)}$, $e^{s_l iH(x)}$, $\frac{\partial^l h_j}{\partial x_{B_l}}$ or 1. By our assumption and the Cauchy-Schwartz inequality, each term $T$ satisfies $\|T\|_\infty \leq 1$. Furthermore, by the product rule, the number of terms is smaller than $\prod_{j=1}^{|B|} (2j + 1)$. Since each term of the $l$th partial derivative (including $k$) is the product of at most $2l + 1$ factors depending on $x$, such that the $(l + 1)$th derivative contains at most $2l + 1$-times as many factors. Thus, the number of terms is bounded above by

$$\prod_{j=1}^{|B|} (2j + 1) \leq \prod_{j=1}^{|B|+1} (2j) = 2^n n! \leq 2^{|B|+1} (|B| + 1)^{|B|+1}, \tag{C.190}$$

as required. $\qquad \square$

Now we can bound the terms $\bigodot_{l \in A} \partial^{B_l} l$.

**Lemma 21** (Bound components of the derivative). *Let $\bigodot_{l \in A} \partial^{B_l} l$ be as in Definition 8. Then*

$$\left| \bigodot_{l \in A} \partial^{B_l} l \right| \leq (2C_\gamma)^{|A|} \prod_{s=1}^{|A|} 2^{|B_{l_s}|+1} (|B_{l_s}| + 1)^{|B_{l_s}|+1}. \tag{C.191}$$

*Proof.* Recall that

$$D_{\hat{u}}(x) = \int_{-\infty}^{+\infty} W_\gamma(t) e^{itH(x)} \frac{\partial H}{\partial \hat{u}} (x) e^{-itH(x)} dt, \tag{C.192}$$

where $W_\gamma(t)$, such that

$$|W_\gamma(t)| \leq \begin{cases} \frac{1}{2} & 0 \leq \gamma|t| \leq \theta, \\ 35e^2 (\gamma|t|)^4 e^{-\frac{2}{7} \frac{\gamma|t|}{\log^2(\gamma|t|)}} & \gamma|t| > \theta, \end{cases} \tag{C.193}$$

where $\theta$ is chosen to be the largest real solution of $35e^2 (\gamma|t|)^4 \exp\left( -\frac{2}{7} \frac{\gamma|t|}{\log^2(\gamma|t|)} \right) = 1/2$. It is also useful to note that $\sup_t |W_\gamma(t)| = 1/2$.

By definition of the terms $\bigodot_{l \in A} \partial^{B_l} l$, the Cauchy-Schwartz inequality, and $\|[A, B]\|_\infty \leq 2\|A\|_\infty \|B\|_\infty$, we obtain

$$\left| \bigodot_{l \in A} \partial^{B_l} l \right| \leq \left( \int_{-\infty}^{+\infty} |W_\gamma(t)| dt \right)^{|A|} 2^{|A|} \prod_{s=1}^{|A|} \sup_t \left\| \frac{\partial^{|B_{l_s}|}}{\partial x_{B_{l_s}}} \left( e^{itH(x)} \left( \frac{\partial h_{j_s}}{\partial x_s} \right) e^{-itH(x)} \right) \right\|_\infty. \tag{C.194}$$

We bound each term individually. For the first term, we proceed in a similar manner as in [2] (Lemma 3). Namely, by Equation (S32) in [2], we can bound this integral by

$$\int_{t^*}^{+\infty} |W_\gamma(t)| dt \leq \frac{245}{2} e^2 \gamma^{-1} \left( \frac{1}{1 - \frac{35 \log^2(\gamma t^*)}{\gamma t^*}} \right) (\gamma t^*)^{10} e^{-\frac{2}{7} \frac{\gamma t^*}{\log^2(\gamma t^*)}} \triangleq C'_\gamma, \qquad \text{(C.195)}$$

by choosing $t^*$ such that $\gamma t^* = \max(5900, \alpha, 7(d+11), \theta)$ for some constant $\alpha$. Here, we use $C'_\gamma$ to denote a constant that depends only on $\gamma$. Moreover, since $|W_\gamma(t)| \leq \frac{1}{2}$, we can conclude that

$$\int_{-\infty}^{+\infty} |W_\gamma(t)| dt \leq \int_{-t^*}^{t^*} \frac{1}{2} dt + 2 \int_{t^*}^{+\infty} |W_\gamma(t)| \, dt \leq \frac{\max(5900, \alpha, 7(d+11), \theta)}{\gamma} + 2C'_\gamma \triangleq C_\gamma,$$
$$\text{(C.196)}$$

where $C_\gamma$ is also a constant that only depends on $\gamma$. By Lemma 20, we obtain the desired statement. $\square$

**Lemma 22** (Bounding the $k$-th mixed derivative). *The $k$-th mixed derivative of $\alpha_P \operatorname{tr}(P\rho(x))$ is bounded by*

$$\left| \frac{\partial^k}{\partial x_1 \ldots \partial x_k} \alpha_P \operatorname{tr}(P\rho(x)) \right| \leq |\alpha_P| 2^{\mathcal{O}(k \log(k))} \qquad \text{(C.197)}$$

*Proof.* First, we derive an upper bound on the terms $\bigodot_{l \in A} \partial^{B_l} l$, which is independent of $A$ and $\mathcal{B}_A$. Proceeding from the result of Lemma 21, we obtain

$$(2C_\gamma)^{|A|} \prod_{s=1}^{|A|} 2^{|B_{l_s}|+1} (|B_{l_s}| + 1)^{|B_{l_s}|+1} \leq (2C_\gamma)^{|A|} 2^k \prod_{s=1}^{|A|} k^{|B_{l_s}|+1} \leq (C_1 k)^k, \qquad \text{(C.198)}$$

where we used $|A| \leq k$ and $\sum_l (|B_l| + 1) = k$ and $C_1 = 4C_\gamma$. Furthermore, from Lemma 19, it is easy to see that $|\mathcal{B}_A| \leq k^k$. Thus, we obtain

$$\left| \frac{\partial^k}{\partial x_1 \ldots \partial x_k} \alpha_P \operatorname{tr}(P\rho(x)) \right| \leq |\mathcal{B}_A| |\alpha_P| (C_1 k)^k \leq |\alpha_P| C_1^k k^{2k} = |\alpha_P| 2^{\mathcal{O}(k \log(k))}. \qquad \text{(C.199)}$$

$\square$

Note that when deriving this result, we do not require that the parameters for the mixed derivatives are distinct. Assuming that $\|H(x)\|_{W^{k,\infty}([-1,1]^m)} \leq 1$, we can induce an order recover the above bound for any mixed derivative of order $k$.

**Corollary 8.** *If $\|H(x)\|_{W^{k,\infty}([-1,1]^m)} \leq 1$, then $\|\alpha_P \operatorname{tr}(P\rho(x))\|_{W^{k,\infty}} \leq |\alpha_P| 2^{\mathcal{O}(k \log(k))}$.*

*Proof.* Note that the bound from Lemma 22 is agnostic to the explicit directions $\hat{e}_j$ of the derivatives. Thus, we can choose any mixed derivative $\lambda \in \mathbb{N}_0^k$ such that $\sum_{j=1}^k \lambda_j = k$ and fix an order $o : \operatorname{dom}(\lambda) \to [k]$. Then, we can bound the mixed derivative on $o(\lambda)$ using the same approach as in Lemma 22 to obtain the bound. $\square$

# D Details of numerical experiments

In this section, we discuss the numerical experiments in detail.

## D.1 Experimental setup

As in [2], we consider the two-dimensional antiferromagnetic Heisenberg model with spin-$1/2$ particles placed on sites in a two-dimensional lattice. The corresponding Hamiltonian is

$$H = \sum_{\langle ij \rangle} J_{ij} (X_i X_j + Y_i Y_j + Z_i Z_j), \qquad \text{(D.1)}$$

where $\langle ij \rangle$ denotes all pairs of neighboring sites on the lattice. The coupling terms $J_{ij}$ correspond to the parameters $x$ of the Hamiltonian and are sampled uniformly from $[0, 2]$ (and then mapped to lie in $[-1, 1]$ for our ML algorithm). The goal of the numerical experiment is to predict the two-body correlation functions, i.e., the expectation value of

$$C_{ij} = \frac{1}{3}(X_i X_j + Y_i Y_j + Z_i Z_j) \tag{D.2}$$

for all neighboring sites $\langle ij \rangle$.

To this end, we generate data similarly to [1, 2], approximating the ground state and corresponding correlation functions for the Hamiltonian Equation (D.1) of different lattice sizes and choices of coupling parameters $J_{ij}$. We consider lattice sizes of $4 \times 5 = 20$ up to $9 \times 5 = 45$. For each lattice size, we generate two datasets of size 4096, one with uniformly randomly distributed $J_{ij}$ and one where the coupling parameters are distributed as a Sobol sequence. We obtained the data by approximating the ground state using the density-matrix renormalization group (DMRG) [96] based on matrix-product-states (MPS) [97], as has been done in [1, 2]. The simulations were performed on Nvidia T4 and A40 graphical processing units (GPUs). The former were used for lattice sizes from $4 \times 5$ up to $7 \times 5$ while the latter were used for lattice sizes $8 \times 5$ and $9 \times 5$. Depending on system size, we required between $\approx 50$ and 200 hours on the respective hardware component to simulate one dataset of size 4096.

Our deep learning model was also trained on Nvidia T4 and A40 GPUs. We trained the models for all respective correlation terms in parallel, by training a full model $f_{ij}^{\Theta, w}$ (we omit the indices for the model's parameters) for each term and minimizing the combined loss function

$$\sum_{\langle ij \rangle} \sum_{\ell=1}^{N} |f_{ij}^{\Theta, w}(x_\ell) - (C_{ij})_\ell|^2 \tag{D.3}$$

for the sake of time efficiency. For each data point, we trained a combined model for 500 epochs. For the terms of the local models $f_P^{\theta_P}$, as defined in Definition 6, we used fully connected deep neural networks with five hidden layers of width 200. For training, we used the AdamW optimization algorithm [83]. Depending on the system size and the amount of training data, this took between 0.5 and 20 hours. As a baseline, we compared against the best model from [2]. The code can be found at `https://github.com/marcwannerchalmers/learning_ground_states.git`.

## D.2 Additional experiments and discussion

In this section, we discuss the results of the numerical experiments and additional experiments performed that are not mentioned in the main text.

First, we perform additional experiments that analyze the scaling of the training/prediction error with respect to various parameters such as system size, local neighborhood size, and training set size (Figures 4 to 6). Importantly, in each of these, we see that the training error is small, as required by Theorem 5. Thus, as discussed in the main text, this assumption is satisfied in practice.

Moreover, as shown in Figure 2 (Left), the empirical prediction accuracy (RMSE) of the deep learning model is approximately constant with respect to the size of the lattice. Figure 4 (Right) further underlines this statement. The slight increase in prediction error for $\delta_1 > 0$ (size of the local neighborhood in Equation (A.2)) present in Figure 4 (Right) when increasing the system size from $4 \times 5$ to $5 \times 5$ may occur due to numerical errors in the data. From system size $5 \times 5$ onwards, we rather witness random fluctuations in test errors than a systematic increase.

Furthermore, we observe that the deep learning model significantly outperforms the regression model with random Fourier features from [2]. On the one hand, we notice that the performance of the latter could be improved, since the hyperparameters considered for hyperparameter tuning were selected for a substantially smaller dataset. This is underlined by the drop in RMSE for the regression model on Figure 5 for $\delta_1 = 1$, whereas a smaller RMSE is possible when choosing $\delta_1 = 0$. On the other hand, we think that the vast body of deep learning research also offers room for practical improvement of our deep learning model.

For $\delta_1 = 0$, we believe that our model achieves the best possible prediction error. For training set size larger than 2048, there is little improvement on the prediction error, as opposed to all experiments

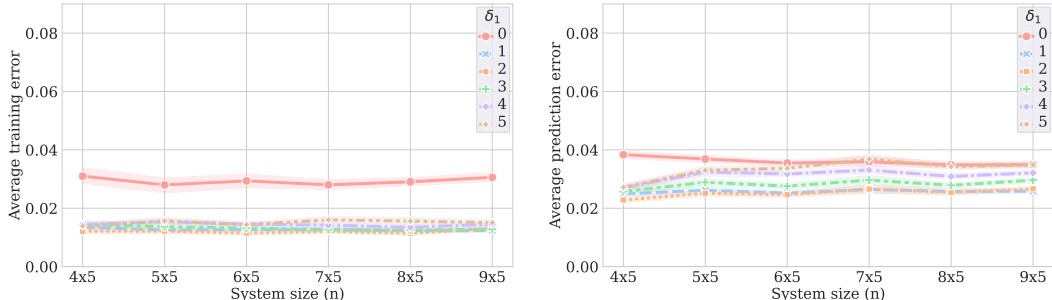

Figure 4: **Training/Prediction Error vs. System Size.** This figure shows the scaling of the training (left) and prediction (right) RMSE with respect to system size for different values of $\delta_1$. All training sets are distributed as Sobol sequences and were trained on $N = 3686$ samples. The shaded areas denote the 1-sigma error bars across the assessed ground state properties.

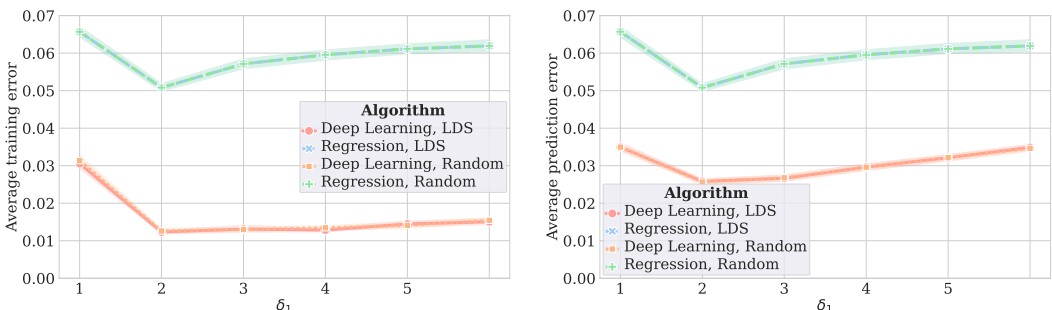

Figure 5: **Training/Prediction Error vs. Local Neighborhood Size.** This figure shows the scaling of the training (left) and prediction (right) RMSE with respect to the local neighborhood size $\delta_1$. All training sets are of size $N = 3686$ with system size $9 \times 5$. The shaded areas denote the 1-sigma error bars across the assessed ground state properties.

with $\delta_1 > 0$ (see Figure 2 (Center)). Furthermore, the training error remains relatively large compared to other choices of $\delta_1$ (see Figure 4). Hence, we conclude that the error arising from approximating the ground state property via local functions dominates the prediction error.

When increasing $\delta_1$, we witness an increase in prediction error, especially for small training sets. This is consistent with Lemma 10, which states that the bound on the prediction error is a combination of the training error and a term proportional to the star-discrepancy (and thus increases with the dimension of the domain of the local models). Our experimental results underline the balance which must be achieved between the two in order to obtain a small prediction error. This can clearly be observed in Figure 6. The training error decreases when increasing $\delta_1$ and increases with the size of the training set. Meanwhile, the test error increases when increasing $\delta_1$ and decreases with the size of the training set.

Another interesting observation is that the ML algorithm's performance on LDS seem to be almost the same as that of uniformly random points. We believe this is due to the dominance of the local approximation error for small $\delta_1$ and the drastic increase in dimensionality of the local models with increasing $\delta_1$ outweighing the benefit of using LDS in practice. The dominance of approximation error is also a possible explanation for the slight decrease in prediction error with respect to the system size in Figure 2 (Left) and Figure 5. For our concrete choice of lattice shape and ground state properties, the local approximation error may be decreasing with respect to system size. However, we do not expect this to be the case in general.

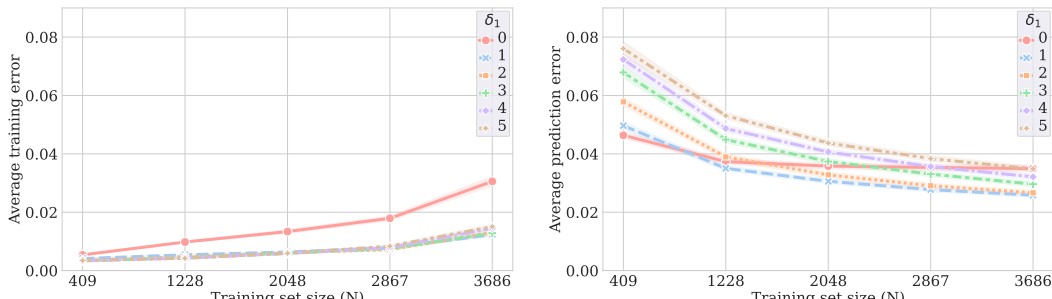

Figure 6: **Training/Prediction Error vs. Training Set Size.** This figure shows training (left) and prediction (right) RMSE with respect to training set size for different values of $\delta_1$. All training sets are distributed as Sobol sequences and the grid size is $9 \times 5$. The shaded areas denote the 1-sigma error bars across the assessed ground state properties.

## D.3 Experiments with non-geometrically-local Hamiltonians

In this section, we assess the necessity of the geometric locality assumption by conducting numerical experiments for non-geometrically-local systems. We conclude that geometric locality is necessary for our theoretical results.

We conduct experiments on for a Hamiltonian given by

$$H = \sum_{j<i} J_{ij}(X_i X_j + Y_i Y_j + Z_i Z_j). \tag{D.4}$$

The difference between this Hamiltonian and Equation (D.1) is that the sites $i$ and $j$ are not required to be neighboring, thus violating the geomtric locality assumption needed for our rigorous guarantees. We predict the same ground state properties as in the previous section, i.e., two-body correlation functions on neighboring sites. Our ML model still uses the local coordinate set $I_P$ from Equation (2.3). However, notice that the non-geometric-locality of the terms in the Hamiltonian impacts the number of parameters used. In other words, a larger number of parameters now affects a site in the neighborhood of each local Pauli. Furthermore, our adapted ML model assumes observables with 2-local terms[1]. Hence, the adapted ML model reads

$$\sum_{P \in S^{(2\text{-local})}} f_P^{\theta_P}. \tag{D.5}$$

Due to the lack of geometric locality and the larger number of terms of the Hamiltonian, the ground state properties are substantially harder to simulate, compared to the previous ones. We limit ourselves to uniformly random parameters and lattice shapes $4 \times 5, 5 \times 5$ and $6 \times 5$. The former two were simulated on Nvidia T4 GPUs and the latter on Nvidia A40 GPUs, using approximately $100 - 500$ hours per data set of size $4096$. We also notice that the approximation error due to MPS may be larger in this dataset than in the previous one. As for the previous results, we trained the models for each ground state property in parallel, by optimizing the sum of their training objectives. For the local models $f_P^{\theta_P}$, we used fully connected neural networks with five hidden layers of width $100$. This may not be optimal, but sufficient for the purpose of assessing the scaling of the prediciton error. We trained the models for different training set sizes using $\delta_1 = 0$. Since the adapted models consisted substantially more terms than the previous ones, training them for $500$ epochs took between $5$ and $35$ hours on Nvidia T4 GPUs for lattice shapes $4 \times 5$ and $5 \times 5$ and on Nvidia A40 GPUs on a $6 \times 5$-lattice.

In Figure 7 (Right), we witness system size-dependent prediction error for the smallest training set size we investigate. Since the respective training error is very small, the respective prediction errors arise due to overfitting. This effect diminishes for larger training sets. This is what one would expect when directly applying the techniques of our theoretical results to this setting. Since the number

---

[1] 2-local in the sense that $P$ does not act on more than two not necessarily neighboring sites.

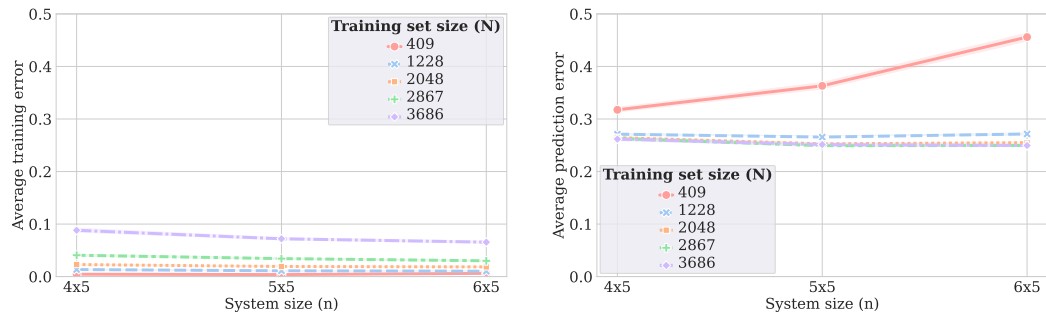

Figure 7: **Training/Prediction Error vs. System Size for Non-Geometrically-Local Systems.** This figure shows training (left) and prediction (right) RMSE with respect to the system size for the model given in Equation (D.4), which *violates* geometric locality. All training sets are of size $N = 3686$ and $\delta_1 = 0$. The shaded areas denote the 1-sigma error bars across the assessed ground state properties.

of terms increases quadratically in system size, the norm of the weights in the final layer can not be bounded by a constant anymore. Furthermore, the properties of the local approximation do not hold true anymore. Hence, the predictive capabilities of a model with $\delta_1 = 0$ may be more limited here than in the geometrically local case. However, the prediction error may also be impacted by possible numerical errors in the training data, as well as the architecture of the local deep neural networks. Overall, these experiments illustrate the necessity of the geometric locality assumption in our theoretical results.

