# OpenReview forum: "Predicting Ground State Properties: Constant Sample Complexity and Deep Learning Algorithms"
_NeurIPS.cc/2024/Conference — NeurIPS 2024 poster_

### Official Review · Reviewer_KHZF · 2024-07-11

**Soundness:** 3
**Presentation:** 2
**Contribution:** 3
**Rating:** 5
**Confidence:** 2

**Summary:**

The paper presents two novel machine learning algorithms for predicting ground state properties of quantum systems with constant sample complexity, independent of system size. The first algorithm modifies an existing ML model, while the second introduces a deep neural network model, both showing improved scaling in numerical experiments.

**Strengths:**

1. The introduction of a deep learning model with rigorous sample complexity bounds is a significant contribution to the field. The constant sample complexity, regardless of system size, is particularly noteworthy and addresses a critical challenge in quantum many-body physics.

2. The authors provide numerical experiments that validate the theoretical claims. The experiments demonstrate the practical effectiveness of the proposed algorithms, especially the deep learning model, which outperforms previous methods.

**Weaknesses:**

1. The training objective for the neural network is non-convex, which poses challenges in finding a global optimum efficiently. The paper does not address how to overcome this issue or guarantee convergence to optimal weights.

2. While the paper claims improved computational complexity, the actual implementation details and computational resources required for the deep learning model are not thoroughly discussed.

**Questions:**

1. The reviewer would appreciate if the authors could elaborate on how the performance of the deep learning model generalizes to Hamiltonians that extend beyond the specific cases examined in the numerical experiments.

2. Can the authors provide more insights into the practical implementation of their algorithms, particularly regarding the initialization and regularization procedures used during training? This will be helpful for readers reproduce the results of the paper.

**Limitations:**

The authors have discussed the limitations.

---

> ### Author Rebuttal · Authors · 2024-08-06
>
> **Reviewer Comment:** The training objective for the neural network is non-convex, which poses challenges in finding a global optimum efficiently. The paper does not address how to overcome this issue or guarantee convergence to optimal weights.
>
> **Author response:** To address non-convexity of the training objective, we refer to literature on overparametrized deep neural networks, Ref. [74], (see Lines 283-284 in our manuscript), where it is shown that Gradient Descent finds the global optimum in settings very close to ours. A similar result was proven for Stochastic Gradient Descent [Oymak et al., "Overparameterized Nonlinear Learning: Gradient Descent Takes the Shortest Path?" PMLR 2019]. Those results can likely be adapted to our setting.
> Furthermore, the experimental results show that this does not seem to be an issue in practice.
>
> **Reviewer Comment:** While the paper claims improved computational complexity, the actual implementation details and computational resources required for the deep learning model are not thoroughly discussed.
>
> **Author response:** The computational requirements and implementation details for generating the training data and training are detailed in Appendix D.1.
> Moreover, we provide code for the experiments in https://anonymous.4open.science/r/anonymous-B093, as referenced in Section 4.
>
>
> **Reviewer Question:** The reviewer would appreciate if the authors could elaborate on how the performance of the deep learning model generalizes to Hamiltonians that extend beyond the specific cases examined in the numerical experiments.
>
> **Author response:** The performance of the deep learning model holds for any Hamiltonian satisfying the conditions outlined in Section 2.1. Namely, this is any $n$-qubit gapped geometrically-local Hamiltonian that can be written as a sum of local terms such that each local term only depends on a constant number of parameters. There are many physically relevant Hamiltonians where this holds. Some examples are the 2D random Heisenberg model (as considered in the numerical experiments), the XY model on an $n$-spin chain with a disordered $Z$ field, and the Ising model on an $n$-spin chain with with a disordered $Z$ field.
>
> **Reviewer Question:** Can the authors provide more insights into the practical implementation of their algorithms, particularly regarding the initialization and regularization procedures used during training? This will be helpful for readers reproduce the results of the paper.
>
> **Author response:** We will add that we used Xavier initialization [71] and no regularization. The latter is implicitly shown in the loss function we used [Equation D.3]. The code for implementing the numerical experiments is given in https://anonymous.4open.science/r/anonymous-B093, as referenced in Section 4.

---

> > ### Comment · Area_Chair_pjyJ · 2024-08-13
> >
> > Dear Reviewer KHZF,
> >
> > The author-reviewer discussion period is ending soon. Please check if the authors’ response has addressed your concerns and feel free to adjust your score accordingly. If you find the authors’ response unsatisfactory, please explain your reasons and discuss them with the authors immediately.
> >
> > Best regards,
> > AC

---

> ### Comment · Reviewer_KHZF · 2024-08-13
>
> Thank you for your reply. I would like to keep my score.

---

### Official Review · Reviewer_MeLw · 2024-07-12

**Soundness:** 4
**Presentation:** 4
**Contribution:** 3
**Rating:** 7
**Confidence:** 3

**Summary:**

In this paper, the authors focused on utilizing deep learning methods to predict the ground states. They made an important assumption that brings theoretical improvement to achieve constant sample complexity in the training data. They also made two main alternations to the learning model compared to previous literature, including incorporating Pauli coefficients in feature mapping and utilizing kernel ridge instead of Lasso. Numerical results for up to 45 qubit systems are provided, supporting the theoretical findings.

**Strengths:**

1. High-quality paper with rigorous theoretical findings and comprehensive numerical results.
2. Improved the sampling overhead to constant complexity, independent of the system size.
3. Explored new possibilities for predicting the ground state properties of quantum many-body systems using neural network models.

**Weaknesses:**

The main concern is that the improvement of this paper against precious works is limited. The main theoretical finding is based on an additional assumption that we know the property we'd like to predict in advance. The proposed learning method has only two minor alternations. These issues prevent me from giving a higher evaluation score, but they do not overshadow the fact that this article is of high quality.

**Questions:**

No questions

**Limitations:**

See weakness

---

> ### Author Rebuttal · Authors · 2024-08-06
>
> We thank the reviewer for the careful reading of our paper. The reviewer's comments only apply to our first result in Section 3.1, and we acknowledge that the reviewer's comments are accurate for this portion of our paper. However, the improvement is significant - reducing sample complexity from logarithmic in system size to constant. Moreover the reviewer appears to have missed entirely  our second result in Section 3.2, we obtain a rigorous guarantee for *neural-network-based* algorithms predicting ground state properties.
>
> - This is the *first* work to give a practical deep learning based approach for predicting ground state properties with both theoretical guarantees and good experimental performance, and there is nothing similar in previous literature. We introduce several new techniques to this problem setting, such as the use of the Hardy-Krause variation, which is likely to be widely applicable in future research.
>
> - This neural network result does *not* require knowledge of the property (unlike the result in Section 3.1). In addition, we would like to comment that knowing the property in advance is a natural assumption, as often scientists will have a particular observable in mind that they would like to study (as further discussed in lines 168-175).
>
> Moreover, since our algorithm in Section 3.1 also applies to classical representations of the ground state (such as classical shadows), this can also be used to mitigate the issue of having to know the observable in advance (as we can learn a classical shadow of the state and then use that to predict properties of an observable of interest which need not be known in advance). See corollary 1.

---

> > ### Comment · Area_Chair_pjyJ · 2024-08-13
> >
> > Dear Reviewer MeLw,
> >
> > The author-reviewer discussion period is ending soon. Please check if the authors’ response has addressed your concerns and feel free to adjust your score accordingly. If you find the authors’ response unsatisfactory, please explain your reasons and discuss them with the authors immediately.
> >
> > Best regards,
> > AC

---

> > > ### Comment · Reviewer_MeLw · 2024-08-14
> > >
> > > Thanks to the authors for their response, and I have no further questions. I'd like to keep the score.

---

### Official Review · Reviewer_u4Mp · 2024-07-12

**Soundness:** 3
**Presentation:** 2
**Contribution:** 2
**Rating:** 3
**Confidence:** 4

**Summary:**

This paper studies the sample-efficient learnability of properties of grounds states of local Hamiltonians. Ground states of local Hamiltonians are hard to compute, even for quantum computers and to circumvent this hardness, several recent works proposed learning the trace inner product of local observables with the ground state given labeled training data. This setting is exactly PAC learning, i.e. given labeled data from a worst-case distribution, the goal is to get low prediction error wrt the same distribution on future samples. The best sample complexity for this problem is known to be log(n) 2^{polylog(1/eps)}, shown by Lewis et al.

The main questions addressed in this work are (1) whether the sample complexity can be improved to be independent of the system size and (2) whether there are rigorous guarantees for learning properties of the ground state via neural network based algorithms.

**Strengths:**

The paper provides several technical results on the representation and learnability of ground-state properties. The improved sample complexity follows from tweaking the algorithm is [2] and making additional assumptions about the training distribution. The neural net sample complexity result proceeds in two steps. First the authors prove an approximation-theoretic result for functions that look like ground state properties and show they can be well-approximated by neural networks. They then obtain a generalization bound using fairly sophisticated technical machinery.

**Weaknesses:**

I think the paper does not resolve the questions it claims to resolve and does so in a slightly camouflaged way.

1. Question 1 in the paper asks whether you can get sample complexity that is independent of system size for learning properties of ground states, aka the PAC learning setting for ground states of local Hamiltonians. The answer obtained is yes, under two crucial caveats, the observable is known in advance and the distribution over the training data is not worst-case.  This diverges significantly from the PAC learning model. The same critique holds for Question 2. Further, reference [1] does not state this as an open problem.

2. The assumptions on the training distribution are not stated upfront and do not appear to be mild. The assumptions include the distribution g to be strictly non-zero on [-1,1], and zero outside; g being continuously differentiable; and component-wise independent. Are there any natural distributions that satisfy all these properties simultaneously?

3. There is no discussion of why each of these properties is needed, which ones are crucial to the argument and which ones are for technical convenience. Having 4 non-trivial technical assumptions of the pdf is a major weakness, especially since it makes the result incomparable to prior work [1,2], where the setting is truly PAC learning.

4. In the numerical experiments, I see no discussion of what distribution was used to generate the training data, and how many of the technical conditions this distribution satisfies. It remains unclear to me what the experimental section is trying to convey, since it does not complement the main theorems.

5. When is it reasonable to expect labeled data for the observable / property you want to learn? What is a real-world scenario where one would expect to obtain such labeled data?

6. Is there a way to get some non-trivial sample complexity (not necessarily system independent) bound for PAC learning via neural networks, without the extra assumptions on the training distribution?

7. There are completely unjustified claims such as the neural network achieving low loss and finding a bounded magnitude solution after constant many steps and O(n) time. It is not clear why this should ever be the case.

**Questions:**

I included the questions with the weaknesses.

**Limitations:**

Yes.

---

> ### Author Rebuttal · Authors · 2024-08-06
>
> We thank the reviewer for the careful reading of our paper.\
> First, we would like to clarify some statements made by the reviewer.
> We remark that our first answer to Question 1, namely our result discussed in Section 3.1, does *not* make any assumptions on the data distribution (as in the PAC learning model) but does require that the observable is known in advance.
> In contrast, our Neural Network (NN) guarantee in section 3.2 introduces assumptions on the data distribution but does *not* require the observable to be known.
> Note that the assumptions on the data distribution for the NN guarantee (stated in [Line 1336-1340]) are satisfied by natural distributions such as Uniform and Gaussian distributions on $[-1,1]^m$. Hence, they are milder than in Ref. [1], where the guarantees hold exclusively for the uniform distribution. Those distributions often suffice in physically relevant models such as the Sachdev-Ye-Kitaev (SYK) model, which requires Gaussian parameters.
> Moreover, given that there are few rigorous guarantees for NNs in the literature, we view our work as a step in this direction.
> In the following, we will do our best to address each point of criticism by the referee in detail.
> 1. Mostly addressed above. Ref. [1] does state that "Rigorously establishing that NN-based ML algorithms can achieve improved prediction performance and efficiency for particular classes of Hamiltonians is a goal for future work'' in the sentence right above Section III.B.
> 2. We agree with the reviewer that we could emphasize and further discuss the assumptions on the distribution, although we do state them in the main text in Lines 234-238. We want to further emphasize that $g$ needs to be continuously differentiable **only on** $[-1,1]^m$.
> Examples of natural distributions satisfying all conditions simultaneously are uniform and the normal distribution.
> The latter is important for physical models such as the SYK model.
> 3. We note that our assumptions on the distribution do not make our work incomparable with Refs. [1, 2]. Ref. [1] applies only to the uniform distribution over $[-1,1]^m$, which is a *stronger* restriction than ours. Ref. [2] covers any arbitrary distribution, similar to PAC learning, as our first result in Section 3.1.
> In Section C.3, Line 1344, we mention that assumptions (a) and (b) can be relaxed. Assumption (a) ensures strict positivity on $[-1,1]^m$ to avoid divisions by zero, but this can be managed by splitting the integral on $g$'s support. Requiring $g=0$ outside $[-1,1]^m$ is natural since it is the parameter space. Assumption (b) can be relaxed to continuous $g$ using mollifiers. Assumption (c) on component-wise independence is necessary to reduce the input domains of $f_P^{\theta_P}$. We will add this in Section C.3.
> 4. We briefly mention the distributions used in the caption of Figure 2 and Lines 354-355 in the main text and discuss this further in Section D.1, line 1628-1630.
> Namely, we utilize points from the uniform distribution over $[-1,1]^m$ (as in the numerics in Refs. [1,2]) and low-discrepancy Sobol sequences, complementing Theorem 14.
> We respectfully disagree with the comment that the numerics do not complement our theorems.
> We reproduce the setup of the numerical experiments in Refs. [1,2] and compare the performances of our model and the previously-best-known algorithm from Ref. [2].
> The experiments illustrate that our method outperforms the method from Ref. [2] in practice (Figure 2, left).
> Moreover, they show that our assumptions in Theorem 5 are often satisfied in practice, i.e., small training error is achieved for proper choice of locality (Figure 2, right) and is independent of system size (Section D.2, Figure 4).
> 5. We briefly mention this in Lines 169-171, but agree it needs more discussion. The setting of receiving labeled data is consistent with Refs. [1, 2]. Our ML approach applies in scenarios like a scientist studying ground states of Hamiltonians $H(x)$. The scientist can use quantum experiments or classical simulations to investigate ground state properties. ML algorithms like ours and those in Refs. [1, 2] help extrapolate from existing data without additional costly experiments. In this scenario, it is realistic to assume the scientist can choose the training data, reducing the need to cover the worst-case distribution. Relevant examples of experiments generating data for quantum states include [A,B,C].
> 6. As discussed in point 3, the only assumption that is truly necessary for the argument is Assumption (c) (component-wise independence).
> Allowing long-range correlations for the parameters of a geometrically local system seems unnatural, so we are unsure whether this request makes sense physically. In that case, our methods yield a similar bound as in Ref. [1].
> 7. We justify those claims by referring to literature on overparametrized deep NNs Refs. [74], (see Lines 283-284 in the text), where it is shown that Gradient Descent finds the global optimum in settings very close to ours. A similar result was proven for Stochastic Gradient Descent [D]. Those results can likely be adapted to our setting.
> Moreover, our experimental results demonstrate that these claims are satisfied in practice in the setup we tested.
> The cost per iteration is $\mathcal{O}(n)$, due to the model's size. The model's size does not affect the number of gradient steps required for convergence (see e.g. Ref. [74], Theorem 5.1).
>
> ## References
>
> [A] Huang et al., "Quantum advantage in learning from experiments.", Science 2022 \
> [B] Struchalin et al., "Experimental estimation of quantum state properties from classical shadows." PRX Quantum 2021\
> [C] Elben et al., "Mixed-state entanglement from local randomized measurements." Physical Review Letters 2020.\
> [D] Oymak et al., "Overparameterized Nonlinear Learning: Gradient Descent Takes the Shortest Path?" PMLR 2019

---

> > ### Comment · Area_Chair_pjyJ · 2024-08-13
> >
> > Dear Reviewer u4Mp,
> >
> > The author-reviewer discussion period is ending soon. Please check if the authors’ response has addressed your concerns and feel free to adjust your score accordingly. If you find the authors’ response unsatisfactory, please explain your reasons and discuss them with the authors immediately.
> >
> > Best regards,
> > AC

---

### Official Review · Reviewer_cLpw · 2024-07-13

**Soundness:** 3
**Presentation:** 3
**Contribution:** 2
**Rating:** 5
**Confidence:** 1

**Summary:**

This work builds upon the work of Huang et al. and Lewis et al. by introducing two new approaches to get constant sample complexity for predicting properties of a ground state of a many-body local Hamiltonian. The two new approaches are a modified ML model that requires knowledge of the property of interest and a deep neural network model that does not need prior knowledge of the property. In this paper, the authors provide both proves and small experimental evaluations to show that both approaches achieve constant sample complexity, independent of system size.

**Strengths:**

The paper is well-organized and clearly written. The paper includes rigorous theoretical guarantees and small numerical experiments to confirm the efficacy of the proposed methods compared to the existing algorithm.

**Weaknesses:**

Even though it is a strong paper, the issue addressed here is a specific case that builds upon two other papers. Additionally, the related published works are mostly, if not all, published in physics journals. I do not see why the results shared in this paper are valuable to share with the broader NeurIPS community, especially since the mathematical proofs are very rigorous, I would expect that it is not accessible to the broader audience. Some assumptions and conditions required for the theoretical guarantees may also limit the applicability of the results.

**Questions:**

How do the parameters or phases of the random Heisenberg model affect the training performance?

**Limitations:**

Limitations are addressed in the paper.

---

> ### Author Rebuttal · Authors · 2024-08-05
>
> Our work aims to solve an important physics problem by leveraging machine learning. Thus, we expect it to be of broad interest to physicists, theoretical computer scientists, and machine learning practictioners, as our algorithms not only have rigorous proofs but are also readily implementable, as seen in the numerical experiments.
>
> - We identify the *first* result using deep learning for predicting ground state properties, with both theoretical guarantees and good experimental performance.  This is a very practical algorithm accessible to the broad community and the code is available via an anonymous repo. The assumptions are natural and realistic (as discussed in other responses below) and conclusions are of direct practical relevance to physicists and chemists in their experimental work.
>
> - We highlight that many papers of similar levels of mathematical rigor on topics in physics/quantum information have been present at ICML and NeurIPS in the past.  A few examplex are [A,B,C,D]. The recent ICML 2024 conference featured a workshop devoted to "AI for Science: Scaling AI for Scientific Discovery."
>
> In regards to the question about parameters/phases of the random Heisenberg model, could you please clarify what you mean by this? In the below, we try to answer to our understanding of your question, but please let us know if we misunderstood what you meant. The parameters of the Hamiltonian are inputs to the machine learning model (as a part of the training data). Although our rigorous guarantee only holds for training and testing points in the same quantum phase of matter (as in previous work), in the 2D random Heisenberg model, we may predict across phase boundaries. However, as seen in the numerical experiments, our algorithm still performs well.
>
> ## References
> [A] Yamasaki et al., "Learning with optimized random features: Exponential speedup by quantum machine learning without sparsity and low-rank assumptions." NeurIPS 2020\
> [B] Aaronson et al., "Online learning of quantum states." NeurIPS 2018\
> [C] Abbas et al., "On quantum backpropagation, information reuse, and cheating measurement collapse. NeurIPS 2024\
> [D] Michaeli et al., "Exponential Quantum Communication Advantage in Distributed Inference and Learning." ICML 2024

---

> > ### Comment · Reviewer_cLpw · 2024-08-12
> >
> > I thank the authors for the response. I increased my rating for the paper.

---

### Official Review · Reviewer_7tNg · 2024-07-13

**Soundness:** 1
**Presentation:** 2
**Contribution:** 3
**Rating:** 5
**Confidence:** 2

**Summary:**

In this work, the authors give two algorithms that predict (geometrically local) properties of ground states of gapped geometrically local Hamiltonians. This problem has been introduced by Huang et al. [HKT+22], and the previous best known algorithm is given by Lewis et al. [LHT+24], which uses $\log(n)$ samples, where $n$ is the number of qubits in the Hamiltonian. This paper further improves on the $log(n)$ sample complexity, and gives two algorithms that only use a constant number of samples. The first  algorithm is modified from the algorithm of [LHT+24], changing the regression part of the algorithm from LASSO to kernel ridge regression. The second algorithm uses deep neural network, having the advantage of not needing to know the observables in advance, but requires more restriction on the distribution of the Hamiltonian parameters. The authors complement their theoretical results with numerical simulations.

[HKT+22] Huang, Hsin-Yuan, Richard Kueng, Giacomo Torlai, Victor V. Albert, and John Preskill. "Provably efficient machine learning for quantum many-body problems." Science 377, no. 6613 (2022): eabk3333
[LHT+24] Lewis, Laura, Hsin-Yuan Huang, Viet T. Tran, Sebastian Lehner, Richard Kueng, and John Preskill. "Improved machine learning algorithm for predicting ground state properties." nature communications 15, no. 1 (2024): 895.

**Strengths:**

The work achieves the optimal sample complexity of the problem and is written in good English.

**Weaknesses:**

The part of preliminaries that are restating definition and result of [LHT24+], is not well written, and I believe has led to a critical bug to the first algorithm. In particular, Theorem 8 claims that for every $O\sum_{P} \alpha_P P$ that can be written as sum of geometrically observables, $\sum_{P} |\alpha| =O(1)$. However, the counterpart in  [LHT24+] has extra restrictions: $||O||_{infty}=1$ and $O$ need to be inside a radius of $R=O(1)$.  Therefore, where the authors uses Theorem 8 in equation (B.28) to bound the kernel, the result is incorrect since they do not have $R=O(1)$.

Other minor inconsistencies includes:

line 642: $S^{geo}$ not defined

line 660: $h_{c(j)}$ not defined

**Questions:**

Some typos:

line 121: geometrically [local] observable

line 148: || \omega || -> || w ||

**Limitations:**

The authors adequately addressed the limitations.

---

> ### Author Rebuttal · Authors · 2024-08-05
>
> **Bug**
> The claim by the reviewer is incorrect, as *our observables satisfy exactly the same conditions* as those considered in [LHT+24].\
> In particular, we state throughout the paper that
> - we only consider observables that satisfy $\lVert O\rVert_\infty \leq 1$, e.g., in lines 122, 225, 301, etc.
> - We state clearly (lines 121-122) that $O$ is an observable that can be written as a sum of geometrically local observables. In [LHT+24], requiring that $R = \mathcal{O}(1)$ is exactly the definition of geometric locality (see, e.g., Definition 5 in [LHT+24]). Thus, the two definitions are *exactly the same* and conditions of Corollary 4 in [LHT+24] still hold in our setting.
>
> We will clarify this explicitly in the final version.
>
> We define $S^{(\mathrm{geo})}$ later in line 654. We will make this change to introduce it earlier.
> The term $h_{j(c)}$ denotes the same as in [LHT+24]. This is discussed in lines 655-657 and we will make this explicit in the main text.
>
> We should also highlight our second result in section 3.2 which is completely different and uses deep learning techniques. This is a very practical algorithm (code available via anonymous repo) which has strong empirical performance and for which we give theoretical guarantees using innovative new techniques such as bounding the Hardy-Krause variation.

---

> > ### Comment · Area_Chair_pjyJ · 2024-08-13
> >
> > Dear Reviewer 7tNg,
> >
> > The author-reviewer discussion period is ending soon. Please check if the authors’ response has addressed your concerns and feel free to adjust your score accordingly. If you find the authors’ response unsatisfactory, please explain your reasons and discuss them with the authors *immediately*.
> >
> > Best regards,
> > AC

---

> > ### Comment · Reviewer_7tNg · 2024-08-13
> >
> > I would like to thank the authors for the response. I have raised my score.

---

> ### Author Response · Authors · 2024-08-13
>
> We thank 7tNg for acknowledging that their misunderstanding has been clarified and are happy that they are convinced of the soundness of the results. So, we wonder why the soundness score is still 1. Also, if they have any other issues we could address that leads to the overall score of only 5, we are happy to respond.

---

### Official Review · Reviewer_vDXc · 2024-07-13

**Soundness:** 4
**Presentation:** 4
**Contribution:** 3
**Rating:** 7
**Confidence:** 3

**Summary:**

The authors propose an ML based method to predict properties of ground states of quantum systems which comes with provable guarantees. Improving on recent work by Huang et al and Lewis et al, they give sample complexity bounds which are independent of the number of qubits. This approach is applicable when the observable one is trying to predict is predetermined. The authors also suggest a deep learning based approach for the case where the observable is not known in advance. They support their theoretical wok with numerical experiments.

**Strengths:**

The paper adresses an important probelm, and is well written and argued. The authors clearly explain the previous state of the art in ML based prediction of ground state properties, as well as their own contribution. Their proposed modification to the procedure suggested by Lewis et al., which results in Theorem 1 of the paper, seems interesting and worthwhile. Likewise the guarantees obtained for the training of a custom Neural Network architecture are intriguing from a learning theoretic perspective.

**Weaknesses:**

It is unclear to me how the Neural Network generalization result compares to known results in the literature- the setting which the authors study is quite specific and thus it is not easy to relate the result they obtained to those in the theoretical deep learning literature.

**Questions:**

How restrivtive is the assumption about the dependence of each local term on a fixed number of parameters? It would be instructive to give physically relevant examples where this does and does not hold.
Presumably the results would not apply to standard Neural Network architectures- what specifically would break in the analysis?
How does the sample complexity depend on the constant the authors assume the network weights are bounded by?

**Limitations:**

I believe the authors have adequately adressed the limitations of the paper.

---

> ### Author Rebuttal · Authors · 2024-08-05
>
> We thank the reviewer for the careful reading of our paper and their constructive comments.
>
> **Reviewer Comment:** It is unclear to me how the Neural Network generalization result compares to known results in the literature.\
> **Author Response:** This is the first rigorous sample complexity bound on a neural network model for predicting ground state properties. We extend the tools from Ref. [54] beyond low-discrepancy sequences and explicitly bound the Hardy-Krause variation, such that our result is comparable to Refs. [1,2].
>
> **Reviewer Question:** How restrivtive is the assumption about the dependence of each local term on a fixed number of parameters? It would be instructive to give physically relevant examples where this does and does not hold.\
> **Author Response:** The assumption that each local term depends on a fixed, i.e., system-size independent, number of terms is not particularly restrictive. Since each local term only acts on a system size-independent number of particles in the system, it is natural that this also holds for its parametrization.\
> Moreover, we note that this assumption is not something new to our work and was introduced in Ref. [2].
>
> There are many physically relevant examples where this holds. A few examples are are
> - the 2D random Heisenberg model (as considered in the numerical experiments).
> - the XY model on an $n$-spin chain with a disordered $Z$ field.
> - the Ising model on an $n$-spin chain with with a disordered $Z$ field (see, e.g., [A]) .
> - the Sachdev-Ye-Kitaev (SYK) model, which requires Gaussian parameters.
>
> We thank the referee for this comment and will add such examples to demonstrate its wide applicability.
>
> [A] Lieb et al., "Two soluble models of an antiferromagnetic chain." Annals of Physics 1961
>
> **Reviewer Question:** Presumably the results would not apply to standard Neural Network architectures- what specifically would break in the analysis?
>
> **Author Response:**
> - As mentioned in the discussion [starting from line 1660], the practical performance of our model can most likely be improved by applying the vast body of recent results in the theory of deep neural networks.
> - Approximation results like those we used for tanh neural networks also exist for the ReLU activation function [53] and require neural networks of depth $\mathcal{O}(\log{\frac{1}{\epsilon}})$. Directly bounding the respective Hardy-Krause variation does not work however, since ReLU does not have bounded partial derivatives around $0$. It may be possible to work around this issue by bounding it with mollifiers, but we leave this open for future work.
> - We believe that our analysis can be extended to standard architectures such as convolutional or residual neural networks, as long as the width and number of hidden layers does not exceed the that needed for fully connected neural networks (so that the Hardy-Krause variation is $\mathcal{O}(2^{\mathrm{polylog}(1/\epsilon)})$. We limited our attention to fully connected networks here, as bounding the Hardy-Krause variation of more elaborate architectures would have been more tedious than instructive.
>
> **Reviewer Question:** How does the sample complexity depend on the constant the authors assume the network weights are bounded by?\
> **Author Response:** One can see this dependence in Eq. (C.93) in the appendices.

---

> > ### Comment · Reviewer_vDXc · 2024-08-08
> >
> > I would like to thank the authors for their detailed response. I will leave my score unchanged.

---

### Author Rebuttal · Authors · 2024-08-06

We thank all the reviewers for their consideration and feedback. We're gratified to see appreciation from most reviewers: several described the work as important and appreciated the novelty of the deep learning approach with both theoretical guarantees and strong practical performance.

We believe that the negative reviews are mainly the result of misunderstanding that we correct and clarify in the detailed rebuttals below:
-  the technical objections of 7tNg and u4Mp are the result of misunderstanding that we have clarified in detail below. Our first result does not make any additional assumptions on the observables nor on the input distribution (but only that the observable of interest is known, which is reasonable, and which can be ameliorated via the use of shadows as we clarify below). Our neural network guarantee holds for natural distributions such as uniform or Gaussian. We extend the tools from Ref. [54] to more general distributions than low-discrepancy sequences. By exploiting physical structure and deriving an explicit bound on its Hardy-Krause variation, we show that our neural network algorithm achieves theoretical improvements over Refs. [1, 2] with respect to sample complexity. Experimentally, in the settings tested in Refs. [1, 2], our neural network algorithm also shows practical improvements. We emphasize that the deep learning approach is the *first* such result on this problem. We believe that our methods apply to broader classes of physical systems.
- Reviewer cLpw opines that it is a *strong* paper with *mathematically rigorous guarantees* but it comes from physics literature and is not relevant to the wider NeurIPS community. We note that AI for Science is one of the most active areas at premier ML conferences like NeurIPS and ICML today, and our work falls in this area. Our second result is also very practical (code available via anonymous repo) and shows strong empirical performance.

Among the positive reviews,
- Reviewer MeLw opines that the paper is of "high quality" and gives a score of 7, stating the main reason for not giving a higher score is that they consider the result somewhat incremental. Actually the improvement over previous work is significant - reducing sample complexity from logarithmic to constant. In addition, MeLw appears to have missed entirely, our second main result, namely the *first* deep learning approach to the problem which is not only practical and shows strong empirical improvement over previous work, and for which we also give rigorous results using innovative new techniques such as the Hardy-Krause variation with wide applicability.

---

### Decision · Program_Chairs · 2024-09-25

**Decision:**

Accept (poster)

**Comment:**

Regarding the potential impact of this paper on introducing rigorous machine learning techniques to quantum physics research, I recommend accepting it. Predicting ground state properties is a fundamental task in quantum physics. This paper is the first theoretical work on predicting ground state properties using deep learning and, modulo some assumptions that deserve further study, achieves state-of-the-art sample complexity results. Although there are several gaps between theory and practice in the proposed algorithms, potential concerns are explicitly raised and mostly discussed in the paper.

Below are some additional comments.
- Some discussion is needed to justify the bounded weights assumption in Theorem 5.
- The claim that "we cannot guarantee a priori that the network will indeed achieve a low training error" in Section 3.2 deserves further consideration. I agree with the claim, but I believe the reason is the agnostic nature of machine learning.
- Please consider revising the paper to avoid potential concerns and misunderstandings as those raised by Reviewer u4Mp.